# Guaranteed Optimal Compositional Explanations for Neurons

**Biagio La Rosa** [1]    **Leilani H. Gilpin** [1]

## Abstract

Compositional explanations are a family of methods that aim to describe the spatial alignment between neurons' receptive field activations and concepts through logical rules, typically computed via a search over all possible concept combinations. Since computing the spatial alignment over the entire state space is computationally infeasible, the literature commonly adopts assumptions related to the structure of the combinations and beam search to restrict the state space. However, beam search cannot provide any theoretical guarantees of optimality, and it remains unclear how close current explanations are to the true optimum. In this theoretical paper, we address this gap by introducing the first framework for computing guaranteed optimal compositional explanations over the entire state space spanned by the adopted assumptions. Specifically, we propose: (i) a decomposition that identifies the factors influencing the spatial alignment, (ii) a heuristic to estimate the alignment at any stage of the search, and (iii) the first algorithm that can compute optimal compositional explanations in a time comparable to exhaustive beam search. Using this framework, we demonstrate that 10-40% of explanations previously obtained with beam search are suboptimal when overlapping concepts are involved. Finally, we evaluate a beam-search variant guided by our proposed decomposition and heuristic, showing that it matches or improves runtime over prior methods while offering greater flexibility in hyperparameters and computational resources.

## 1. Introduction

Compositional explanations (Mu and Andreas, 2020) are a method for expressing the alignment between the locations

[1]Computer Science and Engineering Department, University of California, Santa Cruz, US. Correspondence to: Biagio La Rosa <bilarosa@ucsc.edu>.

*Proceedings of the $43^{rd}$ International Conference on Machine Learning*, Seoul, South Korea. PMLR 306, 2026. Copyright 2026 by the author(s).

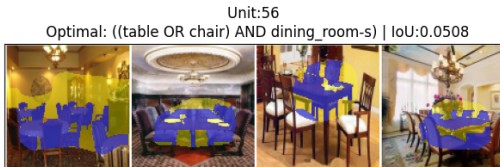

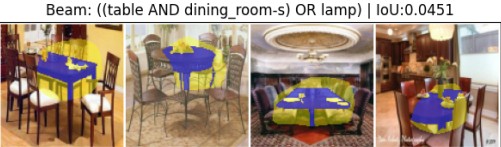

*Figure 1.* An example where the optimal algorithm finds a combination of concepts expressing the highest spatial alignment that beam search fails to capture. The colored areas indicate the activation captured (blue) and not captured (yellow) by the explanation.

of a given neuron activation range and the location of concepts, identified by annotations in datasets. The key idea is capturing the interaction between low-level (e.g., colors, textures, shapes) and high-level concepts (e.g., objects) via propositional logic formulas able to express the alignment between these complex relationships and neuron activations.

One of the key problems to achieve this goal is that the full search space encompassing all of the possible combinations between concepts cannot typically be exhaustively explored in a reasonable time due to its size. As a result, prior work has adopted multiple assumptions to restrict the structure of these combinations and relied on beam search to identify the highest spatial alignment within the restricted space (Mu and Andreas, 2020). However, because beam search restricts the exploration to a subset of the admissible formula state space (i.e., the one spanned by the best candidates at each beam level), it does not guarantee the optimality of the solution over the entire state space. Therefore, it is unclear whether the resulting explanations are, in fact, optimal (i.e., whether they identify the specific combination that expresses the highest absolute spatial alignment in the entire search space; see Figure 1) or how close they are to the optimal solution. Identifying and characterizing the optimal solution, and thus the ground-truth that previous algorithms approximate, is important for multiple reasons. First, the explanations produced by beam search may represent only

a subset of the alignment structure, offering a partial and potentially misleading view of neuron behavior. Second, if beam search converges to the optimal solution, this could indicate unstudied or unknown properties of the underlying datasets. Third, the availability of the ground-truth has the potential to guide and promote research on the development of novel algorithms that reduce approximation errors and mitigate the trade-off between approximation quality and runtime.

**This work contributes to this area by providing, for the first time, formal guarantees on compositional explanations.** This goal is achieved by making navigating the full state space tractable and the computation of the optimal solution feasible. We define *feasibility* as the ability to find the optimal solution in finite time and within the same order of magnitude as previous approaches under the same assumptions and settings adopted by previous literature for compositional explanations. To achieve this, we propose a decomposition of the Intersection over Union (IoU) metric that identifies a set of fundamental quantities governing alignment quality, and we design a heuristic and a corresponding algorithm that jointly reduce the size of the state space and guide the search process. As a first step, we apply our method to the computer vision domain and Convolutional Neural Networks, given their prominence in compositional explanation research. We leave for future work the extension of this class of method to other architectures (e.g., Transformers) and the application to additional domains.

Our contributions are as follows:

- We propose a decomposition of IoU score (dIoU) that identifies fundamental quantities governing the alignment and enables a better characterization of the impact of logical operators on spatial alignment.

- We design a heuristic and a corresponding algorithm that jointly reduce the size of the state space to be explored and guide the search process. We demonstrate that this algorithm computes guaranteed optimal explanations in a feasible time, and we analyze the differences between optimal and non-optimal explanations.

- We demonstrate that a portion of our proposed heuristic can be used to guide beam search, yielding significant gains in flexibility and achieving competitive or better performance than competitors. Specifically, our variant scales more effectively than competitors with respect to explanation length and beam size, and offers greater flexibility in computational resources.

We release the code at the following repository.

## 2. Related Work

Neuron explanations aim to understand what individual neurons learn during the training process. Different categories of methods have been proposed to decode different behaviors. Among the most popular in computer vision, we can cite the ones that generate samples that capture features recognized by a neuron (Erhan et al., 2009; Olah et al., 2017; Nguyen et al., 2016) and the ones that generate textual descriptions that correlate neuron activations and samples associated with a given concept (Hernandez et al., 2022; Oikarinen and Weng, 2023; 2024; Kalibhat et al., 2023; Ahn et al., 2024; Wang et al., 2022) through foundational models.

Differently from them, compositional explanations are a family of neuron explanations that specifically focus on the **spatial alignment** between the location of a neuron activation range and the location of concepts and express them through propositional logic formulas (refer to Section G for a more detailed description of the relationship between different methods studying the interpretability of neurons). The seminal work in this area is Network Dissection (Bau et al., 2017; 2020), which associates each neuron with the single concept that maximizes this alignment. This approach was extended by Mu and Andreas (2020) to associate relationships between multiple concepts, in an attempt to capture a higher degree of polysemantic behavior (Elhage et al., 2022; La Rosa, 2024). Relationships explored in the literature include co-occurrence (Bau et al., 2017), exclusion (Mu and Andreas, 2020), relative position (Harth, 2022), and hierarchy (Massidda and Bacciu, 2023). These relationships are typically extracted by search procedures based on vanilla (Mu and Andreas, 2020; Makinwa et al., 2022) or informed (La Rosa et al., 2023) beam search with a small beam size. However, beam search does not guarantee optimality, leaving open the question of whether better-aligned explanations exist.

Finally, in the context of probabilistic models, prior work (Nowozin, 2014; Li et al., 2013) has studied expectation-based and statistical approximations of the IoU score, showing their usefulness for probabilistic segmentation and structured prediction settings. Our decomposition is similar in spirit but derives an exact and deterministic decomposition of the IoU objective to design admissible heuristics, pruning strategies, and guaranteed-optimal search over logical explanations.

## 3. Optimal Compositional Explanations

**Preliminaries and Terminology** Let $\mathfrak{L}^1$ be a concept set including properties of interest for a given model and task (e.g., names of colors, objects, categories, shapes). Let $\mathfrak{D} = \{x_1, x_2, ..., x_n\}$ be a dataset, where each input $x_i$ has

dimension $d$ and is associated with a set of annotations (possibly empty) identifying the specific locations of concepts in that sample. Annotations are assumed to be provided with the dataset or automatically computed by an external model (La Rosa and Gilpin, 2025). In this paper, we use the term "location" to refer to the pair $(x, j)$ where $x$ identifies the position of a sample in the dataset and $j$ identifies the position of a feature in $x$. In vision tasks, these annotations correspond to segmentation masks that group features (i.e., pixels) semantically related, and a "concept" is the semantic label assigned by a human to that annotation. Note that the dataset can differ from the one used to train the model and is assumed to be used solely for neuron analysis. For this reason, $\mathfrak{D}$ is commonly referred to as a "probing dataset". The literature define the *Concept Tensor* $\mathbf{M}^{\mathfrak{L}^1} \in \{0,1\}^{|\mathfrak{L}^1| \times |\mathfrak{D}| \times d}$ as the binary tensor corresponding to the localization of each concept within the dataset samples, where an element in position $[k, x, j]$ is 1 if the annotations associated with the feature (i.e., pixel) in position $j$ of the sample $x$ include the concept $k$, and 0 otherwise. From the Concept Tensor, we can extract, for each concept $k$, the *Concept Matrix* $\mathbf{M}_k \in \{0,1\}^{|\mathfrak{D}| \times d}$. We use the notation $^1\cdot$ to indicate the set of locations equal to 1 in a binary matrix (e.g., $^1\mathbf{M}_k$).

Let $z$ be a neuron to be explained in a probed model, and let $d_z$ be the dimensions of its activations. In this paper, without loss of generality and following previous literature (Mu and Andreas, 2020), we assume $z$ denotes the feature maps of a neuron in a convolutional layer and its dimensions $d_z$ are transformed to match the input dimensions $d \in \mathbb{R}^2$ (e.g., through upscaling). From its activation, we can identify the *Neuron Activation Matrix* $\mathbf{N} \in \{0,1\}^{|\mathfrak{D}| \times d}$ as the binary matrix indicating the locations where the neuron fires within the dataset samples, where an element in position $[x, j]$ is 1 if the activation corresponding to position $j$ in sample $x$ lies within the considered activation range $[\tau_1, \tau_2]$, and 0 otherwise. $\tau_1$ and $\tau_2$ are typically chosen by considering the highest quantile in the activations (Bau et al., 2020) (commonly 0.005) or identifying semantic regions though clustering (La Rosa et al., 2023).

Let $\mathfrak{L}^n$ be the set of the admissible logical formulas (i.e., labels) of arity at most $n$ between concepts in $\mathfrak{L}^1$. Compositional explanations aim to assign to $z$ the logical combination $L \in \mathfrak{L}^n$ of concepts in $\mathfrak{L}^1$ (e.g., ((Cat OR Car) AND White)) that maximizes the alignment between the localization of a given neuron's activation range and the localization of the concepts within the probing dataset.

Formally, the algorithm identifies the label (i.e., formula) $L \in \mathfrak{L}^n$ that maximizes the following objective:

$$\arg\max_{L \in \mathfrak{L}^n} IoU(L, \mathbf{N}, \mathbf{M}) \qquad (1)$$

*Table 1.* Visualization of identified quantities for a sample $x$. In pink the unique extras. In yellow the common extras. In green the unique intersection. In cyan the common intersection.

| | | Vector | | | | | | $dIoU$ |
|---|---|---|---|---|---|---|---|---|
| N(x) | | 1 | 1 | 1 | 0 | 0 | 0 | |
| $\mathbf{M}_{c_1}[x]$ | | 1 | 1 | 0 | 0 | 1 | 1 | 2/5 |
| $\mathbf{M}_{c_2}[x]$ | | 1 | 1 | 0 | 1 | 0 | 0 | 2/4 |
| $\mathbf{M}_{c_3}[x]$ | | 1 | 0 | 1 | 0 | 1 | 1 | 2/5 |

where the Intersection Over Union ($IoU$) measures the overlap between label annotations and neuron activations, and it is defined as:

$$IoU(L, \mathbf{N}, \mathbf{M}) = \frac{|^1\mathbf{N} \cap \, ^1 \circ_L (\mathbf{M})|}{|^1\mathbf{N} \cup \, ^1 \circ_L (\mathbf{M})|} \qquad (2)$$

$|\cdot|$ indicates the cardinality of a set, and $\circ_L (\mathbf{M})$ is a function that returns the logical combination of the matrices in $\mathbf{M}$ of the concepts involved in the label $L$. The label $L$ is typically identified through a search algorithm.

To reduce the search space, compositional explanations typically make two assumptions: concepts in the explanation are distinct (**Assumption 1**), and concepts are combined incrementally to form labels (e.g., $((((A \oplus_1 B) \oplus_2 C)\ldots) \oplus_3 Z))$ (**Assumption 2**), where $\oplus$ indicates a logical connective. Even with these assumptions, $\mathfrak{L}^n$ is too large to be explored exhaustively. The total number of combinations is $\sum_{k=1}^{n} n_o^{k-1} \prod_{i=0}^{k-1}(|\mathfrak{L}^1| - i)$, where $n_o$ is the number of logic connectives, and each combination requires the comparison of $2d$ values to compute the alignment. In the settings considered by Mu and Andreas (2020), this leads to $2.8 \times 10^{14}$ operations, rendering both storage and runtime infeasible. To cope with this, prior work adopts beam search with a small beam size. However, beam search does not guarantee optimality, leaving open the question of whether the resulting compositional explanations are the best possible or if better-aligned explanations exist. In the following, we characterize the fundamental quantities that influence alignment and propose both a heuristic and an algorithm that guarantees the optimality of explanations.

### 3.1. Identifying Fundamental Quantities

This section introduces our proposed decomposition (dIoU) of the IoU score and the corresponding fundamental quantities (shown in Table 1). Alongside the assumptions mentioned earlier, we introduce an additional assumption (**Assumption 3**): *the logic operators connecting concepts are 00-preserving* (i.e., they cannot produce a 1 from two zeros). This assumption has been implicitly adopted in prior literature, where the commonly used bitwise OR ($\vee$), AND ($\wedge$), and AND NOT ($\neg \wedge$) logic operators all satisfy the 00-

preserving property. In this paper, we follow the previous literature and consider these operators as the connectives to chain concepts. Given this setup, we propose the definitions of the following quantities.

**Definition 3.1.** We define the set $U$ of *unique elements* of $\mathfrak{D}$ as the set of locations associated with exactly one annotation, and the set $C$ of *common elements* of $\mathfrak{D}$ as the set of locations associated with multiple annotations.

$$U = \{(x, j) \mid \exists! k \in \mathfrak{L}^1 \text{ s.t. } x \in \mathfrak{D}, j \in [d], \mathbf{M}^{\mathfrak{L}^1}[k, x, j] = 1\}$$
$$C = \{(x, j) \mid \exists k_1, k_2 \in \mathfrak{L}^1 \text{ s.t. } x \in \mathfrak{D},$$
$$\quad j \in [d], k_1 \neq k_2, \mathbf{M}^{\mathfrak{L}^1}[k_1, x, j] = 1, \mathbf{M}^{\mathfrak{L}^1}[k_2, x, j] = 1\}$$

In other words, unique elements are features associated with one and only one concept, while common elements are features associated with multiple concepts.

**Definition 3.2.** Given a neuron activation matrix $\mathbf{N}$, we define the *unique activation* set $N^U$ as the set of locations where the neuron fires on unique elements, and the *common activations* $N^C$ as the set of locations where the neuron fires on common elements.

$$N^U = \{(x, j) \mid (x, j) \in U, \mathbf{N}[x, j] = 1\}$$
$$N^C = \{(x, j) \mid (x, j) \in C, \mathbf{N}[x, j] = 1\}$$

**Definition 3.3.** Given a neuron activation matrix $\mathbf{N}$, a concept $k \in \mathfrak{L}^1$, and its Concept Matrix $\mathbf{M}_k$, we define the *unique intersection* set $I^U$ as the set of unique elements that are annotated with $k$ and where the neuron fires, and the *common intersection* set $I^C$ as the set of common elements annotated with $k$ where the neuron fires.

$$I^U(k) = \{(x, j) \mid \mathbf{M}_k[x, j] = 1, (x, j) \in N^U\}$$
$$I^C(k) = \{(x, j) \mid \mathbf{M}_k[x, j] = 1, (x, j) \in N^C\}$$

**Definition 3.4.** Given a neuron activation matrix $\mathbf{N}$, a concept $k \in \mathfrak{L}^1$, and its Concept Matrix $\mathbf{M}_k$, we define the *unique extras* set $E^U$ as the set of all unique elements in $\mathfrak{D}$ annotated with the concept $k$ and where the neuron does not fire, and the *common extras* set $E^C$ as the set of all common elements in $\mathfrak{D}$ annotated with the concept $k$ where the neuron does not fire.

$$E^U(k) = \{(x, j) \mid \mathbf{M}_k[x, j] = 1, (x, j) \in U, (x, j) \notin N^U\}$$
$$E^C(k) = \{(x, j) \mid \mathbf{M}_k[x, j] = 1, (x, j) \in C, (x, j) \notin N^C\}$$

Definition 3.3 and Definition 3.4 can be generalized to the case of a label $L \in \mathfrak{L}^n$. In this case, the binary label matrix $M_L$ is obtained as the result of the bitwise logic operations induced by the logic operators connecting single concepts $k \in L$.

**Definition 3.5.** Given a binary neuron activation matrix $\mathbf{N}$ and a label $L \in \mathfrak{L}^n$, the decomposed alignment between the annotations associated with $L$ and the neuron activations is defined as:

$$dIoU(\mathbf{N}, L, \mathfrak{D}, I^C, I^U, E^U, E^C) =$$
$$= \frac{\sum_{x \in \mathfrak{D}} |I^U(L)_x| + |I^C(L)_x|}{|^1\mathbf{N}| + \sum_{x \in \mathfrak{D}} |E^U(L)_x| + |E^C(L)_x|} \quad (3)$$

where the subscript indicates the quantity per sample.

**Lemma 3.6.** *Given a binary neuron activation matrix $\mathbf{N}$ and a label $L \in \mathfrak{L}^n$, the decomposed alignment between the annotations associated with $L$ and the neuron activations is equivalent to the IoU score if the logical operators involved in $L$ are 00-preserving:*

$$\frac{\sum_{x \in \mathfrak{D}} |I^U(L)_x| + |I^C(L)_x|}{|^1\mathbf{N}| + \sum_{x \in \mathfrak{D}} |E^U(L)_x| + |E^C(L)_x|} = \frac{|^1\mathbf{N} \cap {}^1\mathbf{M}_L|}{|^1\mathbf{N} \cup {}^1\mathbf{M}_L|} \quad (4)$$

*Proof.* See Section A. $\qquad\square$

Note that the purpose of $dIoU$ is not to provide a new metric (since it is equivalent to the IoU), but rather to reformulate an established metric in a way that allows for a more fine-grained analysis of the fundamental quantities governing the alignment.

*Observation* 1 (Impact of Operators on Quantities). Given the above definitions, we can quantify the impact of the bitwise logic operators connecting two concepts. The $\vee$ operator is 1-preserving (i.e., any 1 in either concept remains 1), and thus it preserves the common elements shared by the two concepts and sums the non-shared ones and unique elements. The $\wedge$ operator is 0-preserving (i.e., any 0 in either concept forces 0) and therefore removes all of the unique elements as well as common elements not shared by both concepts. Finally, the $\wedge\neg$ operator preserves all of the unique elements but removes the common elements shared by both concepts.

### 3.2. Heuristic

This section introduces our proposed heuristic. Given a label $L \in \mathfrak{L}^i$ s.t. $i \leq n$, the goal of this heuristic is to estimate the following quantities: (i) given a logic operator $\oplus$, a label $\mathfrak{L}^i$, and a concept $k$, the alignment of $L \oplus k$, (ii) the alignment reachable starting from $L$ and chaining additional concepts not yet included in $L$, up to a length of $n$.

#### 3.2.1. ESTIMATE LABEL QUANTITIES

To facilitate the estimation of the alignment of $L \oplus k$, we introduce the binary *Disjoint Matrix* $D \in \{0, 1\}^{|\mathfrak{L}^1| \times |\mathfrak{L}^1|}$, which encodes whether two concepts share any annotation overlap. Specifically, for every pair $(k_1, k_2)$ of concepts:

$$D[k_1, k_2] = \begin{cases} 1, & \text{if } {}^1\mathbf{M}_{k_1} \cap {}^1\mathbf{M}_{k_2} = \emptyset \\ 0, & \text{otherwise} \end{cases} \quad (5)$$

In addition, we introduce two new quantities, *Space for Common Extras* and *Space for Unique Extras*, derived from Definitions 3.1, 3.2 and 3.4, to characterize and constrain the available space for the set of extra elements:

$$SE^C = \{(x,j) \mid (x,j) \in C, \boldsymbol{N}[x,j] = 0\}$$
$$SE^U = \{(x,j) \mid (x,j) \in U, \boldsymbol{N}[x,j] = 0\} \tag{6}$$

By Assumption 2, we can separate the label into its left ($L_\leftarrow$) and right ($L_\rightarrow$) sides.

We can observe that unique elements are always disjoint elements and cannot be shared by Definition 3.1 . Therefore, their value can be computed exactly by using Observation 1. Regarding the common elements, we can use $D$ to check whether the left and right sides of $L$ are disjoint and estimate alignment differently in the two cases.

**Disjoint:** If the two sides are disjoint, then common elements follow the same principle as the unique elements since the sides do not share elements. Note that for the AND NOT operator, all the quantities converge to the $L_\leftarrow$, representing a mathematical equivalence and a degenerate case.[1] To force the algorithm to ignore these duplicate and degenerate cases, we manually set their dIoU to 0.

**Overlap:** If the two sides overlap, we can estimate the exact value of the unique quantities for OR and AND using the same equations as in the disjoint case. For the AND NOT operator, by Observation 1, the value equals the quantity of the left side. For the common quantities, we can derive them by combining the definitions in Section 3.1 and Observation 1, obtaining:

$$|I^C_{min}(L)_x| = \begin{cases} \max(|I^C_{min}(L_\leftarrow)_x|, \\ \quad |I^C(L_\rightarrow)_x|) & \text{if } \oplus = \vee \\ \max(|I^C_{min}(L_\leftarrow)_x|+ \\ \quad |I^C(L_\rightarrow)_x| - |N^C_x| \\ \quad , 0) & \text{if } \oplus = \wedge \\ \max(|I^C_{min}(L_\leftarrow)_x|- \\ \quad |I^C(L_\rightarrow)_x, 0) & \text{if } \oplus = \neg\wedge \end{cases} \tag{7}$$

$$|I^C_{max}(L)_x| = \begin{cases} \min(|I^C_{max}(L_\leftarrow)_x| \\ \quad +|I^C(L_\rightarrow)_x|, \\ \quad |N^C_x|) & \text{if } \oplus = \vee \\ \min(|I^C_{max}(L_\leftarrow)_x|, \\ \quad |I^C(L_\rightarrow)_x|) & \text{if } \oplus = \wedge \\ \min(|I^C_{max}(L_\leftarrow)_x|, \\ \quad |N^C_x| - |I^C(L_\rightarrow)_x)| & \text{if } \oplus = \neg\wedge \end{cases} \tag{8}$$

$$|E^C_{min}(L)_x| = \begin{cases} \max(|E^C_{min}(L_\leftarrow)_x|, \\ \quad |E^C(L_\rightarrow)_x|) & \text{if } \oplus = \vee \\ \max(|E^C_{min}(L_\leftarrow)_x|+ \\ \quad |E^C(L_\rightarrow)_x|- \\ \quad |SE^C_x|, 0) & \text{if } \oplus = \wedge \\ \max(|E^C_{min}(L_\leftarrow)_x|- \\ \quad |E^C(L_\rightarrow)_x|, 0) & \text{if } \oplus = \neg\wedge \end{cases} \tag{9}$$

$$|E^C_{max}(L)_x| = \begin{cases} \min(|E^C_{max}(L_\leftarrow)_x|+ \\ \quad |E^C(L_\rightarrow)_x|, \\ \quad |SE^C_x|) & \text{if } \oplus = \vee \\ \min(|E^C_{max}(L_\leftarrow)_x|, \\ \quad |E^C(L_\rightarrow)_x|) & \text{if } \oplus = \wedge \\ \min(|E^C_{max}(L_\leftarrow)_x|, \\ \quad |SE^C_x| - |E^C(L_\rightarrow)_x|) & \text{if } \oplus = \neg\wedge \end{cases} \tag{10}$$

Note that, since $L_\rightarrow$ is always an atomic concept (by Assumption 2), its quantities are exact. A detailed derivation is provided in Section E.

**Aggregated Computation** The above quantities are defined per-sample, requiring $|\mathfrak{D}|$ comparisons for each computation. This can still be costly for large state spaces. To mitigate this, we introduce a lighter aggregated computation, obtained by summing the values per label. For example, $\sum_{x \in \mathfrak{D}} \min(|E^C_{max}(L_\leftarrow)_x|, |SE^C_x| - |E^C(L_\rightarrow)_x|)$ can be transformed in $\min(\sum_{x \in \mathfrak{D}} |E^C_{max}(L_\leftarrow)_x|, \sum_{x \in \mathfrak{D}} |SE^C_x| - \sum_{x \in \mathfrak{D}} |E^C(L_\rightarrow)_x|)$. The trade-off is precision versus efficiency: the sample version is more accurate but computationally intensive, while the aggregated version can be pre-computed once per label, reducing the cost to a single comparison per quantity at the expense of precision (see Section C for details).

### 3.2.2. ESTIMATE PATHS

In this section, we estimate the maximum IoU achievable by concatenating an arbitrary number of concepts to a label. If the explanation length were unbounded, this maximum would converge to 1, making the heuristic uninforma-

---

[1]For example, "cat AND NOT dog" is always true, and the right side does not add meaningful information to the explanation.

tive. However, compositional explanations are inherently bounded, as long explanations would not aid user understanding; in practice, the maximum length is typically small (i.e., 3). This boundedness allows us to produce tighter estimates. Given a label $L$, our goal is to estimate the numerator and denominator in Definition 3.5 for all possible **paths obtained by concatenating additional concepts through logic operators**. To this end, we introduce the *maximum and minimum improvement*. For each sample, we extract the $n$ highest and $n$ lowest values for each quantity and compute their cumulative sums into the $Top$ and $Bott$ vectors, starting from the largest or smallest values, respectively. For example, $Top_k(I^C)_x$ represents the cumulative sum of the common intersection values of the $k$ concepts with the highest scores for that quantity. These quantities, combined with Observation 1 and the equations in Section 3.2.1, allow us to compute the maximum and minimum factors for $\vee$, $\wedge$, and $\wedge\neg$ exclusive paths (i.e., paths where only one operator is used from that point onward to concatenate additional concepts and at least 1 concept is added to the label):

$$|I_{min}(L)_x| = \begin{cases} \max(|I^C_{min}(L)_x| + |I^U(L)_x|, \\ \quad Bott_1(I^C)_x + Bott_1(I^U)_x) & \vee \text{ Path} \\ 0 & \wedge \text{ Path} \\ |I^U(L)_x| & \neg\wedge \text{ Path} \end{cases}$$
(11)

$$|I_{max}(L)_x| = \begin{cases} \min(|I^C_{max}(L)_x| + Top_t(I^C)_x, \\ \quad |N^C_x|) + \min(|I^U(L)_x| + \\ \quad Top_t(I^U)_x, |N^U_x|) & \vee \text{ Path} \\ \min(|I^C_{max}(L)_x|, Top_1(I^C)_x) & \wedge \text{ Path} \\ |I^U(L)_x| + \min(|I^C_{max}(L)_x|, \\ \quad |N^C_x| - Bott_1(I^C)_x) & \neg\wedge \text{ Path} \end{cases}$$
(12)

$$|Union_{min}(L)_x| = |^1N_x| +$$
$$\begin{cases} \max(|E^C_{min}(L)_x| + |E^U(L)_x|, \\ \quad Bott_1(E^C)_x + Bott_1(E^U)_x) & \vee \text{ Path} \\ 0 & \wedge \text{ Path} \\ |E^U(L)_x| & \neg\wedge \text{ Path} \end{cases}$$
(13)

$$|Union_{max}(L)_x| = |^1N_x| +$$
$$\begin{cases} \min(|E^C_{max}(L)_x| + Top_t(E^C)_x, \\ \quad |SE^C_x|) + \min(|E^U(L)_x| \\ \quad + Top_t(E^U)_x, \\ \quad |SE^U_x|) & \vee \text{ Path} \\ \min(|E^C_{max}(L)_x|, Top_1(E^C)_x) & \wedge \text{ Path} \\ |E^C_{max}(L)_x| + \min(|E^U(L)_x|, \\ \quad |SE^C_x| - Bott_1(E^C)_x) & \neg\wedge \text{ Path} \end{cases}$$
(14)

where $t$ denotes the difference between the maximum length and the length of the label $L$. To estimate the values for paths involving multiple operators, we take the maximum and minimum of each quantity across the operators considered (see Section E.3 for a discussion about explicitly modeling every possible combination). To these paths, we also add the *final path*, computed by Definition 3.5, to denote the case where the label is not further expanded.

These factors are finally used to compute the maximum and the minimum dIoU:

$$dIoU_{max} = \frac{\sum_{x \in \mathfrak{D}} |I_{max}(L)_x|}{\sum_{x \in \mathfrak{D}} |Union_{min}(L)_x|}$$
$$dIoU_{min} = \frac{\sum_{x \in \mathfrak{D}} |I_{min}(L)_x|}{\sum_{x \in \mathfrak{D}} |Union_{max}(L)_x|}$$
(15)

**Aggregated Computation** As in Section 3.2.1, the aggregated computation estimates the quantities more efficiently by operating on aggregate values (i.e., sums) per label instead of computing them on a per-sample basis. For example, rather than using $|Union^{sample}_{max}(L)_x| = |N_x| + \min(|E^C_{max}(L)_x|, Top_1(E^C)_x)$ the aggregated formulation uses $|Union^{aggr}_{max}(L)| = |N| + \min(\sum_{x \in \mathfrak{D}} |E^C_{max}(L)_x|, Top^A_1(E^C))$, where $Top^A$ is computed concept-wise.

### 3.3. Optimal Algorithm

By leveraging the quantities identified in the previous section, we propose a novel optimal algorithm based on a best-first search guided by our proposed heuristic. We provide a textual overview of the main steps below and a more detailed discussion and pseudocode in Section C.

1. Compute the exact quantities (Section 3.1) for every concept in the dataset.

2. For each concept, compute $dIoU_{max}$ and $dIoU_{min}$ for all possible paths starting from it, using the heuristic aggregated computation.

3. Initialize the frontier with all paths whose estimated $dIoU_{max}$ is greater than the global maximum of the minimum estimates.

4. Iteratively pop nodes from the frontier, starting from those with the highest estimated $dIoU_{max}$.

   (a) If the estimate is aggregated, refine it by computing the sample-based estimate and reinsert the node. Otherwise, proceed to the next step.

   (b) If the node path is not a final one (i.e., can still be extended), expand its label by concatenating every possible concept with every allowed connective. For each new label, compute $dIoU_{max}$

and $dIoU_{min}$ via the heuristic aggregated computation and insert them into the frontier.

(c) If the node path is a final one, compute its exact IoU. During this step, the algorithm stores the intermediate quantities of the sub-labels composing the label. This information is then backpropagated to the frontier: nodes that share sub-labels with the evaluated node update their estimates using these exact quantities.

(d) Continue until the frontier is empty. During the process, whenever a new maximum of the $dIoU_{min}$ estimates is found, prune the frontier by removing all nodes whose $dIoU_{max}$ falls below this threshold.

Additionally, to reduce redundant computation, the algorithm incorporates a limited set of logical equivalence rules, which are applied before and during the expansion phase, and maintains a buffer to cache recently explored nodes associated with the same estimated $dIoU_{max}$. Because the maximum $dIoU$ of each path is an overestimation, and since the algorithm is designed to visit all nodes whose $dIoU$ exceeds that of the current best explanation, **the algorithm is guaranteed to return the most aligned explanation**. In other words, the returned explanation is always the optimal one (proof in Section F).

## 4. Analysis

### 4.1. Feasibility of Optimality and Heuristic-Guided Beam Search

This section evaluates the feasibility of the proposed optimal algorithm and the effectiveness of the heuristic when used to guide beam search. We consider three scenarios that vary in annotation complexity: **low, moderate, and high**. The low-complexity setting (Cityscapes (Cordts et al., 2016)) includes a small number of concepts (25), all of which are disjoint. The moderate-complexity setting (the extended version of Ade20K (Zhou et al., 2017b) provided by Detectron2 (Wu et al., 2019)) includes a much larger number of concepts (847) but with no overlapping annotations. Finally, the high complexity setting (Broden (Bau et al., 2017)) involves frequent overlaps combined with a large number of concepts (1198). As reference baselines, we include the MMESH-guided beam search (La Rosa et al., 2023) and the vanilla beam search (Mu and Andreas, 2020). The beam search variant that uses our heuristic replaces MMESH with label-quantity estimation (see Section D for details). For all the beam variants, we fix the beam size to 5 as in (La Rosa et al., 2023). For each setting, we report the average number of visited nodes (i.e., those for which the exact IoU is computed), expanded nodes, estimated nodes, and the computation time per unit. Note that, for the optimal algorithm,

*Table 2.* Average number of visited, expanded, and estimated nodes, along with runtime per unit (in seconds), by the optimal algorithm, a beam search guided by our heuristic, and two alternative beam search algorithms.

| Algorithm | Visited | Exp. | Esti. | s/u |
|---|---|---|---|---|
| **Low Complexity** | | | | |
| Optimal (our) | 1 | 101 | 778 | 0.08 |
| Beam | | | | |
|    Our | 6 | 14 | 639 | 0.17 |
|    MMESH | 121 | 15 | 697 | 10.37 |
|    Vanilla | 716 | 15 | - | 2.77 |
| **Intermediate Complexity** | | | | |
| Optimal (our) | 1 | 4915 | $10^6$ | 90.57 |
| Beam | | | | |
|    Our | 10 | 15 | 37956 | 11.55 |
|    MMESH | 39 | 15 | 37956 | 38.42 |
|    Vanilla | 37979 | 15 | - | 450 |
| **High Complexity** | | | | |
| Optimal (our) | 47 | $10^5$ | $10^8$ | 5768 |
| Beam | | | | |
|    Our | 27 | 15 | 53752 | 123.33 |
|    MMESH | 43 | 15 | 53752 | 102.35 |
|    Vanilla | 53775 | 15 | - | 5929 |

we do not aim to improve runtime compared to beam-based algorithms, as this would likely be impossible given that the explored state space is significantly larger (by orders of magnitude) than that considered by beam search. Following the setup in Mu and Andreas (2020), we extract 50 random units from the last convolutional layer of a ResNet (He et al., 2016) model trained on Places365 (Zhou et al., 2017a) and use the highest activations (top 0.005 percentile) as the activation ranges. The reader can find additional results (standard deviation, two additional models, and different activation ranges) in Section B. We do not report the IoU in Table 2 because all the beam variants find the same solution, since all of them have access to the same restricted state space (i.e., the one reachable by beam search) and they use admissible heuristics, as proven in La Rosa et al. (2023) and Section F.1.

Table 2 shows that **the optimal algorithm consistently finds the optimal solution within feasible runtimes across all scenarios**. As expected, informed beam search methods are faster, since they explore a much smaller portion of the state space (i.e., estimated nodes). However, the running time of the optimal algorithm remains comparable to that of the vanilla beam search, even in the most complex settings. **This result represents a major milestone** in this area of research, since navigating the entire state space and finding

the optimal solution was considered infeasible in practice by previous literature (Mu and Andreas, 2020). As a practical point of comparison, breadth-first search would require $\sim 400$ million hours in intermediate and high-complexity settings[2], and $\sim 140$ seconds in the lowest-complexity settings, which is $\times 1000$ slower than the proposed optimal algorithm. Importantly, the number of expanded states is a small fraction of the overall state space (less than 0.1%) and is significantly smaller than the number of estimated states. This property is crucial for refining heuristic estimates without compromising runtime efficiency (see Section C).

More notably, beam search guided by our heuristic outperforms all baselines across all settings in terms of visited states while achieving comparable or better runtimes. Compared to MMESH, our approach offers several improvements. First, while MMESH uses spatial information, our heuristic relies only on binary information, making it applicable to a broader range of contexts than MMESH. Because of this reliance, MMESH requires computing updated information during beam selection and, therefore, keeping annotations in memory and relying on GPU resources. Conversely, our heuristic can approximate updated information similarly to the optimal algorithm, trading precision for efficiency while using annotations only when visiting states and allowing them to be loaded from disk on demand. This possibility, combined with its higher efficiency, may enable easier parallelization in low-resource scenarios. Our beam variant also scales efficiently with changes in hyperparameters. For example, when varying the explanation length (3, 5, 10, and 20) and beam size (5, 10, and 20) in moderate settings, the runtime of our variant remains stable between 0.19 and 0.42 minutes per unit. By contrast, MMESH slows down progressively with both explanation length and beam size, starting from 0.62 for 3-concept explanations and beam size fixed to 5 to 25 mins/unit (see Table 8 in Appendix) for 20-concept explanations.

Regarding memory consumption, the dominant factors are the neuron activation matrix and the concept tensor. Heuristic-related information accounts for only a small fraction of the memory. The impact of storing the frontier is negligible compared to these quantities. Additionally, the optimal algorithm does not need to store the beam (concept) tensors. As a result, the optimal algorithm is the lightest in terms of memory.

### 4.2. Explanation Analysis

This section addresses the question we originally posed about beam search–based algorithms: **are the explanations they compute optimal**? In general, the answer is **no**. Our analysis (Tables 3 and 4) shows that, in the High

---

[2]Computed under an optimistic assumption of 0.005 seconds per node

*Table 3.* Percentage of non-optimal explanations found by beam search and their distribution over different categories.

| Model | Non-Optimal | Cat 1 | Cat 2 | Cat 3 |
|---|---|---|---|---|
| ResNet | 9% | 76% | 6% | 17% |
| AlexNet | 23% | 93% | 5% | 2% |
| DenseNet | 39% | 73% | 0% | 27% |

*Table 4.* Avg. IoU per category over all the units in the layer for which the optimal and the beam search find two different solutions.

| Solution | Tot | Cat 1 | Cat 2 |
|---|---|---|---|
| | **ResNet** | | |
| Non-Optimal (Beam) | 0.073 | 0.073 | 0.065 |
| Optimal | 0.077 | 0.078 | 0.065 |
| | **AlexNet** | | |
| Non-Optimal (Beam) | 0.046 | 0.045 | 0.039 |
| Optimal | 0.049 | 0.048 | 0.040 |
| | **DenseNet** | | |
| Non-Optimal (Beam) | 0.039 | 0.036 | 0.056 |
| Optimal | 0.041 | 0.038 | 0.057 |

Complexity settings, between 10% and 40% of them differ from the optimal ones across several models studied in previous literature (Mu and Andreas, 2020): ResNet18 (He et al., 2016), AlexNet (Krizhevsky et al., 2012), and DenseNet161 (Huang et al., 2017). We classify these differences into three categories: (1) explanations differ in both concepts and IoU, (2) explanations involve the same concepts but differ in how they are connected, resulting in different IoU, and (3) explanations share the same IoU but differ in the way the concepts are connected.

The first category is the most prominent and often involves explanations connected by AND and AND NOT operators, suggesting that beam search struggles to express explanations that describe units specialized in recognizing complex scenarios and falls short of reflecting the highest degree of alignment that the unit actually exhibits.

The second and third categories include cases where the explanations share the same concepts but their semantics change. Specifically, in the second category, the optimal explanations are more precise and more aligned (e.g., from ((table OR sink) AND white-c) (IoU=0.036) to ((white-c AND table) OR sink) (IoU=0.040)). In the third case, the difference is subtle but highlights a pitfall of beam search algorithms. For example, consider the explanation ((ball_pit-s OR flower) AND NOT dining_room-s). At first glance, one might interpret this unit as recognizing $ball\_pit$ when it is specifically not located in dining rooms. However, inspecting the dataset reveals that these concepts never co-occur (i.e., they are disjoint), and therefore, a counterexample

for the specialization (where the neuron does not fire) does not exist. Conversely, there are instances including both flowers and dining rooms. Thus, part of the explanation is effectively "unverified". In contrast, the optimal algorithm correctly identifies the alignment as ((flower AND NOT dining_room-s) OR ball_pit-s) and exposes a key limitation of beam search: it cannot backtrack on earlier decisions, and the search may compensate for errors by producing explanations that rely on unverified scenarios (further details and examples for 10 units per model are provided in Section I).

### 4.3. Algorithm Analysis: Insights and Limitations

This section summarizes insights, design choices, and limitations of the optimal algorithm. The reader can read Sections C and H for a more extended discussion about design choices, generalizability, and further insights.

Overall, the beam search guided by our heuristic represents a safe choice when the reduced time for computation is a priority. Conversely, the optimal algorithm represents a promising first step towards guaranteeing optimality in compositional explanations and a better choice when optimality is considered more important than the time of execution. Note that the optimality property is independent of any dataset and model tested, as long as the assumptions are satisfied. In summary, all the quantities and steps proposed in this algorithm are necessary for the search to be tractable in high complexity scenarios. Among them, the use of aggregated computations to decide which nodes enter the frontier and sample-based ones to parse it is motivated by the fact that estimated nodes far outnumber those that are actually expanded, while backpropagation and minimum estimation are crucial to reduce redundant nodes and prevent exhaustive exploration of overestimated candidates. Finally, we identified the following insights and areas of improvement for future research:

**Time Complexity and Convergence Towards Breadth-first search** As discussed previously, the search space grows exponentially with the maximum explanation length $k$, but only polynomially with the number of concepts and logical operators for fixed $k$. In practice, the maximum explanation length is fixed by the user and it is small (typically 3-10), which makes the approach computationally tractable. We also noted that the state space is explored similarly to breadth-first search, since the top vector dominates the search and roughly overestimates the maximum improvement. In theory, this reliance could make it infeasible to provide long explanations (although shorter explanations are generally preferred). To mitigate this problem, future research could explore vectors that are (1) specific to a label (which is costly) or (2) novel and better representations of the maximum improvement.

**Unmeaningful Units** Our optimal algorithm can become significantly slower when either a unit is not interpretable (Bau et al., 2020) (i.e., the IoU $< 0.04$) or it is unspecialized (La Rosa et al., 2023) (converging towards a default rule). In these extreme cases, the space to be explored is very large since there is no clear alignment and the combinations of concepts are all similar. Because in these cases the optimality is not of interest, one possible solution is to run beam search and then refine the units deemed interpretable for optimality. Alternatively, one could automatically switch from the optimal algorithm to beam search once the frontier grows too large, and use the beam search output to initialize and guide the optimal search.

**Importance of Unique Elements** The optimal algorithm can become slower when applied to a dataset with an extremely low number (or none) of unique elements (e.g., the NLP case explored in (Mu and Andreas, 2020)). In this case, the algorithm relies only on estimates of the common elements, which are associated with larger ranges of uncertainty, thus resulting in closer estimations for multiple paths and an increase in the state space that must be explored. Future research could investigate mitigation strategies for this issue, such as dataset manipulation or conditioned paths.

## 5. Conclusion

This paper presents the first attempt to guarantee optimality in compositional explanations. Specifically, we identified and formalized the fundamental quantities governing spatial alignment, proposed a heuristic to estimate the potential alignment from any label in the search space, and developed an algorithm capable of computing optimal compositional explanations within feasible runtimes. We further demonstrated that our heuristic can also improve existing beam search–based approaches for non-optimal explanations. Since our method does not rely on spatial information, the proposed heuristic could be potentially applicable across domains. Moreover, our theoretical contribution may extend beyond neural explanations, as bitwise computations are of interest in areas such as semantic segmentation and communication. Finally, we call for further research on refining this heuristic and on designing new algorithms for computing alternative forms of compositional explanations.

## Impact Statement

This paper presents work whose goal is to advance the field of Machine Learning. There are many potential societal consequences of our work, none which we feel must be specifically highlighted here.

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

## A. Proof Equivalence Alignment-IoU Score

*Proof.* Let $\boldsymbol{M}_L$ be the binary label tensor representing the result of the bitwise logic operations induced by the logic operators connecting single concepts $k \in L$. The Intersection over Union (IoU) score is then defined as:

$$IoU(\boldsymbol{N}, L, \mathfrak{D}, \boldsymbol{M}_L) = \frac{|^1\boldsymbol{N} \cap {}^1\boldsymbol{M}_L|}{|^1\boldsymbol{N} \cup {}^1\boldsymbol{M}_L|} \tag{16}$$

We first observe that $|^1\boldsymbol{N} \cap {}^1\boldsymbol{M}_L|$ is necessarily equal to $\sum_{x \in \mathfrak{D}} |I^U(L)_x| + |I^C(L)_x|$. Indeed, all elements in $^1\boldsymbol{N} \cap {}^1\boldsymbol{M}_L$ are associated with both the neuron and the label $L$. Consider a generic element $j \in {}^1\boldsymbol{N} \cap {}^1\boldsymbol{M}_L$. Since the logic operators are 00-preserving (Assumption 3), this element must be associated with at least one annotation of one of the concepts in $L$. But if the element is associated with at least a concept, then it is either associated with exactly one, and thus $j \in I^U$, or with multiple concepts, and thus $j \in I^C$. This also holds in the case of concepts connected by the AND NOT operator, because by design, this operator must be chained to a positive concept, and AND NOT is 00-preserving.

Regarding the denominator, we can note that

$$|^1\boldsymbol{N} \cup {}^1\boldsymbol{M}_L| = |^1\boldsymbol{N}| + |^1\boldsymbol{M}_L| - |^1\boldsymbol{N} \cap {}^1\boldsymbol{M}_L| \tag{17}$$

Observe that for all $j \in {}^1\boldsymbol{N} \cap {}^1\boldsymbol{M}_L$, the contribution $|^1\boldsymbol{M}_L| - |^1\boldsymbol{N} \cap {}^1\boldsymbol{M}_L| = 0$, since these elements are counted in both sets. Hence, $|^1\boldsymbol{M}_L| - |^1\boldsymbol{N} \cap {}^1\boldsymbol{M}_L|$ represents the number of elements that are labeled as 1 in $\boldsymbol{M}_L$ but for which the neuron does not fire. This definition coincides with Definition 3.4 Therefore, we can rewrite the denominator as

$$|^1\boldsymbol{N} \cup {}^1\boldsymbol{M}_L| = |^1\boldsymbol{N}| + \sum_{x \in \mathfrak{D}} |E(L)_x|. \tag{18}$$

Similarly to the numerator case, because the logic operators are 00-preserving, every element in $E(L)_x$ is either included in $E^U(L)_x$ or in $E^C(L)_x$. Thus,

$$|^1\boldsymbol{N} \cup {}^1\boldsymbol{M}_L| = |^1\boldsymbol{N}| + \sum_{x \in \mathfrak{D}} \left( |E^U(L)_x| + |E^C(L)_x| \right). \tag{19}$$

and thus

$$dIoU(\boldsymbol{N}, L, \mathfrak{D}, I^C, I^U, E^U, E^C) = IoU(\boldsymbol{N}, L, \mathfrak{D}, \mathbf{M}). \tag{20}$$

$\square$

## B. Complete Results

Tables 5 to 7 report the complete results (including averages and standard deviations) for three probed models: ResNet18, AlexNet, and DenseNet161. The statistics are computed over 50 units randomly extracted from the penultimate layer of each probed model trained on Places365, consistent with prior work on compositional explanations (Mu and Andreas, 2020; Harth, 2022; Makinwa et al., 2022; La Rosa et al., 2023). We follow Bau et al. (2020) and Mu and Andreas (2020) and use the highest activations corresponding to the top 0.005 percentile across the probing dataset as the activation range, fix the maximum explanation length to 3, and the beam size to 5. As probing datasets, we used Cityscapes (Cordts et al., 2016) (accessible at: https://www.cityscapes-dataset.com/ under MIT License) for the low complexity settings, Ade20kFull (Zhou et al., 2017b) (accessible via the Detectron2 (Wu et al., 2019) framework) for the moderate settings, and Broden (Bau et al., 2017) (accessible at https://github.com/CSAILVision/NetDissect under MIT license) for the highest complexity settings.

We also tested and confirmed our findings on a different activation range than the 0.005 quantile used in the main text. Specifically, in Table 9 we consider, as the activation range, cluster 5 identified following the La Rosa et al. (2023) protocol. Note that this cluster includes the highest activations but considers a broader activation range. We apply the optimal algorithm to 50 randomly selected units in a DenseNet model and compare its explanations against those obtained by beam search. Similarly to the threshold used in the main text, more than 30% of the explanations found by beam search in this setting are not optimal. We observe a higher number of non-optimal explanations falling into Category 2, suggesting that

*Table 5.* Average number of visited, expanded, and estimated nodes, along with runtime per unit (in seconds), by the optimal algorithm, a beam search guided by our heuristic, and two alternative beam search algorithms. The statistics are computed over 50 units of a trained ResNet18 model.

| Algorithm | Optimal | Visited | Expanded | Estimated | Time (sec/unit) |
|---|---|---|---|---|---|
| **Low Complexity** | | | | | |
| Optimal (our) | ✓ | 1 | $100.55 \pm 33.61$ | $778.14 \pm 220.72$ | $0.08 \pm 0.01$ |
| Beam + Our H. | ✗ | $5.88 \pm 1.04$ | $13.88 \pm 0.52$ | $638.92 \pm 21.52$ | $0.17 \pm 0.01$ |
| MMESH Beam | ✗ | $121.22 \pm 31.15$ | $14.94 \pm 0.42$ | $697.10 \pm 17$ | $10.37 \pm 0.62$ |
| Vanilla Beam | ✗ | $716.22 \pm 17.42$ | $14.94 \pm 0.42$ | - | $2.77 \pm 0.16$ |
| **Intermediate Complexity** | | | | | |
| Optimal (our) | ✓ | 1 | $4914.94 \pm 2010.77$ | $2.69 \times 10^6$ | $90.57 \pm 35.49$ |
| Beam + Our H. | ✗ | 10.00 | 15 | $37956.24 \pm 1.78$ | $11.55 \pm 1.41$ |
| MMESH Beam | ✗ | $38.92 \pm 155.39$ | 15 | $37956.31 \pm 1.79$ | $38.42 \pm 2.65$ |
| Vanilla Beam | ✗ | $37979.18 \pm 1.27$ | 15 | - | $449.94 \pm 10.37$ |
| **High Complexity** | | | | | |
| Optimal (our) | ✓ | $47.36 \pm 99.27$ | $3.54 \times 10^5$ | $1.28 \times 10^8$ | $5768 \pm 1297$ |
| Beam + Our H. | ✗ | $27.02 \pm 40.46$ | 15 | $53752.20 \pm 2.10$ | $123.33 \pm 10.83$ |
| MMESH Beam | ✗ | $43.49 \pm 68.13$ | 15 | $53752.20 \pm 2.10$ | $102.35 \pm 16.28$ |
| Vanilla Beam | ✗ | $53775 \pm 1$ | 15 | - | $5929 \pm 548$ |

the broader the activation range, the more difficult it becomes for beam search to find the most precise and well-aligned explanations given a set of concepts.

All results were computed on a workstation equipped with an NVIDIA RTX 3090 GPU, without parallelization, to avoid timing overhead.

## C. Optimal Algorithm

This section describes the optimal algorithm we introduced in the main paper. Algorithm 1 shows the pseudocode of our algorithm. The optimal algorithm is a best-first search guided by our proposed heuristic and consists of the following steps:

1. Compute the exact quantities (Section 3.1) for every concept in the dataset (line 6).

2. For each concept, compute $dIoU_{max}$ and $dIoU_{min}$ for all possible paths starting from it, using the heuristic aggregated computation (line 7).

3. Initialize the frontier with all paths (line 8) and keep track of the greatest $dIoU_{min}$ (line 9). The frontier is sorted by the estimated $dIoU_{max}$.

4. Reduce the frontier by removing nodes whose estimated $dIoU_{max}$ is lower than the global maximum of the minimum estimates (line 11).

5. Iteratively pop nodes from the frontier, starting from those with the highest estimated $dIoU$ (lines 13).

   (a) If the estimate is aggregated, refine it by computing the sample-based estimate and reinsert the node (lines 14-21). Otherwise, proceed to the next step.

   (b) Apply logical equivalence rules when possible (lines 22-27), recompute the sample-based estimate for the equivalent expression, and reinsert the node if the estimate has changed. Otherwise, proceed to the next step. Currently, we only check distributive properties as logical equivalences. Note that, although the expressions are logically equivalent, their estimates may differ due to overestimation. The goal of this step is to select the form with the smallest overestimation among all possible equivalents.

*Table 6.* Average number of visited, expanded, and estimated nodes, along with runtime per unit (in seconds), by the optimal algorithm, a beam search guided by our heuristic, and two alternative beam search algorithms. The statistics are computed over 50 units of a trained AlexNet model.

| Algorithm | Visited | Exp. | Esti. | s/u |
|---|---|---|---|---|
| **Low Complexity** | | | | |
| Optimal (our) | 1 | $80 \pm 44$ | $646 \pm 285$ | $2.39 \pm 0.20$ |
| Beam | | | | |
|    Our | $6 \pm 1$ | $13.48 \pm 1.15$ | $623 \pm 46.81$ | $0.17 \pm 0.01$ |
|    MMESH | $120 \pm 34$ | $14.56 \pm 1.10$ | $681 \pm 43.17$ | $10.39 \pm 0.62$ |
|    Vanilla | $681 \pm 42$ | $13.56 \pm 1.10$ | - | $2.90 \pm 0.40$ |
| **Intermediate Complexity** | | | | |
| Optimal (our) | 1 | $3576 \pm 2339$ | $10^6$ | $378.56 \pm 98.60$ |
| Beam | | | | |
|    Our | $10 \pm 1$ | 15 | $37697 \pm 767$ | $12.41 \pm 1.51$ |
|    MMESH | $190 \pm 398$ | 15 | $37979 \pm 2$ | $41.95 \pm 8.00$ |
|    Vanilla | $37956 \pm 3$ | 15 | - | $485.59 \pm 41.79$ |
| **High Complexity** | | | | |
| Optimal (our) | $233 \pm 333$ | $10^5$ | $10^8$ | $6949.16 \pm 2465.46$ |
| Beam | | | | |
|    Our | $60 \pm 48$ | 15 | $53751 \pm 4$ | $149.16 \pm 32.67$ |
|    MMESH | $101 \pm 67$ | 15 | $53751 \pm 4$ | $158.50 \pm 42.38$ |
|    Vanilla | $53751 \pm 4$ | 15 | - | $6672.82 \pm 727$ |

(c) Check whether the same node has been recently explored by comparing its maximum IoU with the most recent one. If they match but the node has not yet been explored, add it to memory; if it has already been explored, skip it. Otherwise, clear the memory and initialize the most recent IoU with the node's maximum IoU (lines 28-37).

(d) If the node is not a final one (i.e., can still be extended), expand its label by concatenating every possible concept with every allowed connective (line 47). For each new label, compute $dIoU_{max}$ and $dIoU_{min}$ via the heuristic aggregated computation and insert them into the frontier (lines 48-49). If a new maximum is found among $dIoU_{min}$ of the new paths, update the global maximum and reduce the frontier (lines 50-53).

(e) If the node is a final one, compute its exact IoU (line 41). During this step, the algorithm stores the intermediate quantities of the sub-labels composing the label (line 39). This information is then backpropagated to the frontier: nodes that share sub-labels with the evaluated node update their estimates using these exact quantities (line 40). If the IoU is greater than that of the best label found so far, update the best label with the current node (lines 42-44).

(f) Continue until the frontier is empty (line 11).

In the following, we provide the rationale behind several design choices made during the development of this algorithm.

**Memory Mechanism** The memory mechanism (lines 28-37) was introduced to account for logical equivalences not handled elsewhere in the algorithm and to prevent redundant computation. We considered several alternatives, but this solution proved to be the least computationally expensive. Without such a mechanism, some logically equivalent rules could be expanded multiple times (lines 47-54), leading to a significantly larger search space. Alternative options would be to check for logical equivalences or the presence of a node directly when adding nodes to the frontier. However, this would make exploration prohibitively expensive: verifying logical equivalence or membership would offset the efficiency gains of the heuristic. In addition, since the frontier is implemented as a heap, checking whether a node is already present is slower than with a sorted list. Maintaining a sorted frontier would require re-sorting on every insertion, which would be too costly and therefore infeasible.

*Table 7.* Average number of visited, expanded, and estimated nodes, along with runtime per unit (in seconds), by the optimal algorithm, a beam search guided by our heuristic, and two alternative beam search algorithms. The statistics are computed over 50 units of a trained DenseNet161 model.

| Algorithm | Visited | Exp. | Esti. | s/u |
|---|---|---|---|---|
| **Low Complexity** | | | | |
| Optimal (our) | 1 | $74 \pm 41$ | $616 \pm 290$ | $0.08 \pm 0.02$ |
| Beam | | | | |
|     Our | $5 \pm 1$ | $13.40 \pm 1.28$ | $622 \pm 47$ | $0.18 \pm 0.01$ |
|     MMESH | $127 \pm 32$ | $14.46 \pm 1.22$ | $680 \pm 44$ | $15.51 \pm 0.59$ |
|     Vanilla | $680 \pm 44$ | $13.46 \pm 1.22$ | - | $3.06 \pm 0.20$ |
| **Intermediate Complexity** | | | | |
| Optimal (our) | 1 | $4903 \pm 2716$ | $10^6$ | $97.03 \pm 44.68$ |
| Beam | | | | |
|     Our | 10 | 15 | $37801 \pm 607$ | $12.17 \pm 0.93$ |
|     MMESH | $218 \pm 946$ | 15 | $37957 \pm 3$ | $60.10 \pm 18.46$ |
|     Vanilla | $37957 \pm 3$ | 15 | - | $472.51 \pm 33.96$ |
| **High Complexity** | | | | |
| Optimal (our) | $241 \pm 289$ | $10^5$ | $10^8$ | $7698.68 \pm 2200$ |
| Beam | | | | |
|     Our | $64 \pm 47$ | 15 | $53751 \pm 4$ | $144.46 \pm 34.05$ |
|     MMESH | $129 \pm 196$ | 15 | $53752 \pm 4$ | $135.25 \pm 42.94$ |
|     Vanilla | $53751 \pm 4$ | 15 | - | $6556.32 \pm 697.77$ |

**Concept Quantities**   A key design choice concerns the handling of concept quantities. We store quantities only for atomic concepts and do not cache them when estimating (lines 7, 15, and 48) or computing the $dIoU$ (line 41). Consequently, the intermediate quantities of labels are recomputed multiple times during the search. This decision was made for two main reasons. First, storing additional quantities increases access time, which is critical because the algorithm frequently accesses these values and this overhead could easily exceed the time required to recompute them from scratch. Second, caching more quantities increases memory usage, potentially limiting the ability to run parallel processes. Given the substantial runtime required for the most complex datasets, preserving the ability to parallelize is essential. Moreover, avoiding extensive caching keeps the approach lightweight in terms of memory and resource requirements. In conclusion, the marginal gains from more precise estimations are outweighed by the overhead, especially given that the algorithm tends to converge toward a breadth-first search (Section 4.3).

**Backpropagation**   This process was introduced to reduce the number of visited states. While it has no (or bad) impact on low and moderate-complexity settings, it significantly reduces runtime in high complexity scenarios, especially when running on CPUs. Specifically, the frontier in the later stages of the search often contains similar explanations that differ only in the last added concept (e.g., ((A AND B) OR C) and ((A AND B) OR D)). These nodes typically appear in the frontier because the left-hand terms provide a rough overestimation relative to the current maximum. When a node is visited, this overestimation is temporarily corrected, and the backpropagation step updates the estimates of all remaining nodes. This correction helps the algorithm to avoid visiting unpromising states. On average, 2000-3000 nodes per unit are updated via backpropagation, highlighting the importance of this mechanism.

**Sample vs Aggregated Computation**   The optimal algorithm leverages both sample-based and aggregated computations for path and label estimations. This design is necessary due to the large state space and the overestimation introduced by the aggregated computation. We explored several alternatives during the development of this work, but this trade-off is the only one that ensures feasibility in high complexity settings. As explained in the main text, sample computation requires $|\mathfrak{D}|$ comparisons per calculation, whereas aggregated computation relies solely on the sum of concept quantities computed at the first step of the algorithm, combining them in a single operation. These two approaches represent a trade-off between

*Table 8.* Avg. Time across 50 units (min) per hyperparameters on moderate settings.

| Value | Our | MMESH |
|-------|------|--------|
| | **Explanation Len** | |
| 3 | 0.19 | 0.62 |
| 5 | 0.28 | 1.28 |
| 10 | 0.28 | 3.43 |
| 20 | 0.42 | 26.63 |
| | **Beam Size** | |
| 5 | 0.17 | 0.62 |
| 10 | 0.17 | 1.24 |
| 20 | 0.20 | 2.84 |

*Table 9.* Percentage of non-optimal explanations found by beam search and their distribution over different categories considering Cluster 5, as identified by La Rosa et al. (2023), as an activation range .

| Model | Non-Optimal | Cat 1 | Cat 2 | Cat 3 |
|-------|-------------|-------|-------|-------|
| DenseNet | 33% | 68% | 25% | 6% |

precision and efficiency: the sample version is more accurate but computationally intensive, while the aggregated version is faster but less precise. In practice, using only aggregated computation for both node expansion and frontier exploration would yield faster per-node computations but result in a frontier that is orders of magnitude larger than that explored by our combined approach. Conversely, using only sample computation slightly reduces the frontier (by roughly 50,000 nodes per unit in preliminary experiments) but incurs significantly higher computation time per node, effectively nullifying the gains from the smaller frontier.

**MinIoU** As explained previously, the algorithm computes both the minimum and maximum IoU for each label and path. This increases the time required per estimation, since the maximum and minimum calculations are distinct and rarely share terms, effectively doubling the computation time per node. An alternative design could omit the MinIoU computation and explore only nodes whose maximum IoU exceeds the best visited so far. However, without the MinIoU, it becomes impossible to dynamically reduce the frontier at runtime: the frontier can only grow, and the only way to remove a single node is to fully explore it. This would result in a frontier orders of magnitude larger than the one explored by the proposed design, especially in the early phase of the search, when the MinIoU is updated multiple times and allows significant pruning. Future work could explore adaptive strategies that decide dynamically whether to compute MinIoU based on the search space or external information, potentially improving overall runtime.

**Code Optimization** In addition to the previously mentioned design choices, we implemented several code optimizations to avoid unnecessary computation of quantities and to handle logical equivalences. For instance, we enforce an order on concept indices when chaining two consecutive applications of the same operator during node expansion. For example, for a concept with index 10, it can only be chained with concepts having an index greater than 10, since all other combinations will be captured when expanding nodes with lower indices. The same rule applies to consecutive operators of the same type. For example, if we have (3 OR 15), we can only chain concepts with indices greater than 15 when applying the OR operator again (e.g., (3 OR 15) OR 18), while we are free to choose any concept for other operators (AND or AND NOT). Other code optimizations involve avoiding the computations of paths when the intersection of the right or left sides is 0 and avoiding the sample computation related to equations involving $Bott_1$ in datasets where this vector is always 0 (see Section E)

## D. Beam Search Algorithm

In this section, we provide a high-level description of the beam search guided by our heuristic and present its pseudo-code in Algorithm 2. This algorithm replaces the MMESH heuristic (La Rosa et al., 2023) with the label quantity estimates

introduced in Section 3.1, within a standard heuristic-guided beam search framework. The procedure begins by following the initial two steps of the optimal algorithm but keeps only the best $b$ concepts to form the initial beam. It then proceeds iteratively for $n$ steps: at each iteration, the algorithm expands the nodes in the current beam, sorts the resulting state space according to the heuristic, and evaluates candidate nodes by computing their $IoU$. The top $b$ nodes are then selected to form the next beam. The process terminates when no candidate improves the current best $IoU^{max}$, or when no nodes remain to be expanded or visited. Differently from the optimal algorithm, this algorithm relies solely on sample computations and does not make use of, or estimate, the paths introduced in Section 3.2.2. Beyond the advantages discussed in Section 4.1, our heuristic does not rely on spatial information and can therefore be applied across multiple domains without requiring modifications to the code or formulation.

## E. Estimating Quantities

This section presents the rationale and derivation of all estimations computed by our heuristic. We divide the discussion into per-sample estimations and aggregated computations.

### E.1. Sample Computation

#### E.1.1. LABEL QUANTITIES

In the following, we derive the estimations of the label quantities introduced in the main text for the sample-based computation. For clarity, we group the equations by operator and discuss each operator separately. Because the quantities for the unique elements are exact and are directly derived from the definition and Observation 1, here we focus on the derivation of the common quantities.

$$|I_{min}^C(L)_x| = max(|I_{min}^C(L_\leftarrow)_x|, |I^C(L_\rightarrow)_x|) \tag{21}$$

$$|I_{max}^C(L)_x| = min(|I_{max}^C(L_\leftarrow)_x| + |I^C(L_\rightarrow)_x|, |N_x^C|) \tag{22}$$

$$|E_{min}^C(L)_x| = max(|E_{min}^C(L_\leftarrow)_x|, |E^C(L_\rightarrow)_x|) \tag{23}$$

$$|E_{max}^C(L)_x| = min(|E_{max}^C(L_\leftarrow)_x| + |E^C(L_\rightarrow)_x|, |SE_x^C|) \tag{24}$$

*Derivation Equations* (21) *to* (24) *for the OR operator*: We can start by noting that Equation (21) corresponds to the case where one side is a subset of the other. In this case, the equation gives the maximum cardinality of the intersection between the left and right sides, since the minimum elements are already included in the maximum and the OR is 1-preserving (i.e., the number of ones cannot be lower than before the combination). Conversely, Equation (22) corresponds to the case where the sides are disjoint in their extras, adjusted by the $|N_x^C|$ quantity. Indeed, because the left side is an overestimation, the sum may exceed the limits, requiring readjustment using $|N_x^C|$. The estimation of the maximum and minimum extras follows the same reasoning, with the only difference that the common space extra $|SE_x^C|$ is used to adjust the quantity of the maximum extras.

$$|I_{min}^C(L)_x| = max(|I_{min}^C(L_\leftarrow)_x| + |I^C(L_\rightarrow)_x| - |N_x^C|, 0) \tag{25}$$

$$|I_{max}^C(L)_x| = min(|I_{max}^C(L_\leftarrow)_x|, |I^C(L_\rightarrow)_x|) \tag{26}$$

$$|E_{min}^C(L)_x| = max(|E_{min}^C(L_\leftarrow)_x| + |E^C(L_\rightarrow)_x| - |SE_x^C|, 0) \tag{27}$$

$$|E_{max}^C(L)_x| = min(|E_{max}^C(L_\leftarrow)_x|, |E^C(L_\rightarrow)_x|) \tag{28}$$

*Derivation Equations* (25) *to* (28) *for the AND operator*: In this case, the AND operator is 0-preserving. Therefore, when computing the minimum, we consider the scenario where a guaranteed overlap (i.e., both sides equal to 1) occurs. This happens when the sum of the two sides exceeds the maximum available space, represented by $|N_x^C|$ and $|SE_x^C|$, respectively. In such cases, the minimum guaranteed overlap is given by the difference between the sum and the cardinality of the available space. In all other cases, the best estimation we can provide is simply 0. The computation of the maximum corresponds to the case of fully overlapping concepts. Since the operator is 0-preserving, the equation in this case selects the minimum of the two sides for each sample.

$$|I^C_{min}(L)_x| = max(|I^C_{min}(L_\leftarrow)_x| - |I^C(L_\rightarrow)_x|, 0) \tag{29}$$

$$|I^C_{max}(L)_x| = min(|I^C_{max}(L_\leftarrow)_x)|, |N^C_x| - |I^C(L_\rightarrow)_x|) \tag{30}$$

$$|E^C_{min}(L)_x| = max(|E^C_{min}(L_\leftarrow)_x| - |E^C(L_\rightarrow)_x|, 0) \tag{31}$$

$$|E^C_{max}(L)_x| = min(|E^C_{max}(L_\leftarrow)_x|, |SE^C_x| - |E^C(L_\rightarrow)_x|) \tag{32}$$

*Derivation Equations* (29) *to* (32) *for the AND NOT operator*: This operator behaves like the AND operator but combines a negated concept, flipping all the bits in the corresponding vectors. The maximum estimations follow the same reasoning as for the AND operator, except that in this case $|N^C_x| - |I^C(L_\rightarrow)_x|$ and $|SE^C_x| - |E^C(L_\rightarrow)_x|$ represent the bits that were originally 0 and are now 1, corresponding to the common intersection and the common extras of the negated concept. The minimum estimation corresponds to the case where the right side fully overlaps with the left side. In this case, due to the negation, all overlapping bits flip to 0. Since the AND operator is 0-preserving, this results in the loss of all left side elements shared with the right side.

### E.1.2. PATH QUANTITIES

In the following, we derive the estimations of the path quantities introduced in the main text. For clarity, we group the equations by operator and discuss each operator separately. Note that we assume, for all the paths, that at least 1 concept is added to the label, since the case where no concepts are added is covered by the "final path" obtained by applying Definition 3.5 to the label quantities. In all the following equations, $t$ denotes the difference between the maximum length and the length of the label $L$.

$$|I_{min}(L)_x| = max(|I^C_{min}(L)_x| + |I^U(L)_x|, Bott_1(I^C)_x + Bott_1(I^U)_x) \tag{33}$$

$$|I_{max}(L)_x| = min(|I^C_{max}(L)_x| + Top_t(I^C)_x, |N^C_x|) + min(|I^U(L)_x| + Top_t(I^U)_x, |N^U_x|) \tag{34}$$

$$|Union_{min}(L)_x| = |^1N_x| + max(|E^C_{min}(L)_x| + |E^U(L)_x|, Bott_1(E^C)_x + Bott_1(E^U)_x) \tag{35}$$

$$\begin{aligned} |Union_{max}(L)_x| = &|^1N_x| + min(|E^C_{max}(L)_x| + Top_t(E^C)_x, |SE^C_x|) \\ &+ min(|E^U(L)_x| + Top_t(E^U)_x, |SE^U_x|) \end{aligned} \tag{36}$$

*Derivation of Equations* (33) *to* (36) *for the OR operator*: Equation (33) and Equation (35) follow the same derivation as Equations (21) and (23). In this case, the added concept is represented by $Bott_1(I^C)$, which corresponds to the minimum intersection of any concept in a given sample and reflects the case of fully overlapping concepts. In practice, for most datasets, this quantity is always equal to 0; thus, it reduces to $|I^C_{min}(L)_x| + |I^U(L)_x|$. The same reasoning applies to $|Union_{min}(L)_x|$ and the extras. Note also that the label quantities can be lower than the bottom values, since they represent combinations of concepts that may further reduce these quantities. Finally, the maximum quantities correspond to the case where the concepts included in $L$ are disjoint from those cumulated in the $Top$ vectors. In this situation, the quantities are simply the sum of neuron activations, maximum quantities, and the $Top$ vectors, adjusted by the available space $|N^U_x|$ and $|SE^U_x|$, respectively.

$$|I_{min}(L)_x| = 0 \tag{37}$$

$$|I_{max}(L)_x| = min(|I^C_{max}(L)_x|, Top_1(I^C)_x) \tag{38}$$

$$|Union_{min}(L)_x| = |^1N_x| \tag{39}$$

$$|Union_{max}(L)_x| = |^1N_x| + min(|E^C_{max}(L)_x|, Top_1(E^C)_x) \tag{40}$$

*Derivation Equations* (37) *to* (40) *for the AND operator*: This operator is simpler to derive. Specifically, Equations (25) and (27) corresponds to the case where the label and all the concepts included in the $Top$ vectors are disjoint, and thus all the quantities reduce to 0. Conversely, Equation (39) and Equation (40) represent the case where the label fully overlaps with the quantities stored in $Top_1$. Therefore, for the AND operator, only the quantities from the smaller concept are preserved per sample. Note that $Top_1$ is the only applicable $Top$ vector here, since higher indices sum the cardinality of multiple concepts, which would violate the operations defined by the AND operator.

$$|I_{min}(L)_x| = |I^U(L)_x| \tag{41}$$

$$|I_{max}(L)_x| = |I^U(L)_x| + min(|I^C_{max}(L)_x|, |N^C_x| - Bott_1(I^C)_x) \tag{42}$$

$$|Union_{min}(L)_x| = |^1N_x| + |E^U(L)_x| \tag{43}$$

$$|Union_{max}(L)_x| = |^1N_x| + |E^U(L)_x| + min(|E^C_{max}(L)_x|, |SE^C_x| - Bott_1(E^C)_x), \tag{44}$$

*Derivation Equations* (41) *to* (44) *for the AND NOT operator*: Equations (29) and (31) corresponds to the same case as in Equations (25) and (27). However, since the AND NOT operator preserves all unique elements (Observation 1), the minimum intersection corresponds to the unique intersection of the label, and the minimum union includes the unique extras. Conversely, Equations (43) and (44) represents the case where the label fully overlaps with the negated concept, represented by $|N^C_x| - Bott_1(I^C)_x$ and $|SE^C_x| - Bott_1(E^C)_x$, respectively. As previously noted, in practice, for most datasets, $Bott_1$ is always equal to 0. Thus, the equations simplify to $|I^U(L)_x| + |I^C_{max}(L)_x|$ for the intersection and $|^1N_x| + |E^C_{max}(L)_x| + |E^U(L)_x|$ for the union, since by definition $|I^C_{max}(L)_x| < |N^C_x|$ and $|E^U(L)_x| < |SE^C_x|$ due to the fact that $|N^C_x|$ represents the total neuron common space and $|SE^C_x|$ represents the total available space for common extras.

## E.2. Aggregated Computation

This section describes the aggregated computation of the common quantities. To improve readability, we shorten the notation $\sum_{x \in \mathfrak{D}}$ to simply $\sum$, since there are no ambiguities and the summation is used exclusively for this purpose.

### E.2.1. LABEL QUANTITIES

In the case of disjoint concepts, we can use the same equation as in the sample-based case described in Section 3.2.1, since these are already aggregated. For the common quantities, the modification consists of pre-computing the aggregate value per label rather than per sample. This requires computing the dataset-wide sum only for atomic concepts, while all higher-arity labels can be derived through arithmetic operations over these sums. Therefore:

$$|I^C_{min}(L)_x| = \begin{cases} min(\sum |I^C_{min}(L_\leftarrow)_x|, \sum |I^C(L_\rightarrow)_x|) & \text{if } \oplus = \vee \\ max(\sum |I^C_{min}(L_\leftarrow)_x| + \sum |I^C(L_\rightarrow)_x| - |N^C|, 0) & \text{if } \oplus = \wedge \\ max(\sum |I^C_{min}(L_\leftarrow)_x| - \sum |I^C(L_\rightarrow)_x|, 0) & \text{if } \oplus = \neg\wedge \end{cases} \tag{45}$$

$$|I^C_{max}(L)_x| = \begin{cases} min(\sum |I^C_{max}(L_\leftarrow)_x| + \sum |I^C(L_\rightarrow)_x|, |N^C|) & \text{if } \oplus = \vee \\ min(\sum |I^C_{max}(L_\leftarrow)_x|, \sum |I^C(L_\rightarrow)_x|) & \text{if } \oplus = \wedge \\ min(\sum |I^C_{max}(L_\leftarrow)_x)|, |N^C| - \sum |I^C(L_\rightarrow)_x|) & \text{if } \oplus = \neg\wedge \end{cases} \tag{46}$$

$$|E^C_{min}(L)_x| = \begin{cases} max(\sum |E^C_{min}(L_\leftarrow)_x|, \sum |E^C(L_\rightarrow)_x|) & \text{if } \oplus = \vee \\ max(\sum |E^C(L_\leftarrow)_x| + \sum |E^C(L_\rightarrow)_x| - |SE^C|, 0) & \text{if } \oplus = \wedge \\ max(\sum |E^C_{min}(L_\leftarrow)_x| - \sum |E^C(L_\rightarrow)_x|, 0) & \text{if } \oplus = \neg\wedge \end{cases} \tag{47}$$

$$|E^C_{max}(L)_x| = \begin{cases} min(\sum |E^C_{max}(L_\leftarrow)_x| + \sum |E^C(L_\rightarrow)_x|, |SE^C|) & \text{if } \oplus = \vee \\ min(\sum |E^C_{max}(L_\leftarrow)_x|, \sum |E^C(L_\rightarrow)_x|) & \text{if } \oplus = \wedge \\ min(\sum |E^C_{max}(L_\leftarrow)_x|, |SE^C| - \sum |E^C(L_\rightarrow)_x|) & \text{if } \oplus = \neg\wedge \end{cases} \tag{48}$$

The derivation of these quantities follows the same rationale as the sample computation, and the only difference is the larger overestimation produced by the aggregation of label-wise quantities.

### E.2.2. PATH QUANTITIES

Similarly to the sample-based computation, computing the path $dIoU$ requires estimating the maximum possible improvement. In this case, however, the $Top^A$ and $Bott^A$ vectors are computed concept-wise rather than per sample. Specifically, for each quantity, we sort the values of all individual concepts in the dataset and compute cumulative sums into the $Top^A$ and $Bott^A$ vectors, starting from the largest and smallest values, respectively. Substituting these into the equations of the sample-based computation, we obtain:

$$I_{min}(L) = \begin{cases} max(\sum |I^C_{min}(L)_x| + \sum |I^U(L)_x|, Bott^A_1(I^C) + Bott^A_1(I^U)) & \vee \text{ Path} \\ 0 & \wedge \text{ Path} \\ \sum |I^U(L)_x| & \wedge \neg \text{ Path} \end{cases} \quad (49)$$

$$I_{max}(L) = \begin{cases} min(\sum |I^C_{max}(L)_x| + Top^A_t(I^C), |N^C|) + \\ \quad min(\sum |I^U(L)_x| + Top^A_t(I^U), |N^U|) & \vee \text{ Path} \\ min(\sum |I^C_{max}(L)_x|, Top^A_1(I^C)) & \wedge \text{ Path} \\ \sum |I^U(L)_x| + min(\sum |I^C_{max}(L)_x|, |N^C| - Bott^A_1(I^C)) & \wedge \neg \text{ Path} \end{cases} \quad (50)$$

$$|Union_{min}(L)| = |^1\boldsymbol{N}| + \begin{cases} max(\sum |E^C_{min}(L)_x| + \sum |E^U(L)_x|, Bott^A_1(E^C) + Bott^A_1(E^U)) & \vee \text{ Path} \\ max(\sum |E^C_{min}(L)_x| + Bott^A_1(E^C) - |SE^C|, 0) & \wedge \text{ Path} \\ \sum |E^C_{min}(L)_x| & \wedge \neg \text{ Path} \end{cases} \quad (51)$$

$$|Union_{max}(L)| = |^1\boldsymbol{N}| + \begin{cases} min(\sum |E^C_{max}(L)_x| + Top^A_t(E^C), |SE^C|) \\ \quad + min(\sum |E^U(L)_x| + Top^A_t(E^U), |SE^U|) & \vee \text{ Path} \\ min(\sum |E^C_{max}(L)_x|, Top^A_1(E^C)) & \wedge \text{ Path} \\ \sum |E^C_{max}(L)_x| + min(\sum |E^U(L)_x|, |SE^C| - Bott^A_1(E^C)) & \wedge \neg \text{ Path} \end{cases} \quad (52)$$

The derivation of these quantities follows the same rationale as the sample computation case, and the only difference is that these represent a larger overestimation. However, note in this case $|Union_{min}(L)|$ can be greater than 0, unlike in the sample computation where $\forall x \in \mathfrak{D}$, $Bott_1(E^C)_x = 0$, except in the rare degenerate case (not observed in any of the datasets tested in this paper) where a single sample contains all concepts in the dataset.

### E.3. Path Combinations

As in the previous section, we shorten here the notation $\sum_{x \in \mathfrak{D}}$ to simply $\sum$. As mentioned in the main text, to estimate values for paths involving multiple operators, we select the maximum and minimum of each quantity across the operators considered. For example, when both OR and AND can appear along a path, we compute:

$$dIoU_{max}(OR, AND)_x = \frac{max(\sum |I^{OR}_{max}(L)_x|, \sum |I^{AND}_{max}(L)_x|)}{min(\sum |Union^{OR}_{min}(L)_x|, \sum |Union^{AND}_{min}(L)_x|)} \quad (53)$$

and

$$dIoU_{min}(OR, AND)_x = \frac{min(\sum |I^{OR}_{min}(L)_x|, \sum |I^{AND}_{min}(L)_x|)}{max(\sum |Union^{OR}_{max}(L)_x|, \sum |Union^{AND}_{max}(L)_x|)} \quad (54)$$

These expressions simplify to:

$$dIoU_{max}(OR, AND)_x = \frac{\sum |I^{OR}_{max}(L)_x|}{\sum |Union^{AND}_{min}(L)_x|} \quad (55)$$

and

$$dIoU_{min}(OR, AND)_x = \frac{\sum |I^{AND}_{min}(L)_x|}{\sum |Union^{AND}_{max}(L)_x|} \quad (56)$$

We can further rewrite them as:

$$dIoU_{max}(OR, AND)_x = \\ \frac{min(|I^C_{max}(L)_x| + Top_t(I^C)_x, |N^C_x|) + min(|I^U(L)_x| + Top_t(I^U)_x, |N^U_x|)}{|^1\boldsymbol{N}_x|} \quad (57)$$

and

$$dIoU_{min} = 0 \quad (58)$$

Now, let us consider the case where we explicitly design this combined quantity. As before, we observe that $dIoU_{min} = 0$. For the numerator, however, we can refine the estimation: including an AND operator at any step will, by design, remove all

the unique quantities of the current label. Thus:

$$dIoU_{max}(OR, AND) = \frac{min(|I^C_{max}(L)_x| + Top_t(I^C)_x, |N^C_x|) + Top_t(I^U)_x}{|^1\boldsymbol{N}_x| + (|E^C_{max}(L)_x| + |Bott_1(E^C)_x| - |SE^C_x|} \tag{59}$$

However, since $|Bott_1(E^C)_x|$ is 0 in most datasets, the denominator reduces to $|^1\boldsymbol{N}_x|$. Thus, the only practical difference lies in the numerator, where the $|I^U(L)_x|$ term is missing. However, in general, $Top_t(I^U)_x$ is much larger than $|I^U(L)_x|$, since it includes the maximum per sample across all concepts in the dataset. This makes the difference between $dIoU^{(OR,AND)}_{max}$ (from explicit design) and our proposed simplification relatively small. Similar observations can be made for all the other combinations of the operators considered in this paper.

From a practical perspective, and given that the optimal algorithm often converges toward breadth-first exploration (Section 4.3), the gain of the refined design is marginal. Conversely, our simplified approach, based on combining the values of exclusive paths of operators, scales more efficiently and facilitates future extensions. Indeed, new logic operators can be incorporated into the optimal algorithm simply by providing their estimates for the exclusive path.

## F. Proof of Optimality

This section builds upon the derivations introduced in Section E to demonstrate that the heuristics employed are admissible, ensuring the algorithm is guaranteed to find the optimal solution and is complete. This section assumes the reader to be familiar with the terminology introduced in Section 3.

### F.1. Proof of Admissibility of the Label Quantities Heuristic

Let $L$ be the label obtained by concatenating the label $L_\leftarrow$ with the concept $L_\rightarrow$. This heuristic aims at estimating the IoU score of $L$. By Equation (4), the IoU score is equivalent to the decomposed $dIoU$ score. To be admissible, the heuristic cannot underestimate the numerator or overestimate the denominator. Thus, it must satisfy the following constraints:

$$\begin{cases} |I^U_{max}(L)_x| + |I^C_{max}(L)_x| \ge |I^U(L)_x| + |I^C(L)_x| & (60) \\ 0 \le |E^U_{min}(L)_x| + |E^C_{min}(L)_x| \le |E^U(L)_x| + |E^C(L)_x| & (61) \end{cases}$$

for all $x \in \mathfrak{D}$. Equation (60) ensures that the heuristic returns an optimistic estimate of the intersection (numerator), while Equation (61) ensures a pessimistic estimate of the denominator of Equation (3).

We observe that unique elements are always disjoint and cannot be shared by Definition 3.1. Therefore, their value can be computed exactly by using Observation 1, and thus $|I^U_{max}(L)_x| = |I^U(L)_x|$ and $|E^U_{min}(L)_x| = |E^U(L)_x|$.

Therefore, the constraints reduce to:

$$\begin{cases} |I^C_{max}(L)_x| \ge |I^C(L)_x| & (62) \\ |E^C_{min}(L)_x| \le |E^C(L)_x| & (63) \end{cases}$$

where the lower bound in the second inequality is omitted because cardinalities are non-negative.

These constraints are satisfied by design because of the derivations described in the previous section. Namely, if $L_\leftarrow$ and $L_\rightarrow$ are disjoint, then they share no common elements, and the algorithm can compute the exact values. Thus $|I^C_{max}(L)_x| = |I^C(L)_x|$ and $|E^C_{min}(L)_x| \le |E^C(L)_x|$. Note that in this case, the maximum and the minimum of these quantities coincide.

For the AND NOT operator, both the unique and the common elements reduce to those of the left side $L_\leftarrow$. Therefore, $dIoU(L) = dIoU(L_\leftarrow)$, and any path reachable from $L$ is also reachable from $L_\leftarrow$ since they are equivalent.

Conversely, if $L_\leftarrow$ and $L_\rightarrow$ overlap, we analyze each operator $\oplus$.

**Case OR operator.** For the intersection, recall that

$$|I^C_{max}(L)_x| = min\left(|I^C_{max}(L_\leftarrow)_x| + |I^C(L_\rightarrow)_x|, \; |N^C_x|\right) \tag{64}$$

If the sets overlap, then

$$|I^C(L)_x| = |I^C(L_\leftarrow)_x| + |I^C(L_\rightarrow)_x| - |I^C(L_\leftarrow)_x \cap I^C(L_\rightarrow)_x| \tag{65}$$

Because they overlap, they share at least one element, and thus

$$|I^C(L_\leftarrow)_x \cap I^C(L_\rightarrow)_x| \geq 1. \tag{66}$$

This implies

$$|I^C(L)_x| \leq |I^C(L_\leftarrow)_x| + |I^C(L_\rightarrow)_x| \tag{67}$$

Furthermore, by the definition of $N_x^C$,

$$|I^C(L)_x| \leq |N_x^C| \tag{68}$$

Thus $|I_{max}^C(L)_x| \geq |I^C(L)_x|$.

For the extras, recall that

$$|E_{min}^C(L)_x| = \max(|E_{min}^C(L_\leftarrow)_x|, |E^C(L_\rightarrow)_x|) \tag{69}$$

If the sets overlap, then

$$|E^C(L)_x| = |E^C(L_\leftarrow)_x \cup E^C(L_\rightarrow)_x| \tag{70}$$

Because the cardinality of a set cannot exceed the cardinality of its union with another set, Equation (63) is satisfied regardless of which element is selected by the max operator.

**Case AND operator.**   In this case, for the intersection,

$$|I_{\max}^C(L)_x| = \min\left(|I_{\max}^C(L_\leftarrow)_x|, |I^C(L_\rightarrow)_x|\right) \tag{71}$$

The true value is

$$|I^C(L)_x| = |I^C(L_\leftarrow)_x \cap I^C(L_\rightarrow)_x| \tag{72}$$

since the AND operator is 0-preserving. Because an intersection cannot exceed either operand, then

$$|I_{\max}^C(L)_x| \geq |I^C(L)_x| \tag{73}$$

Regarding the extras, recall that

$$|E_{min}^C(L)_x| = \max(|E_{min}^C(L_\leftarrow)_x| + |E^C(L_\rightarrow)_x| - |SE_x^C|, 0) \tag{74}$$

The zero case is proven by the non-negativity of the cardinality. Conversely, if the max operator selects the first factor, then we can prove it never overestimates the true value by induction by fixing $L_\leftarrow$ to an atomic concept. In this case, we have access to the exact computation and thus

$$|E_{min}^C(L)_x| = \max(|E^C(L_\leftarrow)_x| + |E^C(L_\rightarrow)_x| - |SE_x^C|, 0) \tag{75}$$

If the sets overlap, then the true value is

$$|E^C(L)_x| = |E^C(L_\leftarrow)_x \cap E^C(L_\rightarrow)_x| \tag{76}$$

Recall that by the definition of $SE^C$, any sum between the cardinality of an extra disjoint set is lower than the cardinality of $SE^C$. Formally,

$$|SE_x^C| \geq |E^C(L_j)_x| + |E^C(L_k)_x|, \ \forall L_j, L_k \in \mathfrak{L}^1 : E^C(L_j)_x \cap E^C(L_k)_x = \emptyset \tag{77}$$

We rewrite $|E^C(L_\leftarrow)_x|$ and $|E^C(L_\rightarrow)_x|$:

$$|E^C(L_\leftarrow)_x| = |E^C(L_\leftarrow)_x \setminus E^C(L_\rightarrow)_x| + |E^C(L_\leftarrow)_x \cap E^C(L_\rightarrow)_x| \tag{78}$$

$$|E^C(L_\rightarrow)_x| = |E^C(L_\rightarrow)_x \setminus E^C(L_\leftarrow)_x| + |E^C(L_\leftarrow)_x \cap E^C(L_\rightarrow)_x| \tag{79}$$

Since the sets $E^C(L_\leftarrow)_x \setminus E^C(L_\rightarrow)_x$, $E^C(L_\rightarrow)_x \setminus E^C(L_\leftarrow)_x$, and $E^C(L_\leftarrow)_x \cap E^C(L_\rightarrow)_x$ are pairwise disjoint, then:

$$|E^C(L_\leftarrow)_x \setminus E^C(L_\rightarrow)_x| + |E^C(L_\rightarrow)_x \setminus E^C(L_\leftarrow)_x| + |E^C(L_\leftarrow)_x \cap E^C(L_\rightarrow)_x| \leq |SE_x^C| \tag{80}$$

Combining Equation (80) with Equation (75) and the fact that the operator selected the first argument, we obtain

$$|E^C(L_\leftarrow)_x| + |E^C(L_\rightarrow)_x| - |SE_x^C| = |E^C(L_\leftarrow)_x \setminus E^C(L_\rightarrow)_x| + |E^C(L_\leftarrow)_x \cap E^C(L_\rightarrow)_x| \tag{81}$$

$$+ |E^C(L_\rightarrow)_x \setminus E^C(L_\leftarrow)_x| + |E^C(L_\leftarrow)_x \cap E^C(L_\rightarrow)_x| - |SE_x^C| \tag{82}$$

which is smaller than or equal to $|E^C(L_\leftarrow)_x \cap E^C(L_\rightarrow)_x|$ due to Equation (80), thus satisfying Equation (63). Since $|E_{min}^C(L)_x|$ does not overestimate the true value for atomic concepts, and any successive step uses this underestimation to compute $|E^C(L_\leftarrow)_x| + |E^C(L_\rightarrow)_x| - |SE_x^C|$, then Equation (63) holds for any label $L \in \mathfrak{L}^n$.

**Case AND NOT operator.** In this case, for the intersection,

$$|I_{\max}^C(L)_x| = \min\left(|I_{\max}^C(L_\leftarrow)_x|, \ |N_x^C| - |I^C(L_\rightarrow)_x|\right) \tag{83}$$

The true value is

$$|I^C(L)_x| = |I^C(L_\leftarrow)_x \setminus I^C(L_\rightarrow)_x| = |I^C(L_\leftarrow)_x \cap (N_x^C \setminus I^C(L_\rightarrow)_x)| \tag{84}$$

Since AND NOT flips the right side and is 1-preserving on the left:

- If the minimum selects $|I_{\max}^C(L_\leftarrow)_x|$, then because a set difference cannot exceed its left operand and $I_{\max}^C$ is an overestimation,

$$|I_{\max}^C(L_\leftarrow)_x| \geq |I^C(L_\leftarrow)_x \setminus I^C(L_\rightarrow)_x| \tag{85}$$

- If the minimum selects $|N_x^C| - |I^C(L_\rightarrow)_x|$, then this equals $|N_x^C \setminus I^C(L_\rightarrow)_x|$, and because intersections cannot exceed either operand,

$$|I_{\max}^C(L)_x| \geq |I^C(L)_x| \tag{86}$$

Regarding the extras, recall that

$$|E_{min}^C(L)_x| = \max(|E_{min}^C(L_\leftarrow)_x| - |E^C(L_\rightarrow)_x|, 0) \tag{87}$$

The zero case follows from the non-negativity of cardinality. Conversely, if the max operator selects the first quantity, then we can prove it never overestimates the true value by induction, by fixing $L_\leftarrow$ to an atomic concept, similarly to the AND case. In this case, we have access to the exact computation and thus

$$|E_{min}^C(L)_x| = \max(|E^C(L_\leftarrow)_x| - |E^C(L_\rightarrow)_x|, 0) \tag{88}$$

If the sets overlap, then the true value is

$$|E^C(L)_x| = |E^C(L_\leftarrow)_x \setminus E^C(L_\rightarrow)_x| = |E^C(L_\leftarrow)_x| - |E^C(L_\rightarrow)_x \cap E^C(L_\leftarrow)_x| \tag{89}$$

This is clearly greater than $|E^C(L_\leftarrow)_x| - |E^C(L_\rightarrow)_x|$ and thus Equation (63) is satisfied. Similarly to the AND operator case, since $|E_{min}^C(L)_x|$ does not overestimate the true value for atomic concepts, and any successive step uses this underestimation to compute $|E_{min}^C(L_\leftarrow)_x| - |E^C(L_\rightarrow)_x|$, then Equation (63) holds for any label $L \in \mathfrak{L}^n$.

Thus, for all operators considered in this paper, **the heuristic never underestimates the true common intersection, and admissibility is satisfied.**

**Aggregated Computation.** The admissibility of the aggregated version follows directly from the observation that the estimation produced by aggregating quantities for the numerator is always larger than the one computed per sample, due to the min operator in the computation of $|I_{\max}^C(L)_x|$ for all operators. Therefore, because the sample-based computation is an overestimation of the true intersection, the aggregated estimation is also an overestimation.

Conversely, the aggregated numerator is always equal to or lower than the sample-based numerator. Indeed, in the case of OR, $\sum \max(|E_{min}^C(L_\leftarrow)_x|, |E^C(L_\rightarrow)_x|) \geq max(\sum |E_{min}^C(L_\leftarrow)_x|, \sum |E^C(L_\rightarrow)_x|)$ and Equation (63) is satisfied. Similarly, in the case of AND and AND NOT, the estimation uses the max function over a difference and is therefore maximized when $L_\leftarrow$ and $L_\rightarrow$ fully overlap across all samples, which corresponds exactly to the case represented by the aggregated computation. Thus Equation (63) is satisfied.

In conclusion, because the aggregated computation overestimates the numerator and underestimates the denominator, the admissibility of the heuristic using aggregated computation is satisfied.

### F.2. Remaining Minimum Estimations

While not necessary for the proof of admissibility, in this section, we prove the minimum estimations for the quantities not discussed in the previous paragraphs, namely $I_{min}^C(L)_x$ and $E_{max}^C(L)_x$. This proof will be useful to establish the admissibility of the Path Heuristic and the completeness of the algorithm. Both follow the same reasoning used for $E_{min}^C(L)_x$ and $I_{max}^C(L)_x$, respectively.

Namely, the minimum estimations for the intersection used by the algorithm are

$$|I_{min}^C(L)_x| = \begin{cases} \max(|I_{min}^C(L_\leftarrow)_x|, |I^C(L_\rightarrow)_x|) & \text{if } \oplus = \vee & (90) \\ \max(|I_{min}^C(L_\leftarrow)_x| + |I^C(L_\rightarrow)_x| - |N_x^C|, 0) & \text{if } \oplus = \wedge & (91) \\ \max(|I_{min}^C(L_\leftarrow)_x| - |I^C(L_\rightarrow)_x|, 0) & \text{if } \oplus = \neg\wedge & (92) \end{cases}$$

The true values are

$$|I^C(L)_x| = \begin{cases} |I^C(L_\leftarrow)_x \cup I^C(L_\rightarrow)_x| & \text{if } \oplus = \vee & (93) \\ |I^C(L_\leftarrow)_x \cap I^C(L_\rightarrow)_x| & \text{if } \oplus = \wedge & (94) \\ |I^C(L_\leftarrow)_x \setminus I^C(L_\rightarrow)_x| & \text{if } \oplus = \neg\wedge & (95) \end{cases}$$

Following the same reasoning used in Section F.1 to prove the underestimation of $E_{min}^C$, we obtain that $|I_{min}^C(L)_x| \leq |I^C(L)_x|$ holds for the OR operator, because the cardinality of a set cannot exceed that of its union with another set.

Similarly, for the other operators, we replace $E^C$ with $I^C$ and $SE^C$ with $N^C$ when needed. For the AND operator, we expand

$$|I^C(L_\leftarrow)_x| + |I^C(L_\rightarrow)_x| = |I^C(L_\leftarrow)_x \setminus I^C(L_\rightarrow)_x| + |I^C(L_\leftarrow)_x \cap I^C(L_\rightarrow)_x| \\ + |I^C(L_\rightarrow)_x \setminus I^C(L_\leftarrow)_x| + |I^C(L_\leftarrow)_x \cap I^C(L_\rightarrow)_x| \quad (96)$$

and using that the three disjoint parts satisfy

$$|I^C(L_\leftarrow)_x \setminus I^C(L_\rightarrow)_x| + |I^C(L_\rightarrow)_x \setminus I^C(L_\leftarrow)_x| + |I^C(L_\leftarrow)_x \cap I^C(L_\rightarrow)_x| \leq |N_x^C| \quad (97)$$

we obtain $|I_{min}^C(L)_x| \leq |I^C(L)_x|$.

For the AND NOT operator, we can rewrite the true value as

$$|I^C(L)_x| = |I^C(L_\leftarrow)_x \setminus I^C(L_\rightarrow)_x| = |I^C(L_\leftarrow)_x| - |I^C(L_\leftarrow)_x \cap I^C(L_\rightarrow)_x| \quad (98)$$

which is greater than $|I^C(L_\leftarrow)_x| - |I^C(L_\rightarrow)_x|$ and thus $|I_{min}^C(L)_x| \leq |I^C(L)_x|$.

Analogously, for $E^C$, the estimations and the true values are

$$|E_{max}^C(L)_x| = \begin{cases} \min(|E_{max}^C(L_\leftarrow)_x| + |E^C(L_\rightarrow)_x|, |SE_x^C|) & \text{if } \oplus = \vee & (99) \\ \min(|E_{max}^C(L_\leftarrow)_x|, |E^C(L_\rightarrow)_x|) & \text{if } \oplus = \wedge & (100) \\ \min(|E_{max}^C(L_\leftarrow)_x|, |SE_x^C| - |E^C(L_\rightarrow)_x|) & \text{if } \oplus = \neg\wedge & (101) \end{cases}$$

and

$$|E^C(L)_x| = \begin{cases} |E^C(L_\leftarrow)_x \cup E^C(L_\rightarrow)_x| & \text{if } \oplus = \vee & (102) \\ |E^C(L_\leftarrow)_x \cap E^C(L_\rightarrow)_x| & \text{if } \oplus = \wedge & (103) \\ |E^C(L_\leftarrow)_x \setminus E^C(L_\rightarrow)_x| & \text{if } \oplus = \neg\wedge & (104) \end{cases}$$

Using the same reasoning as in Section F.1 for $I_{max}^C$, we see that $|E_{max}^C(L)_x| \geq |E^C(L)_x|$ holds for the AND operator because the cardinality of an intersection cannot exceed that of either operand. For the OR operator, $|E_{max}^C(L)_x| \geq |E^C(L)_x|$ holds because $|E^C(L)_x| \leq |E^C(L_\leftarrow)_x| + |E^C(L_\rightarrow)_x|$ and $|E^C(L)_x| \leq |SE_x^C|$. Finally, for the AND NOT operator, the inequality follows because a set difference cannot exceed its left operand when the first term is selected, and because an intersection cannot exceed either operand when the second term is selected.

### F.3. Proof of Admissibility of the Path Heuristic

Let $L_i$ be a generic label of length $i$. This heuristic aims at estimating the maximum achievable IoU score by any label $L_j$ in the state space $\mathfrak{L}_{L_i}^n$, encompassing all labels of length up to $n$ with $i < n$, $t = n - i$, and $t > 0$ (i.e., at least 1 concept is added) such that the leftmost part of each label $L_j$ corresponds to $L_i$ (i.e., $L_j = (((L_i \oplus_j L_k) \dots) \oplus_n L_l))$.

Let $L_*$ be the label in $\mathfrak{L}_{L_i}^n$ associated with the highest IoU score. To be admissible, the heuristic must satisfy the following constraints:

$$\begin{cases} |I^{\widehat{U(L_*)}}_x| + |I^{\widehat{C(L_*)}}_x| \geq |I^U(L_*)_x| + |I^C(L_*)_x| & (105) \\ 0 \leq |E^{\widehat{U(L_*)}}_x| + |E^{\widehat{C(L_*)}}_x| \leq |E^U(L_*)_x| + |E^C(L_*)_x| & (106) \end{cases}$$

for all $x \in \mathfrak{D}$. In the constraints and in the following, we use the term "true value" to refer to the exact computation and use the symbol $QU\widehat{ANTIT}Y(L_*)_x$ to express the estimation of the maximum possible value reachable from $L_i$ for the considered quantity. Equation (105) ensures that the heuristic returns an optimistic estimate of the intersection (numerator), while equation Equation (106) guarantees a pessimistic estimate of the denominator of Equation (3).

These constraints are satisfied by design due to the derivations described in the previous section for the Top and Bott vectors, which encode the maximum and minimum possible improvements, respectively. To prove this, let us consider the exclusive paths for each operator considered in this paper.

**Case OR Operator.** In this case, the heuristic estimation for both the numerator and denominator in Equation (3) is

$$|I^{\widehat{U(L_*)}}_x| + |I^{\widehat{C(L_*)}}_x| = \min(|I^C_{max}(L)_x| + Top_t(I^C)_x, |N^C_x|) + \min(|I^U(L)_x| + Top_t(I^U)_x, |N^U_x|) \quad (107)$$

and

$$|E^{\widehat{U(L_*)}}_x| + |E^{\widehat{C(L_*)}}_x| = \max(|E^C_{min}(L)_x| + |E^U(L)_x|, \ Bott_1(E^C)_x + Bott_1(E^U)_x \quad (108)$$

where we drop the $|^1N|$ term because it is common to both expressions and does not affect the proof.

Without loss of generality, let us consider the ideal case where there exists a sequence of concepts up to length $n$, $\{L_j, \dots, L_n\}$, such that each of them fully overlaps with $L_i$ on the extras, and each of them is pairwise disjoint and disjoint from the elements of $L_i$ that already overlap with the neuron activations. Moreover, assume that these concepts have the largest possible number of elements overlapping with the neuron activation. This construction represents the ideal scenario for the OR operator, and no other concept in the dataset can further improve the IoU.

In this ideal case, the true value of the denominator is

$$|E^U(L)_x| + |E^C(L)_x| \quad (109)$$

because all concepts fully overlap with $L$ on the extras.

By comparing the true value with Equation (108), we obtain

$$|E^{\widehat{U(L_*)}}_x| + |E^{\widehat{C(L_*)}}_x| \ \leq \ |E^U(L_*)_x| + |E^C(L_*)_x| \quad (110)$$

Indeed, if the max operator selects the first quantity, the inequality holds because $|E^C_{min}(L)_x|$ is an underestimation, as proven in Section F.1. If the max operator selects the second quantity, the inequality follows from the definition of the $Bott_1(E^C)_x$ and $Bott_1(E^U)_x$ vectors (Section 3.2.2). These vectors store, for each sample $x$, the minimum number of extras among all concepts in the dataset (including 0). Thus, for any concept $L_k$ in $\{L_j, \dots, L_n\}$, if $L_k$ has the smallest number of common extras for that sample, then $Bott_1(E^C)_x = |E^C(L_k)_x|$; otherwise,

$$Bott_1(E^C)_x < |E^C(L_k)_x| \quad (111)$$

and thus

$$Bott_1(E^C)_x \leq |E^C(L_*)_x| \quad (112)$$

The same argument applies to unique extras, and therefore Equation (106) is satisfied.

In the numerator case, because all the concepts are pairwise disjoint on the intersection and disjoint with respect to $L_i$, we can sum their quantities for ease of understanding as a single aggregated factor:

$$|I^C(L_*)_x| = |I^C(L_j)_x| + \cdots + |I^C(L_n)_x| \tag{113}$$

$$|I^U(L_*)_x| = |I^U(L_j)_x| + \cdots + |I^U(L_n)_x| \tag{114}$$

The true value of the numerator in this ideal case is given by

$$|I^C(L_i)_x| + |I^U(L_i)_x| + |I^C(L_*)_x| + |I^U(L_*)_x| \tag{115}$$

By comparing the true value with Equation (107), we can note that when the minimum selects $|N_x^C|$ and $|N_x^U|$, we have

$$|\widehat{I^C(L_*)}_x| + |\widehat{I^U(L_*)}_x| \geq |I^C(L_*)_x| + |I^U(L_*)_x| \tag{116}$$

by definition, since $\{L_j, \ldots, L_n\}$ are disjoint in the intersection with respect to $L_i$ and their sum cannot exceed the available space for the common and unique intersections.

If the minimum selects the first terms, then as proven in Section F.1, $|I_{max}^C(L_i)_x|$ is an overestimation, and thus the constraint reduces to proving that

$$Top_t(I^C)_x \geq |I^C(L_*)_x| \tag{117}$$

$$Top_t(I^U)_x \geq |I^U(L_*)_x| \tag{118}$$

This is satisfied by construction of the $Top_t$ vectors. These vectors store, for each sample, the sum of the $t$ concepts in the dataset with the largest number of intersection elements for that sample. Note that $t$ is also the length of $L_*$. Thus, for any sample $x \in \mathfrak{D}$, if the $t$ concepts with the largest intersections correspond to those in $L_*$, then $Top_t(I^C)_x = |I^C(L_*)_x|$, and otherwise $Top_t(I^C)_x > |I^C(L_*)_x|$. The same argument applies to the unique intersection, and therefore Equation (105) is satisfied.

**Case AND Operator.** In the AND case, the heuristic estimation for both the numerator and denominator in Equation (3) is

$$|\widehat{I^U(L_*)}_x| + |\widehat{I^C(L_*)}_x| = \min(|I_{max}^C(L)_x|, Top_1(I^C)_x) \tag{119}$$

and

$$|\widehat{E^U(L_*)}_x| + |\widehat{E^C(L_*)}_x| = 0 \tag{120}$$

where we drop the $|N|$ term because it is common to both expressions and does not affect the proof. We can note that Equation (106) is satisfied by the non-negativity of the cardinality.

To prove Equation (105), without loss of generality, let us consider the ideal case where there exists a sequence of concepts up to length $n$, $\{L_j, \ldots, L_n\}$, such that each of them fully overlaps with $L_i$ on the common intersection. This construction represents the ideal scenario for the AND operator since the AND operator cannot increase the intersection, but this construction allows preserving all the already intersecting common elements of $L_i$.

The true value of the numerator in this ideal case is given by $|I^C(L_i)_x|$ since by definition the AND operator removes all the unique elements and, in this ideal case, all the common intersection of $L_i$ are preserved.

If the minimum selects the first term, then as proven in Section F.1, $|I_{max}^C(L_i)_x|$ is an overestimation, and thus the constraint reduces to proving that

$$Top_1(I^C)_x \geq |I^C(L_i)_x| \tag{121}$$

$$Top_1(I^U)_x \geq |I^U(L_i)_x| \tag{122}$$

This is satisfied by the definition of the $Top_1$ vectors and the fact that in this ideal case, the added concepts fully overlap. These vectors store, for each sample, the largest number of intersection elements for that sample among the concepts in $\mathfrak{L}^1$. Thus, since any concept $L_k$ in $\{L_j, \ldots, L_n\}$ is assumed to fully overlap with $L_i$ in the intersection, it either has the largest number of common intersection for that sample and thus $Top_1(I^C)_x = |I^C(L_k)_x|$, or it is not the largest and thus

$$Top_1(I^C)_x > |I^C(L_k)_x| \tag{123}$$

Therefore

$$Top_1(I^C)_x \geq |I^C(L_*)_x| \tag{124}$$

and Equation (106) is satisfied. In the general case, if any of the added concepts $L_k$ does not fully overlap, then the true value becomes $|I^C(L_i)_x \cap I^C(L_k)_x|$. In this case, Equation (106) is satisfied by the fact that the cardinality of an intersection cannot exceed that of either operand when $L_k$ is the concept associated with the largest common intersection in that sample and always in the other cases.

**Case AND NOT Operator.** In the AND NOT case, the heuristic estimation for both the numerator and denominator in Equation (3) is

$$|\widehat{I^U(L_*)}_x| + |\widehat{I^C(L_*)}_x| = |I^U(L)_x| + \min(|I^C_{max}(L)_x|, |N^C_x| - Bott_1(I^C)_x) \tag{125}$$

and

$$|\widehat{E^U(L_*)}_x| + |\widehat{E^C(L_*)}_x| = |E^U(L)_x| \tag{126}$$

As done previously, let us consider the ideal case where there exists a sequence of concepts up to length $n$, $\{L_j, \ldots, L_n\}$, such that each of them fully overlaps with $L_i$ on the extras, and each of them is pairwise disjoint and disjoint from the elements of $L_i$ that already overlap with the neuron activations. This construction represents the ideal scenario for the AND NOT operator, and no other concept in the dataset can further improve the IoU. We can note that the negation of these concepts corresponds to the ideal case of the AND operator, since the negation fully overlaps in the intersection and is disjoint in the extras. The difference here is that all the unique elements of $L_i$ are preserved.

Thus, the true value of the numerator in this ideal case is $|I^C(L_i)_x| + |I^U(L_i)_x|$ since the AND NOT operator preserves all unique elements and preserves all common intersections of $L_i$.

Since $|I^U(L_i)_x|$ is an exact quantity, the constraint reduces to proving that

$$\min(|I^C_{max}(L_i)_x|, |N^C_x| - Bott_1(I^C)_x) \geq |I^C(L_i)_x| \tag{127}$$

If the minimum selects the first term, then, as proven in Section F.1, $|I^C_{max}(L_i)_x|$ is an overestimation, and thus Equation (105) is satisfied. If the minimum selects the second term, we must prove that

$$|N^C_x| - Bott_1(I^C)_x \geq |I^C(L_i)_x| \tag{128}$$

Recall that $Bott_1$ stores, for each sample $x$, the minimum number of common intersections among all concepts in the dataset (including 0). Therefore, the quantity $|N^C_x| - Bott_1(I^C)_x$ represents the maximum number of intersecting elements obtainable from the negation of these concepts. In other words, it represents the maximum intersection space coverable by 0-entries for that sample.

Thus, for any concept $L_k$ in $\{L_j, \ldots, L_n\}$, if $L_k$ has the smallest number of common intersections for that sample, then its negation corresponds to $|N^C_x| - Bott_1(I^C)_x$, and because it is disjoint with respect to $L_i$, we have

$$|N^C_x| - Bott_1(I^C)_x = |I^C(L_i)_x| \tag{129}$$

Conversely, if $L_k$ does not have the smallest number of common intersections for that sample, then

$$Bott_1(I^C)_x \leq |I^C(L_i)_x| \tag{130}$$

and therefore

$$|N^C_x| - Bott_1(I^C)_x > |I^C(L_i)_x| \tag{131}$$

so Equation (105) is satisfied.

In the general case, if any of the added concepts $L_k$ is not fully disjoint, then the true value becomes

$$|I^C(L_i)_x \setminus I^C(L_k)_x| \tag{132}$$

where $L_k$ is the concept with the largest overlap with $L_i$. Since $I^C(L_k)_x$ removes at least $Bott_1(I^C)_x$ from $I^C(L_i)_x$, and $|I^C(L_i)_x| \leq |N^C_x|$ by definition, Equation (105) still holds.

Regarding the denominator, since the concepts are disjoint in the extras, the true value is $|E^U(L_i)_x|$ because the AND NOT operator preserves all unique elements. As proven in Section F.1, $|E^U_{min}(L_i)_x|$ is an underestimation, and thus Equation (106) is satisfied.

**Combined Path.** For ease of notation, let us consider the case where 2 operators are involved in the combined path. However, the proof is easily extendable to the case of all three operators. In this case, the heuristic estimation for both the numerator and denominator in Equation (3) is

$$|I^U\widehat{(L_*)}_x| + |I^C\widehat{(L_*)}_x| = max(|I^U_{max}\widehat{(L\oplus_1)}_x| + |I^C_{max}\widehat{(L\oplus_1)}_x|, |I^U_{max}\widehat{(L\oplus_2)}_x| + |I^C_{max}\widehat{(L\oplus_2)}_x|) \qquad (133)$$

and

$$|E^U\widehat{(L_*)}_x| + |E^C\widehat{(L_*)}_x| = min(|E^U_{min}\widehat{(L\oplus_1)}_x| + |E^C_{min}\widehat{(L\oplus_1)}_x|, |E^U_{min}\widehat{(L\oplus_2)}_x| + |E^C_{min}\widehat{(L\oplus_2)}_x|) \qquad (134)$$

where $L\oplus_1$ and $L\oplus_2$ represent the estimations for the exclusive paths of each operator involved in the path. The proof that Equation (105) is satisfied follows directly from the observation that any combination of paths must include at least one of the two AND operators (AND or AND NOT), and that both of them cannot increase the number of 1s due to Observation 1, because they are 0-preserving. Therefore, the maximum intersection achievable is given by the intersection reached just before concatenating one of the AND operators, which corresponds, at most, to the maximum achievable by the exclusive path of the other operator. Similarly, Equation (106) follows from the observation that any combination of paths must include at least one of the two AND operators. If it contains the AND operator, then $E^U\widehat{(L_*)}_x| + |E^C\widehat{(L_*)}_x| = 0$ and Equation (106) is satisfied by the non-negativity of the cardinality. Conversely, if it contains only OR and AND NOT operators, then $E^U\widehat{(L_*)}_x| + |E^C\widehat{(L_*)}_x| = |E^U(L)_x|$. In this case, Equation (106) is satisfied because OR is 1-preserving and the AND NOT operator preserves all the unique elements, and thus any new concepts chained will preserve or increase (in the case of OR) the number of unique extras and possibly add or preserve common extras.

**Aggregated Computation.** The admissibility of the aggregated version follows the same proof as in the sample version. The only difference is that the $Top$ and $Bott$ vectors are computed per concept rather than per sample. In this case, they naturally represent overestimations and underestimations since they represent the ideal choice of the concepts to add for each operator at the dataset level (e.g., $Top$ represents the case where the top $t$ concepts in the dataset for the intersection are pairwise disjoint and disjoint with $L_i$ over the full dataset).

### F.4. Remaining Minimum Estimations

While not necessary for the proof of admissibility, in this section, we prove that the minimum estimations for the path discussed in the previous paragraphs are underestimations. This result will be useful for guaranteeing the completeness of the algorithm.

Namely, the minimum estimations for the intersection used by the algorithm are

$$|I_{min}\widehat{(L*)}_x| \begin{cases} max(|I^C_{min}(L)_x| + |I^U(L)_x|, Bott_1(I^C)_x + Bott_1(I^U)_x) & \vee \text{ Path} \qquad (135) \\ 0 & \wedge \text{ Path} \qquad (136) \\ |I^U(L)_x| & \wedge\neg \text{ Path} \qquad (137) \end{cases}$$

We can note that the minimum estimations for the AND and AND NOT paths are underestimations by non-negativity of the cardinality in the AND path and by the property that the AND NOT operator preserves all unique elements.

For the OR path, if the maximum operator selects the first term, then this represents an underestimation since $|I^C_{min}(L)_x|$ is an underestimation of $|I^C(L)_x|$, as proven in Section F.2, and the OR operator cannot reduce the intersection. If the max operator selects the second term, then $|I_{min}\widehat{(L)}_x| \le |I(L)_x|$ because OR is 1-preserving and any concept $L_k$ chained to $L_i$ through an OR operator must satisfy

$$|I^C(L_k)_x| + |I^U(L_k)_x| \ge Bott_1(I^C)_x + Bott_1(I^U)_x \qquad (138)$$

by definition of the $Bott_1$ vectors.

Regarding the maximum denominator, recall that the algorithm uses the following estimate

$$|Union_{max}\widehat{(L*)}_x| = |^1\boldsymbol{N}_x| + \begin{cases} min(|E^C_{max}(L)_x| + Top_t(E^C)_x, |SE^C_x|) & (139) \\ \quad + min(|E^U(L)_x| + Top_t(E^U)_x, |SE^U_x|) & \vee \text{ Path} \qquad (140) \\ min(|E^C_{max}(L)_x|, Top_1(E^C)_x) & \wedge \text{ Path} \qquad (141) \\ |E^C_{max}(L)_x| + min(|E^U(L)_x|, |SE^C_x| - Bott_1(E^C)_x) & \wedge\neg \text{ Path} \qquad (142) \end{cases}$$

where we can drop $|^1\boldsymbol{N}_x|$ since it is shared by both the true values and the estimations. These estimates are similar to the ones used for estimating the maximum possible intersection. Therefore, the result follows by applying the same reasoning used for $I^C_{max}(L)_x$ and replacing $I^C$ with $E^C$, $I^U$ with $E^U$, and $N^C$ with $SE^C$.

### F.5. Optimality of the algorithm

Because the heuristics used to explore the state space are both admissible (Sections F.1 and F.3), because the quantities used to prune the frontier are guaranteed to be underestimations of the real IoU reachable from that node (Sections F.2 and F.4), and because the algorithm explores exhaustively all the nodes in the search space with an estimated alignment greater than the found solution, the algorithm **is complete and guaranteed to find the combination of concepts** $(L_1 \oplus L_2 \oplus \cdots \oplus L_n)$ **among the concepts in $\mathfrak{L}^n$ that captures the highest possible alignment (IoU)**.

## G. Other Neuron and Concept-Based Explanations

While this paper focuses on methods targeting neuron spatial alignment, the literature offers a wide range of explanation approaches that use concepts to derive explanations (Räuker et al., 2023; Gilpin et al., 2018) or to decode other types of neurons (Hesse et al., 2025; Srinivas et al., 2025; Bau et al., 2020; O'Mahony et al., 2025; Bykov et al., 2023) and layer behaviors (Gao et al., 2025; Bricken et al., 2023). Our work is inspired by this growing interest in understanding neuron behaviors and using concepts to build explanations. However, the similarities end there: these approaches belong to distinct families of interpretability methods that differ fundamentally from compositional explanations across multiple dimensions, such as their goal (e.g., alignment vs. correlation), scope (e.g., neuron clusters vs. layers), assumptions (e.g., access to internals and availability of masks), representation (e.g., logical vs. statistical explanations), and phase (post-hoc vs. ante-hoc). Because these families are not able to measure the spatial alignment between activations and the locations of concepts, they have traditionally been regarded as complementary rather than competing categories with respect to compositional explanations.

To clarify the differences, consider Figure 1. In this case, the compositional explanation expresses the fact that the regions of activation within images for neuron 56 match (i.e., are aligned with) the regions of the images corresponding to tables or chairs in a dining room, which correspond to the activation area highlighted in blue. This type of explanation combines three properties: (i) spatial alignment, (ii) logical composition of concepts, and (iii) neuron-level specificity. These properties cannot be simultaneously expressed by alternative families of methods. For example, let us consider two methods from other families of methods: CLIP-Dissect (Oikarinen and Weng, 2023) and SAE (Huben et al., 2024).

For each sample, CLIP-Dissect extracts a single value (e.g., the max over the whole image). Therefore, for a dataset of $n$ samples, CLIP-Dissect computes a vector of $n$ values representing those activations. It then assigns to the neuron the single concept whose representation is most correlated with this vector. For this specific neuron, CLIP-Dissect selects the concept *Inn indoor scene*, which refers to the whole image. As such, there are 3 fundamental differences. First, CLIP-Dissect does not capture spatial alignment between regions, since it abstracts the encoding of an image into a single scalar. Second, the objective is different, and therefore the methods answer different questions. Indeed, correlation does not imply alignment, since the value representing the sample is not constrained to be within any specific region. Third, CLIP-Dissect assigns a single concept to each neuron. We could potentially extract the $top - k$ correlated concepts (for neuron 56, the top 3 would include *hunting lodge indoor scene* and *hotel breakfast area scene*). However, it would be unclear how these concepts relate to each other (if any relationship exists). In summary, there is a difference in objective, explanation specificity, and explanation complexity between the two methods.

On the other hand, SAEs aim to find a projection that maps the layer representation into a more interpretable sparse representation. The sparse representation captures the "whole" layer without distinguishing between information recognized by individual neurons. This means that through SAE we can infer that the layer has learned to recognize a list of concepts [*Table*, *Chair*, *Room*, ...]. However, it cannot tell us exactly which individual neuron recognizes each of them and whether they are spatially aligned within individual neurons. Moreover, it typically does not support the identification of complex relationships between these concepts without external analysis tools. As such, SAE differs in scope (layer vs. neuron), objective, and complexity of explanations.

Consequently, these families of methods are not commonly compared against each other, and no established protocols exist for such comparisons. Given these differences, and because our work is theoretical in nature and specifically focuses on guaranteeing the optimality of compositional explanations without altering their established formulation, we leave the

exploration of connections and complementarities with other interpretability paradigms for future research.

Conversely, our work is highly related to Network Dissection and Compositional Explanations. Network Dissection (Bau et al., 2020) measures the spatial alignment between neurons and a set of individual concepts, identifying the concept that maximizes alignment. Since it focuses on individual concepts, exhaustive search is feasible, and the resulting explanations are optimal within that context. Compositional Explanations extend this approach by searching for combinations of concepts that better express a neuron's alignment. However, they are unable to explore the full state space, which sacrifices optimality. Our work closes this gap between Network Dissection and Compositional Explanations by providing a framework that guarantees optimal compositional explanations. Additionally, the development of our beam variant further improves runtime, narrowing the efficiency gap between Network Dissection and Compositional Explanations.

## H. Further Insights and Discussion

**Impact of the Assumptions**    All of the assumptions listed in this paper are implicitly used in previous work in this area of research. In this regard, we do not introduce any new assumptions, nor do we restrict the possibilities compared with the current literature. Among the three assumptions, Assumptions 1 and 2 are the strongest in the literature, and changing them would imply a major shift in how compositional explanations are currently conceived. Assumption 3 concerns only logical operators. The inclusion of operators that are not 00-preserving (e.g., imply) would require adjustments to the algorithm. Indeed, if this assumption is violated, the equivalence between the decomposed IoU and the IoU no longer holds, and the framework may return suboptimal explanations. However, before such modifications, research would need to demonstrate, similarly to what has been done for the operators considered in this paper, that CNNs (or DNNs) can effectively learn the types of relationships expressed by these potential new logic operators. We leave for future work the guarantee of optimality for operators that are not 00-preserving.

**Support for new Logic Operators**    In this paper, we focus on OR, AND, and AND NOT operators, as they are the most commonly used in prior literature by far. These operators provide a high degree of expressiveness, as they can encode relationships that are difficult to capture with other explanation methods. For example, AND NOT captures the concept of "encoded absence", a relation that is rarely addressed in the literature. In the general case, **the algorithm and the decomposition do not require modifications when adding new operators**, as long as they satisfy the assumption of being 00-preserving. Conversely, **the heuristic does require modifications when adding operators**.

Specifically, researchers must encode the impact of each new operator on the identified quantities and provide the maximum and minimum possible improvements for each exclusive path. Depending on the operator, it may also be necessary to provide new maximum and minimum estimations for the combined paths that include the new operator (see the proof for combined paths in Section F.3). Once this information is supplied, the algorithm handles the new operator natively. In some cases, the estimations provided in this paper can be reused to provide a rough estimate that still preserves optimality. For example, for operators that specialize one of the supported operators, the maximum estimations of the quantities represent an overestimation also for their specializations, thus ensuring optimality. The same holds for generalizations of the operators and minimum estimations.

We do not expect the computational complexity of new operators to affect the framework, especially if they are richer and more complex than the ones supported in this paper. Conversely, general and broad operators can have an impact on runtime. For example, AND NOT is very general in practice because most concepts are disjoint, meaning it can potentially be applied to many explanations (e.g., checking whether the neuron is not aligned with a specific color across all colors), and it is the connective that impacts the later stages of the search process the most. In these extreme cases, the implications for optimality guarantees and runtime will need to be taken into account in future work when proposing new operators.

**Extension to New Models and Domains**    The proposed algorithm, heuristics, and quantities are independent of the underlying model or domain, as they operate on binary tensors. As long as compositional explanations can be applied, our framework can be applied as well. In this regard, this work does not introduce any new restrictions on applicability compared to prior work on compositional explanations.

More generally, the component that may require further research to broaden the applicability of compositional explanations in specific settings (e.g., Transformers in vision) is the projection step used to map activations to the same dimensionality as the annotations. In the case of CNNs and images, this is facilitated by the fact that both annotations and activations share similar spatial dimensions $(h, w, c)$, and a simple upscaling is sufficient. Our work adopts the same setup as prior literature

and does not assume any specific projection mechanism. Consequently, if future work introduces new projection strategies to extend compositional explanations to other architectures or domains, our algorithm will be applicable without modification.

**On the Polysemanticity and Superposition of Neurons**   Superposition is the phenomenon whereby neurons may encode multiple relationships or combinations of concepts in their activations. Compositional explanations are among the most effective methods to (partially) capture this behavior. However, the extent to which this is possible depends on the configuration (e.g., maximum explanation length and underlying assumptions). Therefore, it may happen that multiple optimal solutions (associated with exactly the same IoU) exist in the state space, or that multiple solutions achieve similar scores despite being significantly different in terms of the concepts involved. In these scenarios, neurons may encode multiple parallel or more complex relationships that cannot be fully captured under the current configuration.

Possible solutions to address these cases include increasing the maximum explanation length, returning multiple solutions within a given threshold, relaxing assumptions, or supporting more expressive logical operators. Increasing the length or returning multiple solutions introduces a trade-off with interpretability. The other directions require further research. Among these options, our perspective is that capturing such behaviors would likely require both relaxing the incrementality assumption and increasing the length of the explanation, even if this increases the burden for users.

**On the Optimality of Non-Interpretable Activation Ranges**   As mentioned in the main text, applying the optimal algorithm to non-interpretable activation ranges or units (i.e., those associated with an IoU score lower than 0.04) is generally discouraged, since the algorithm will run slower and requiring optimality for a solution deemed non-interpretable is debatable. However, an interesting point is to reflect on what it means when there is no optimal interpretable solution in the state space for a given neuron.

In these cases, we believe we are either in the presence of unspecialized activation ranges (La Rosa et al., 2023), where the activation range acts as a "lack of information" or the inability to observe meaningful alignment may depend on the constraints of the search problem and the hyperparameters used, as discussed in the previous paragraph. In this regard, such activations may encode more complex relationships that cannot be captured by the configuration considered. Addressing these cases would require further research and mitigation strategies similar to those described in the previous paragraph.

**Human Interpretability**   Because the algorithm does not modify the structure of the resulting explanations with respect to the compositional explanations analyzed in previous research and maintains the same hyperparameters (e.g., explanation length), we argue that there are no changes in the ease of interpretation of this kind of explanation. The improvements in explanation quality (i.e., higher IoU) are technical and represent the ground-truth solutions that previous algorithms approximate. In this context, the differences highlighted in the previous section represent an improvement in the faithfulness of this kind of explanation and a reduction in misleading explanations (third category), without imposing any changes on human effort. More broadly, in terms of human interpretability of compositional explanations, the ground-truth solutions provided by this work could be used as a reference by future research to measure approximation errors of alternative structures and formats of compositional explanations and to explore more in depth the trade-off between precision and ease of understanding of compositional explanations, if any.

## I. Examples of Explanations Difference

This section presents examples of the differences between explanations computed by beam search and those obtained with the optimal algorithm. To visualize the alignment, we select the samples in the dataset where the explanation holds and where the neuron is active within the considered activation range. We sort these samples by the size of the intersection between the explanation area and the activation area to ease interpretation, and then visualize the top 4 samples, highlighting the activation captured by the explanation in blue and the activation not captured by the explanation in yellow. The examples are not cherry-picked; rather, they correspond to the first nine differing explanations for the last convolutional layer of each explained model (ResNet18 (He et al., 2016), AlexNet (Krizhevsky et al., 2012), and DenseNet (Huang et al., 2017)). All the models have been pretrained on the Place365 dataset. Note that due to the selection procedure, samples used for visualization of different explanations or different methods for the same unit may differ, and this is expected. Alternative selection procedures, such as randomly extracting samples where the neuron is active, would produce harder-to-understand visualizations due to the superposition of neurons (Elhage et al., 2022; O'Mahony et al., 2023; Dreyer et al., 2024) and variability in the size of activation. We also stress that the visualization is provided only as a reference and contextualization of the highlighted differences in explanations and not as a means of comparison. Despite the good results in the visualization,

as discussed in the main text, the **optimality is not related to visualization properties or interpretability** but is related to the optimal (i.e., highest possible) solution found in a search problem and, specifically in this context, to the property of identifying the specific combination of concepts that captures (or expresses) the highest absolute spatial alignment with the neuron's activations. Finally, note that due to floating-point precision in visualization, in some figures, the optimal and beam solutions may appear to be associated with explanations using different concepts but achieving the same IoU. This is an artifact of the chosen precision. In our experiments, whenever the optimal and beam solutions differ in terms of concepts, they also differ in IoU (Category 1). For a discussion of theoretical cases where multiple optimal solutions exist, please refer to Section H.

Unit:30
Optimal: ((table AND dining_room-s) OR balcony-interior-s) | IoU:0.1001

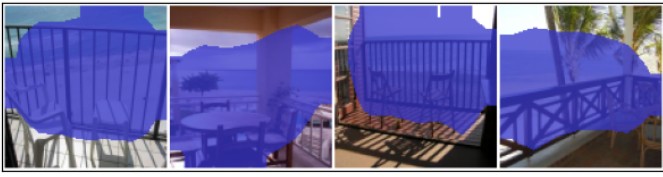

Beam: ((balcony-interior-s OR control_tower-indoor-s) OR dinette-home-s) | IoU:0.0875

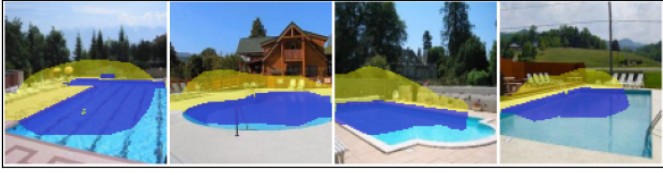

Unit:39
Optimal: ((pillow OR pool table) OR swimming pool) | IoU:0.0466

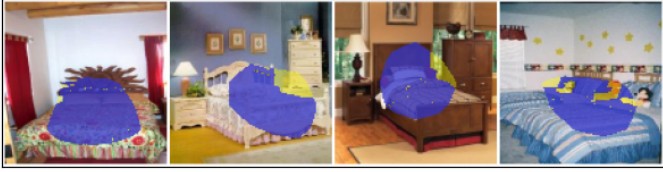

Beam: ((bed AND NOT black-c) OR pillow) | IoU:0.0435

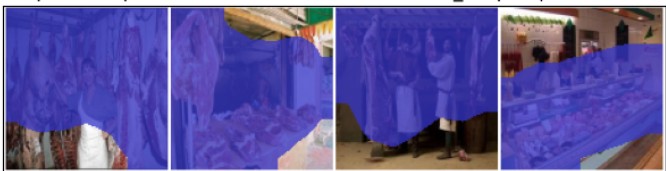

Unit:41
Optimal: ((pink-c AND mountain) OR butchers_shop-s) | IoU:0.0691

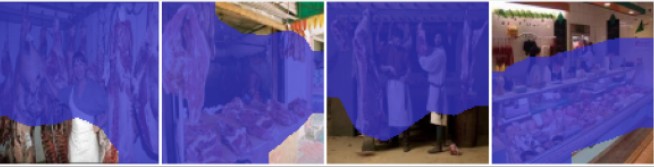

Beam: ((butchers_shop-s OR rubble-s) OR meat) | IoU:0.0687

*Figure 2.* Alignment detected in units of a ResNet18 model by both the optimal and beam-search methods. The blue and the yellow areas indicate the activation captured and not captured by the explanation, respectively.

Unit:45
Optimal: ((house AND NOT garage-outdoor-s) OR roof) | IoU:0.1633

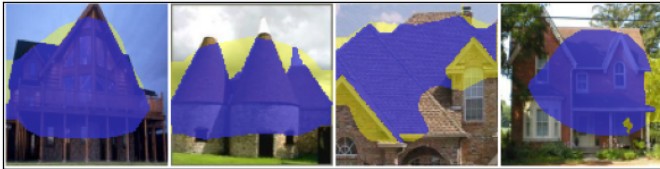

Beam: ((house AND NOT manufactured_home-s) OR roof) | IoU:0.1632

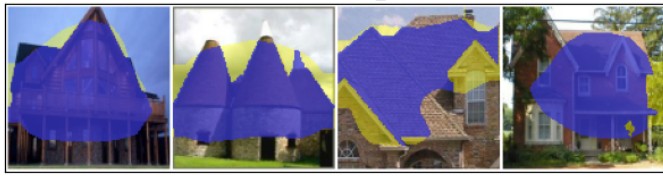

Unit:56
Optimal: ((table OR chair) AND dining_room-s) | IoU:0.0508

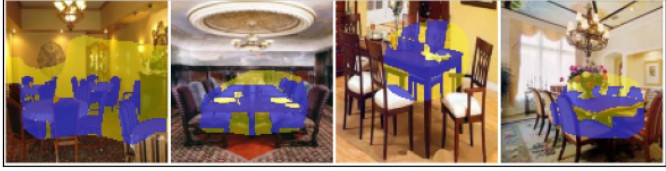

Beam: ((table AND dining_room-s) OR lamp) | IoU:0.0451

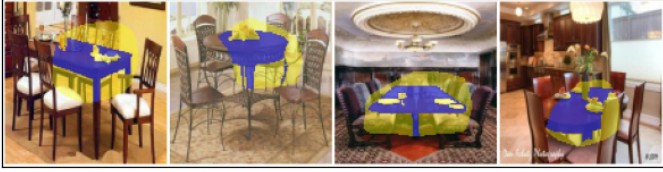

Unit:70
Optimal: ((bed AND NOT brown-c) OR ball_pit-s) | IoU:0.0611

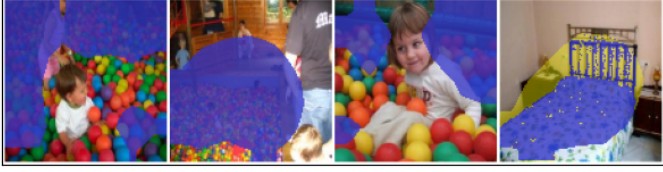

Beam: ((bed OR ball_pit-s) OR pillow) | IoU:0.061

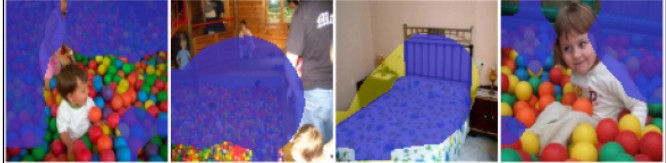

*Figure 3.* Alignment detected in units of a ResNet18 model by both the optimal and beam-search methods. The blue and the yellow areas indicate the activation captured and not captured by the explanation, respectively.

Unit:87
Optimal: ((floor AND corridor-s) OR alley-s) | IoU:0.0908

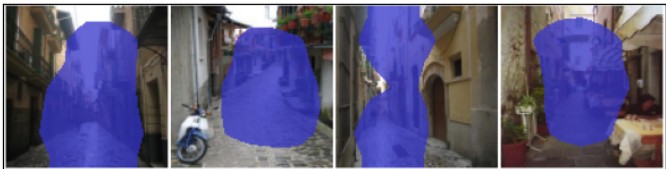

Beam: ((alley-s OR corridor-s) AND NOT wall) | IoU:0.0791

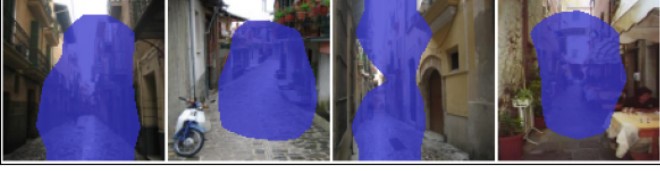

Unit:91
Optimal: ((pasture-s AND NOT sky) OR tent) | IoU:0.0611

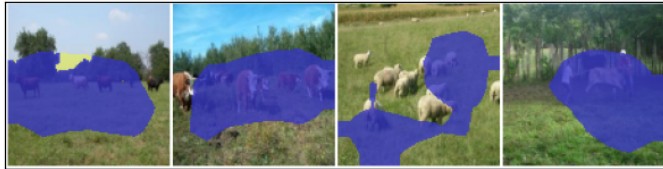

Beam: ((pasture-s OR tent) AND NOT sky) | IoU:0.0611

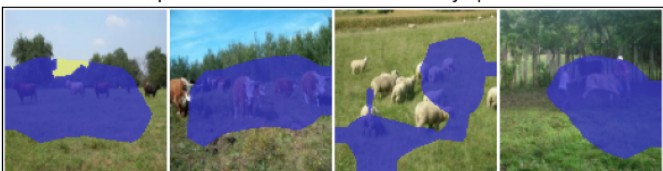

Unit:92
Optimal: ((highway-s AND NOT sky) OR bridge) | IoU:0.044

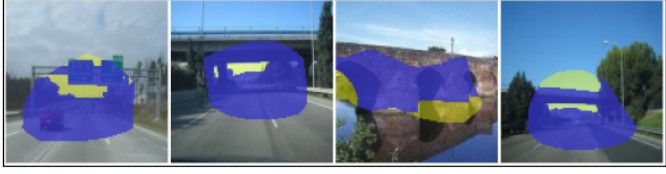

Beam: ((highway-s OR bridge) AND NOT sky) | IoU:0.044

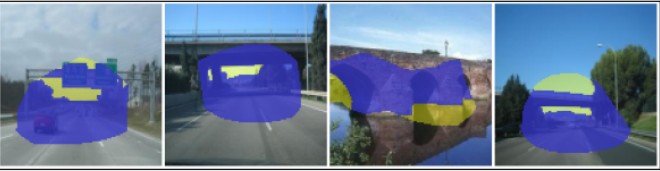

*Figure 4.* Alignment detected in units of a ResNet18 model by both the optimal and beam-search methods. The blue and the yellow areas indicate the activation captured and not captured by the explanation, respectively.

Unit:2
Optimal: ((yellow-c OR airport_terminal-s) AND floor) | IoU:0.0525

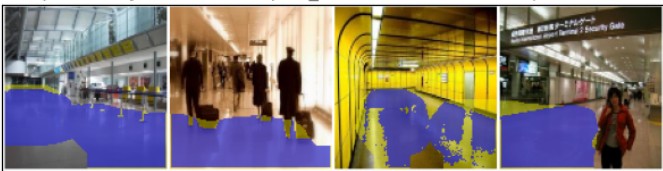

Beam: ((floor AND yellow-c) OR ballroom-s) | IoU:0.0426

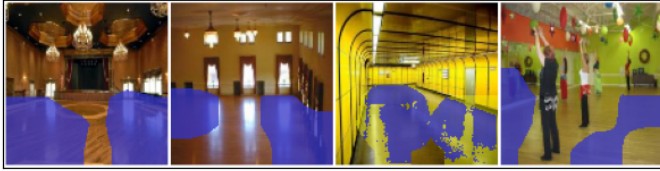

Unit:3
Optimal: ((green-c AND ceiling) OR light) | IoU:0.0529

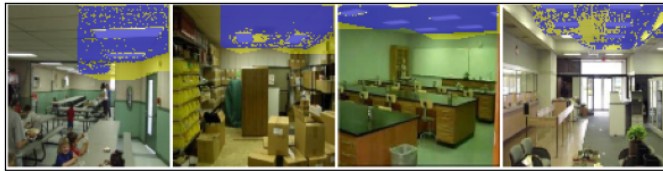

Beam: ((light OR podium-indoor-s) OR fluorescent) | IoU:0.05

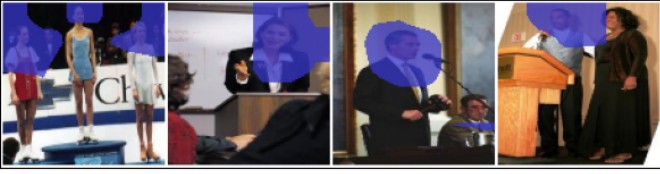

Unit:20
Optimal: ((grey-c OR white-c) AND person) | IoU:0.0293

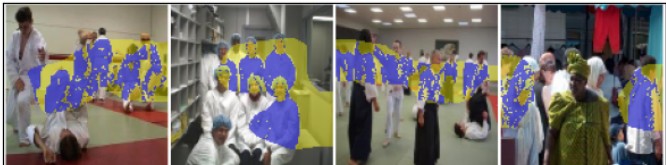

Beam: ((person AND NOT black-c) AND NOT brown-c) | IoU:0.0287

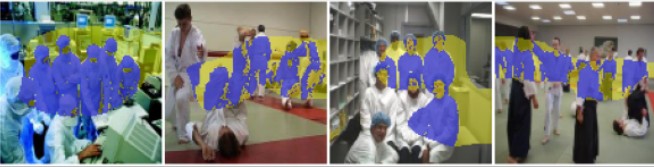

*Figure 5.* Alignment detected in units of an AlexNet model by both the optimal and beam-search methods. The blue and the yellow areas indicate the activation captured and not captured by the explanation, respectively.

Unit:21
Optimal: ((waiting_room-s OR poolroom-home-s) AND floor) | IoU:0.0313

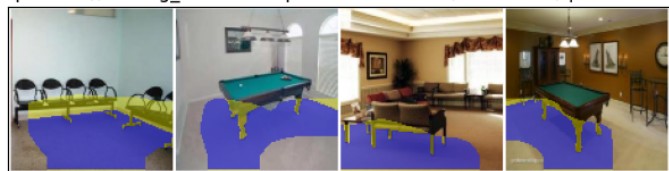

Beam: ((road AND street-s) AND NOT white-c) | IoU:0.0281

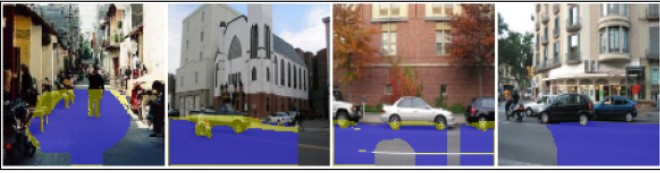

Unit:22
Optimal: ((bedroom-s OR living_room-s) AND ceiling) | IoU:0.0473

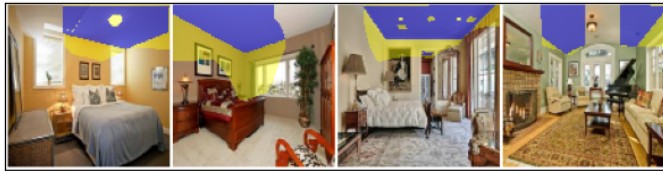

Beam: ((ceiling AND living_room-s) AND NOT black-c) | IoU:0.0406

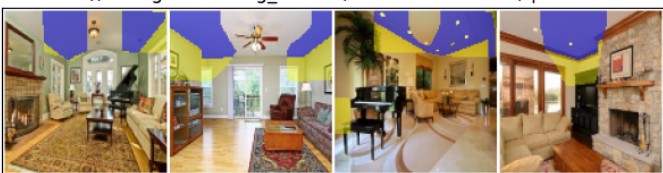

Unit:26
Optimal: ((purple-c AND ceiling) OR pool table) | IoU:0.0377

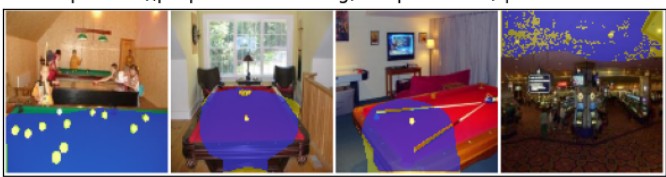

Beam: ((pool table OR ball_pit-s) OR day_care_center-s) | IoU:0.037

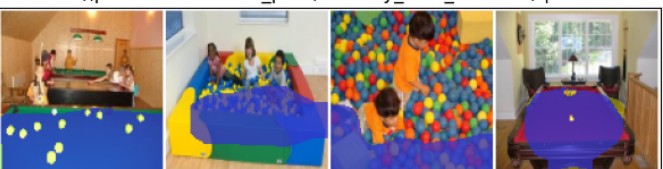

*Figure 6.* Alignment detected in units of an AlexNet model by both the optimal and beam-search methods. The blue and the yellow areas indicate the activation captured and not captured by the explanation, respectively.

Unit:29
Optimal: ((white-c AND person) OR path) | IoU:0.035

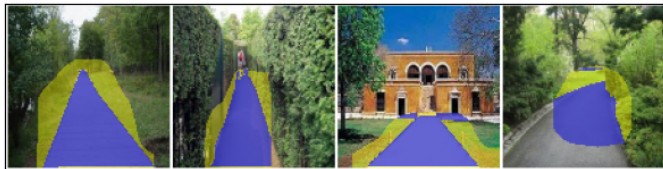

Beam: ((path OR platform) OR forest_road-s) | IoU:0.0314

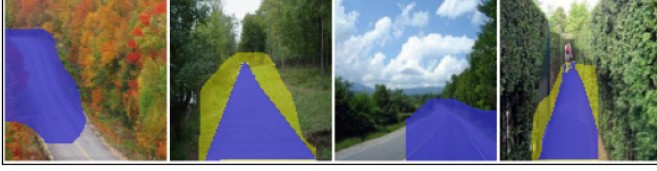

Unit:31
Optimal: ((blue-c AND skyscraper-s) OR skyscraper) | IoU:0.0642

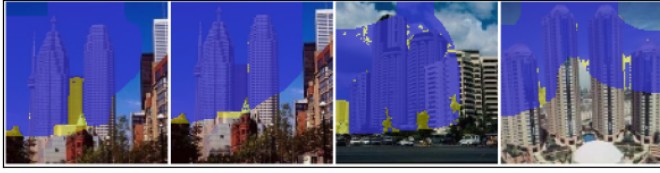

Beam: ((skyscraper AND NOT building_facade-s) OR downtown-s) | IoU:0.0579

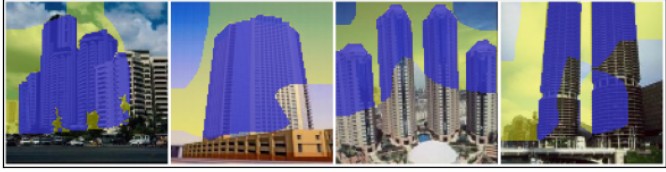

Unit:37
Optimal: ((black-c OR red-c) AND ceiling) | IoU:0.0591

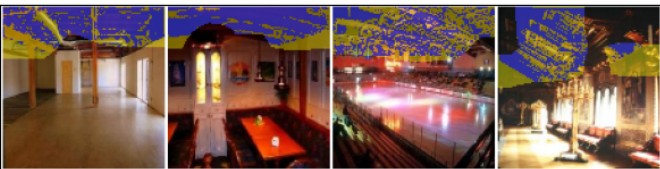

Beam: ((ceiling AND black-c) OR pagoda-s) | IoU:0.0532

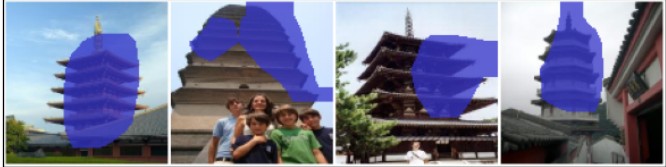

*Figure 7.* Alignment detected in units of an AlexNet model by both the optimal and beam-search methods. The blue and the yellow areas indicate the activation captured and not captured by the explanation, respectively.

Unit:1
Optimal: ((grey-c OR blue-c) AND tree) | IoU:0.0282

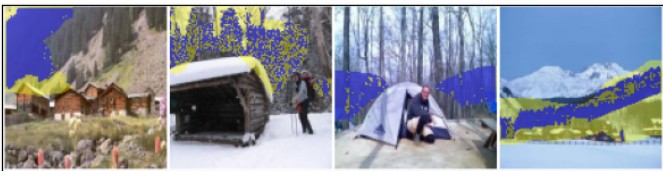

Beam: ((tree AND grey-c) OR ski_resort-s) | IoU:0.0263

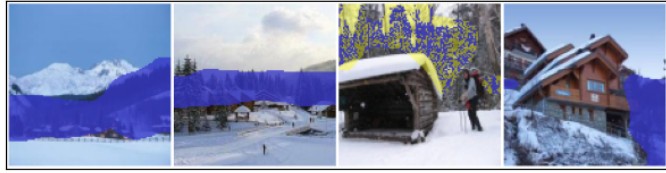

Unit:4
Optimal: ((earth AND batters_box-s) OR tent) | IoU:0.0298

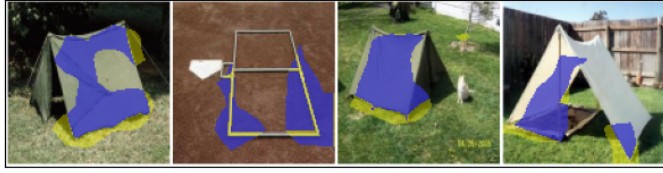

Beam: ((tent OR batters_box-s) AND NOT grass) | IoU:0.0281

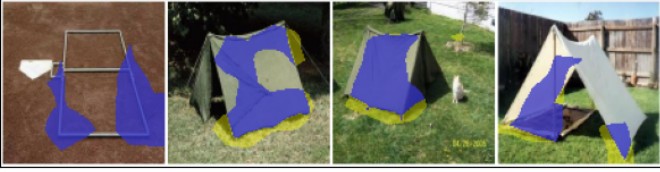

Unit:5
Optimal: ((blue-c OR highway-s) AND mountain) | IoU:0.0386

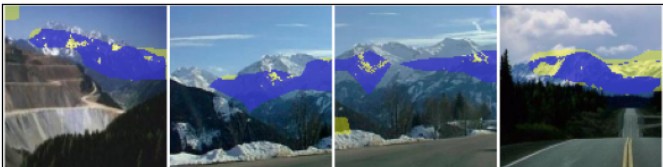

Beam: ((mountain AND blue-c) AND NOT coast-s) | IoU:0.038

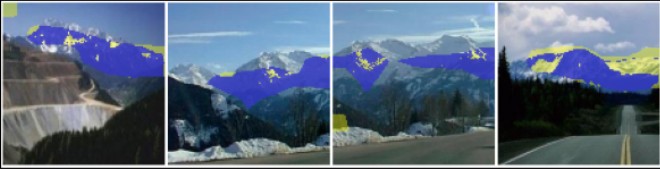

*Figure 8.* Alignment detected in units of a DenseNet161 model by both the optimal and beam-search methods. The blue and the yellow areas indicate the activation captured and not captured by the explanation, respectively.

Unit:7
Optimal: ((black-c OR blue-c) AND floor) | IoU:0.022

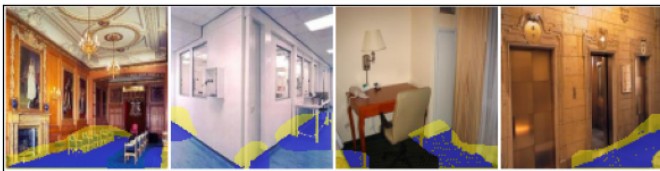

Beam: ((floor AND black-c) OR swimming pool) | IoU:0.0215

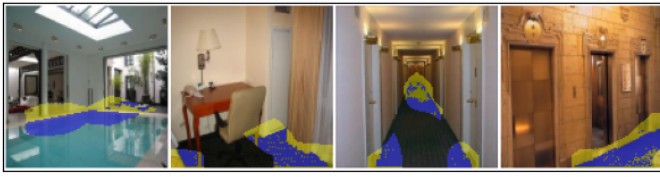

Unit:8
Optimal: ((bedroom-s OR supermarket-s) AND floor) | IoU:0.0125

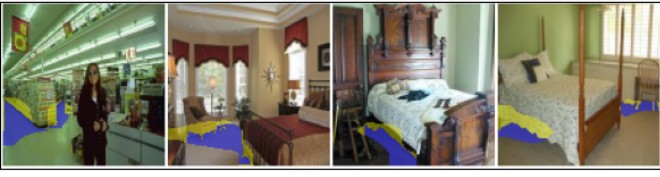

Beam: ((floor AND bedroom-s) OR forest_road-s) | IoU:0.0122

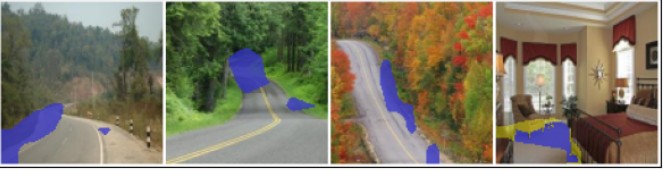

Unit:9
Optimal: ((food OR sconce) OR patty) | IoU:0.0268

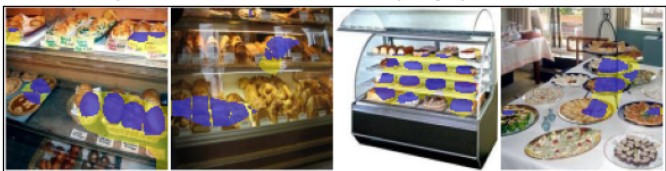

Beam: ((bakery-shop-s OR sconce) OR lighthouse) | IoU:0.0263

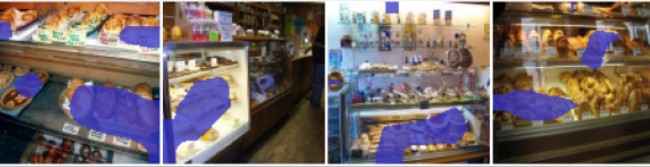

*Figure 9.* Alignment detected in units of a DenseNet161 model by both the optimal and beam-search methods. The blue and the yellow areas indicate the activation captured and not captured by the explanation, respectively.

Unit:15
Optimal: ((harbor-s AND NOT sky) OR hay) | IoU:0.0226

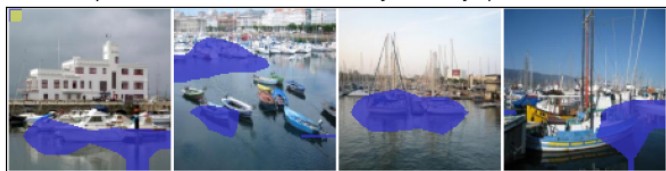

Beam: ((harbor-s OR hay) AND NOT sky) | IoU:0.0226

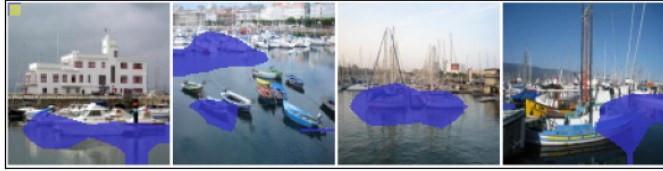

Unit:22
Optimal: ((wall OR ceiling) AND attic-s) | IoU:0.0537

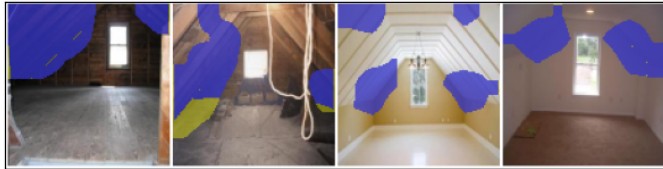

Beam: ((attic-s AND NOT floor) AND NOT bed) | IoU:0.0473

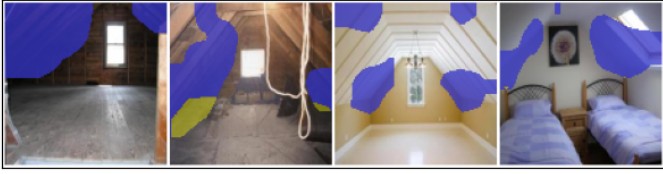

Unit:24
Optimal: ((table AND bedroom-s) OR drawer) | IoU:0.0452

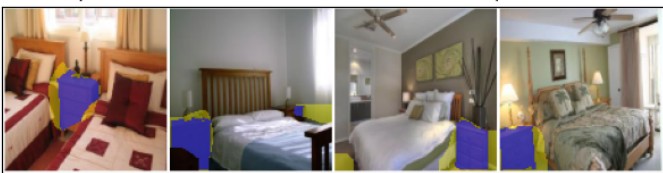

Beam: ((drawer AND NOT grey-c) AND NOT white-c) | IoU:0.0425

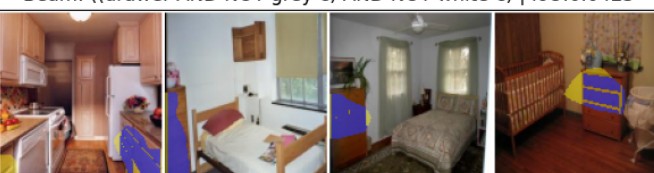

*Figure 10.* Alignment detected in units of a DenseNet161 model by both the optimal and beam-search methods. The blue and the yellow areas indicate the activation captured and not captured by the explanation, respectively.

---

**Algorithm 1** Optimal Algorithm

---

1: **Input:** $\mathfrak{L}^1$, $N$ , $\mathbf{M}$, DisjointMatrix, length
2: Frontier $\leftarrow$ empty priority queue
3: ConceptQuantities, Memory $\leftarrow$ empty lists
4: MinIoU, RecentIoU $\leftarrow 0$
5: **for** $c_{k,i}$ **in** $\mathfrak{L}^1$ **do**
6:     ConceptQuantities[$c_{k,i}$] $\leftarrow$ `compute_quantities`($c_{k,i}$, $\mathbf{M}$, $N$)
7:     Paths $\leftarrow$ `estimate_aggregate_paths`(ConceptQuantities[$c_{k,i}$], length, MinIoU)
8:     Frontier.`add`(Paths)
9:     MinIoU $\leftarrow$ `update_min`(Paths, MinIoU)
10: **end for**
11: Frontier $\leftarrow$ `reduce_frontier`(Frontier,MinIoU)
12: **repeat**
13:     Node $\leftarrow$ Frontier.`pop`()
14:     **if** Node is aggregate_estimation **then**
15:         UpdatedNode $\leftarrow$ `compute_sample_estimate`(Node,MinIoU)
16:         MinIoU $\leftarrow$ `update_min`(UpdatedNode, MinIoU)
17:         **if** UpdatedNode.max_iou > MinIoU **then**
18:             Frontier.`add`(UpdatedNode)
19:         **end if**
20:         **continue**
21:     **end if**
22:     UpdatedNode $\leftarrow$ `apply_logic_equivalences`(Node)
23:     **if** UpdatedNode.max_iou < Node.max_iou **and** UpdatedNode.max_iou > MinIoU **then**
24:         MinIoU $\leftarrow$ `update_min`(UpdatedNode, MinIoU)
25:         Frontier.`add`(UpdatedNode)
26:         **continue**
27:     **end if**
28:     **if** Node.$max\_iou$ == RecentIoU **then**
29:         **if** Node **in** Memory **then**
30:             **continue**
31:         **else**
32:             Memory.`add`(Node )
33:         **end if**
34:     **else**
35:         Memory $\leftarrow$ empty list
36:         RecentIoU $\leftarrow$ Node.$max\_iou$
37:     **end if**
38:     **if** Node is final **then**
39:         TreeQuantities = `compute_tree_quantities`(Node)
40:         Frontier $\leftarrow$ `update_by_tree`(Frontier, TreeQuantities)
41:         IoU = `compute_iou`(TreeQuantities.`get`(Node))
42:         **if** IoU > BestIoU **then**
43:             BestIoU = IoU
44:             BestLabel = Node.label
45:         **end if**
46:     **else**
47:         AdditionalNodes $\leftarrow$ `expand`(Node)
48:         Paths $\leftarrow$ `estimate_aggregate_paths`(AdditionalNodes, Quantities, length,MinIoU)
49:         Frontier.`add`(Paths)
50:         **if** `min`(Paths) > MinIoU **then**
51:             MinIoU $\leftarrow$ `min`(Paths)
52:             Frontier $\leftarrow$ `reduce_frontier`(Frontier, MinIoU)
53:         **end if**
54:     **end if**
55: **until** Frontier is not empty
    **return** *BestLabel, BestIoU*

---

---

**Algorithm 2** Our Informed Beam Search Algorithm

---

1: **Input:** $\mathfrak{L}^1$, $\boldsymbol{N}$, **M**, DisjointMatrix, b, length
2: Beam ← empty list
3: ConceptsQuantities ← empty list
4: **for** $c_{k,i}$ **in** $\mathfrak{L}^1$ **do**
5:     Quantities ← `compute_quantities`($c_{k,i}, \boldsymbol{N}, $**M**)
6:     ConceptsQuantities.`append`(*Quantities*)
7:     IoU ← `compute_dIoU`(*Quantities*)
8:     Beam.`add`(*label = $c_{k,i}$, iou = IoU*)
9: **end for**
10: `sort`(*Beam*)   # Sort by IoU
11: # Select the best b candidates
12: Beam ← Beam[:b]
13: MinIoU ← `find_min`(*Beam*)
14: **for** 2 **to** length **do**
15:     SearchSpace ← `expand_beam`(*Beam*, $\mathfrak{L}^1$)
16:     Estimations ← `estimate_labels_iou`(*SearchSpace*, *ConceptQuantities*, *DisjointMatrix*)
17:     `sort`(*Estimations*)
18:     **for** $L$, EstIoU **in** Estimations **do**
19:       **if** EstIoU $<$ MinIoU **then**
20:         # All the other labels cannot be added to the beam
21:         **break**
22:       **end if**
23:       Iou ← `compute_iou`(L, $\boldsymbol{N}, $**M**)
24:       Beam.`add`(label=$L$, iou=*Iou*)
25:     **end for**
     `sort`(*Beam*)
     # Select the best b candidates
26:     Beam ← Beam[:b]
     # Compute and update info
27:     MinIoU ← `find_min`(*Beam*)
28: **end for**
29: BestLabel, BestIoU ← `max`(*Beam*)

---

