# OpenReview forum: "Guaranteed Optimal Compositional Explanations for Neurons"
_ICML.cc/2026/Conference — ICML 2026 spotlight_

### Official Review · Reviewer_4JCd · 2026-02-27

**Soundness:** 3
**Presentation:** 1
**Significance:** 2
**Originality:** 2
**Overall Recommendation:** 4
**Confidence:** 2

**Summary:**

This paper focusing on the "Compositional explanations" problem, which is aim to explain the neuron activations with propositional logic formulas.
Traditional research [MA20] rely on beam search, with top-K candidates retainment,  to find possible formula that maximize the IoU score.
The key problem is that: beam search can not guarantee optimality of find formula. This is because the traditional IoU measure is a final reward rather than a process reward which can guide the search process.
In this paper, the authors proposed a decomposed version of IoU named dIoU, which is able to guide the process of searching.
The authors based on this measure present heuristic that able to be proved be admissible, like classic A-star algorithm.
The authors proved that their searching algorithm can be guaranteed to find optimal formula that maximize IoU score.

**Compliance With Llm Reviewing Policy:**

Affirmed.

**Final Justification:**

I think the authors have meaningfully clarified the scope of their optimality claim. In particular, they now state more explicitly that the guarantee is with respect to the specific search problem and assumptions considered in the paper. This addresses an important part of my earlier concern about how the claim could be interpreted.

I also think the discussion of applicability is now more appropriately framed as a limitation. The authors acknowledge that broader applicability beyond the current CNN/vision setting depends on progress in compositional explanations more generally, especially the projection/alignment step. I view this point as clarified, though not resolved in a practical sense.

The main issue I still consider insufficiently resolved is the complexity characterization. My original question asked whether the method is polynomial-time, and, if not, whether the corresponding “feasible” claim should be stated more carefully. In the latest reply, the authors state that the method is polynomial-time because the maximum explanation length is constant. I do not think this is the right characterization in the usual sense: fixing explanation length is a restricted setting, and the dependence on explanation length still appears exponential. I therefore think this point should be stated more precisely in the paper.

On the empirical side, the additional clarification improves the interpretation of the existing beam-search comparisons, but my view remains that the empirical support is somewhat narrow relative to the breadth of some of the paper’s claims. I do not regard this as fatal for a theory-leaning paper, but I do think the scope of the evidence should be described carefully.

**Key Questions For Authors:**

Q1. Does this method run in polynomial time? If not, The claimed `This work [xxx] computation of the optimal solution feasible.`  should be clarified with more restricted words.


Q2. This paper builds its algorithm mostly on CNNs and visual data. How can you extend your approach to other models, like ViT, or to non-vision domains?

**Limitations:**

The submission version of this paper does not discuss its limitations.

**Strengths And Weaknesses:**

**Pros:**
+ This paper addresses an important problem in the field of "compositional explanations of neurons." The proposed measure seems straightforward and useful for those working in this area.
+ The algorithm proposed is intuitive and avoids the issues associated with beam search.

**Cons:**
- The organization of this paper is lacking. The introduction can be misleading. In my opinion, the key aspect of this work is the dIoU measure compared to the IoU measure; however, the introduction led me to think that the paper explores a more complex approach to the search space, which is not the case. The propositional logic formula space is the same as that in previous methods; the real change lies in the goal. While the previous IoU served as the final reward, dIoU can act as a progress reward. Furthermore, the organization of the paper is not structured well; Section 2 contains too much information. I suggest that the authors separate related works, preliminary information, and their approach for clearer writing.
- Although I'm not an expert in this field, I find the experiments in this paper somewhat simplistic. The differences between this paper and the comparison methods do not seem significant.
- The paper claims that their method is guaranteed to be optimal, which I find to be an overstatement. The title may easily confuse readers into thinking that the authors have proven this method to be optimal compared to other algorithms based on certain measures, such as convergence rate.

---

> ### Author Rebuttal · Authors · 2026-03-31
>
> > Q1. the optimal solution feasible should be clarified with more restricted words.
>
> A1. The state space is defined in lines 148-150 and corresponds to the worst case scenario for our algorithm. We agree that the definition of feasibility was vague. We will better specify it as *the ability to find the optimal solution in finite time and within the same order of magnitude as previous approaches on the most common settings for compositional explanations*.
>
> > Q2. How can you extend your approach to other models or domains?
>
> A2.The proposed approach is independent of the underlying model or domain as it operates on binary tensors. As long as compositional explanations can be applied, our framework can be applied as well. **This work does not introduce any new restrictions on applicability compared to prior work**.
>
> More in general, **the component that requires research to broaden applicability of compositional explanations is the projection step used to map activations to the same dimensionality as the annotations**. In the case of CNNs and images, this is facilitated by the fact that both annotations and activations share similar spatial dimensions (h, w, c), and a simple upscaling is sufficient. Our work adopts the same setup as prior literature and does not assume any specific projection mechanism. Consequently, if future work introduces new projection strategies to extend compositional explanations to other architectures or domains, our algorithm will be applicable without modification.
>
> > W3A. separate related works, preliminary information, and their approach for clearer writing.
>
> W3A. We will split the section according to your suggestion.
>
> > W3B. The propositional logic formula space is the same as that in previous methods; the change lies in the goal
>
> A3B. While we agree that the dIoU is a key component of our work, the dIoU is a decomposition of the IoU score. Therefore, they are equivalent and there is no change in the objective being optimized. The difference lies in the algorithm, which exploits heuristics derived from the decomposition to navigate the “full” state space. **Regarding the state space, it is not the same** as in previous work. **Beam search explores a *restricted* state space**. By simplifying the structure of the problem (ignoring common terms), the state space explored by previous approaches can be approximated as $b^d$, where $b$ is the beam factor (typically 5) and $d$ is the length of the explanation. Conversely, the state space explored by our algorithm is $n_c^d$, where $n_c$ is the number of concepts in the dataset and $n_c >> b$. Therefore, **our algorithm effectively operates over a much larger state space, as it has access to the full space rather than a restricted subset**.
>
> **The algorithm is different and is novel and more complex compared to the beam search** used in prior approaches. In the case of our beam variant, the algorithm is the same but the heuristic used to estimate the IoU before computing it is different. However, this beam variant represents only a small part of the contribution of this paper. The main contribution is the optimal algorithm, the identified quantities, and the associated heuristics.
>
> > W4. The differences between this paper and the comparison methods do not seem significant.
>
> A4. As noted by other reviewers, and as mentioned in the introduction, **the main contribution of the paper is theoretical**. Finding the optimal solution in this search space has been classified as **infeasible in a reasonable time** in prior literature. **This work makes the navigation of this state space tractable. In this context, the role of the experiments is to validate the theoretical contribution.**
>
> Sect. 3.1 demonstrates that the optimal algo runs in a reasonable amount of time on standard setups. This alone is already a significant result wrt the previous "infeasibility". We are not aiming to improve runtime, as this would likely be impossible given that the explored state space is significantly larger than the one considered by previous methods.
>
> Moreover, **the finding that 10-40% of previously reported explanations are not optimal is a non-trivial finding that was not previously known and could not be established without our work**.
>
> > W5. The paper claims that their method is guaranteed to be optimal, which I find to be an overstatement.
>
> A.5 We respectfully stress the fact that **the claim of optimality is not an overstatement. The mathematical proof of optimality of the algorithm is provided in Appx. F**. As clarified in Sect. 2, the optimality refers to the context of a search problem. The objective function (and the corresponding score) is explicitly defined in Eq. 1 (lines 127-130). Under this formulation, the algorithm and the associated heuristics are guaranteed to identify the optimal solution (i.e., the one achieving the highest IoU) within the state space defined by the assumptions adopted in prior work and described in Sect. 2.

---

> > ### Author Rebuttal · Reviewer_4JCd · 2026-04-01
> >
> > Thank you for the detailed rebuttal. Some of the clarifications are helpful, especially the narrower explanation of what notion of “optimality” is intended. However, I remain not fully convinced, mainly because several of my concerns seem to have been addressed only after being reformulated into narrower questions, rather than in the original context in which they were raised.
> >
> > On feasibility, my concern was not only that the wording was vague. I explicitly asked whether the method is polynomial-time, and if not, whether the corresponding claim should be weakened. The rebuttal mainly addresses the wording issue, but it still does not directly answer the complexity question itself. So while I appreciate the proposed revision, I do not think this fully resolves the concern.
> >
> > Similarly, on “guaranteed optimality,” my point was not simply whether a proof exists under the paper’s formal setup. My concern was also about how this claim is framed in the title and introduction, since it may be read more broadly than what is actually proven. The rebuttal clarifies that the guarantee is with respect to the specific search problem and assumptions defined in the paper, which is useful, but I still think the current framing risks overstating the scope of the result unless this limitation is made much more explicit.
> >
> > Regarding applicability beyond CNN-based vision settings, the response is again only partially reassuring. The argument that the framework applies to binary tensors in general is reasonable at a high level, but the key practical difficulty is precisely the projection/alignment step, which the rebuttal acknowledges remains future work. So I think the limitation still stands in substance, even if the theoretical framework itself may be broader in principle. Reviewer `TEx9` appears to have raised a closely related concern.
> >
> > Finally, on the empirical side, I still find the current evidence somewhat limited relative to the strength of the paper’s claims. I understand the authors’ point that this is mainly a theoretical contribution, and the observation that prior beam-search explanations can be suboptimal is certainly interesting. Still, this does not fully address my concern that the experimental validation remains relatively narrow. In this respect, I note that other reviewers also raised concerns about empirical scope or missing quantitative comparisons.
> >
> > Overall, I appreciate the rebuttal, and I do think it improves the paper's interpretation. But I also remain somewhat unsatisfied, because several of my original concerns are, in my view, only partially addressed and would still require clearer qualification in the paper itself.

---

> > > ### Author Response · Authors · 2026-04-01
> > >
> > > Thank you for your contribution to improving our work and for your detailed feedback. We are grateful for your comments and believe that the suggested refinements in terminology and clarity will strengthen the paper. We would also like to clarify that the information provided in our previous answers will be incorporated into the paper. Unfortunately, due to the limited space available during the rebuttal phase, this may not have been sufficiently clear. Below, we address the remaining concerns.
> > >
> > > We clarify that:
> > > - The method runs in polynomial time (the maximum length is constant). We will explicitly state this in the introduction to better inform the reader.
> > > - We will explicitly specify that the optimality guarantees are defined with respect to the particular search problem and assumptions considered, both in the abstract and in the introduction, as suggested by the reviewer. This, together with the previously proposed feasibility wording, should help align better the narrative of the introduction with the experimental and theoretical contribution.
> > >
> > > Regarding applicability, we acknowledge the reviewer’s concern. At the same time, we note that the paper focuses on resolving a specific limitation (the infeasibility of optimality), while the broader applicability of compositional explanations is an open problem in the literature. We believe addressing both aspects simultaneously would go beyond the scope of a single paper, but we appreciate the reviewer’s perspective and will clarify this limitation more explicitly in the paper.
> > >
> > > Finally, on the empirical side, we believe the concerns have been addressed during the rebuttal. In particular:
> > > - All beam-based variants find the same solution, and therefore produce identical IoU values (reported in Table 6). We will move this clarification to the beginning of the section to make it more explicit.
> > > - The quantitative comparison in terms of IoU between the optimal algorithm and beam search, across all units and three different networks, is reported in Table 6. We will replace Table 2 with Table 6 in the main text to improve clarity.
> > > - The quantitative comparison in terms of visited, expanded, and estimated states, as well as runtime over 3 different networks, is reported in Table 1 and in the additional results provided during the rebuttal (https://tinyurl.com/2s4es9a2). We will complete these experiments and include the results in the appendix.
> > >
> > > Overall, we believe that all key quantitative comparisons are covered and that the evaluation settings are consistent with those adopted in prior work (e.g., La Rosa et al., 2023; Massidda et al., 2022; Hard et al., 2022; Mu et al., 2020).

---

### Official Review · Reviewer_QvXr · 2026-03-05

**Soundness:** 4
**Presentation:** 3
**Significance:** 3
**Originality:** 3
**Overall Recommendation:** 5
**Confidence:** 3

**Summary:**

This work focuses on compositional explanations, a class of post-hoc interpretability techniques that align a given neuron's activations with combinations of high-level concepts (compositional formulas). In practice, exploring the entire combinatorial space to find the optimal alignment (measured via the IoU metric) is not feasible. As such, researchers typically rely on incremental derivation and beam search to find good alignments.

In this work, the authors show that by adding a weak assumption (00-preservation) that is often already implicitly made in the literature, the IoU can be decomposed in a way that allows finer analysis of the important factors influencing alignment. They then produce a heuristic that leverages aggregated computation to estimate alignment during search, and propose an algorithm that can produce a probably optimal alignment in a tractable amount of time. Finally, they use their modified heuristic to produce a modified beam search.

The authors find that their optimal algorithm manages to find a solution at a reasonable cost, and give a qualitative and quantitative analysis of the situations in which beam search produces suboptimal results. They also find that their heuristic-guided beam search is competitive with existing beam search methods.

**Compliance With Llm Reviewing Policy:**

Affirmed.

**Final Justification:**

I do not have any major concerns with the paper, either before or after rebuttal, and am maintaining my initial score as this is a convincing and timely paper.

**Key Questions For Authors:**

- As mentioned above, the paper does not report the IoU obtained by the heuristic-guided variant, nor does it compare it to that obtained by other beam search methods. Can the authors comment on whether they have ran such experiments, and whether the heuristic beam search is competitive with existing search methods?
- Can the authors comment on whether there is a difference in IoU change between the three categories identified as failure modes?
- Is choosing the top 0.005 percentile activations a common choice in the field? Do the authors have any intuition or evidence as to how changing that value may change the obtained results? Would using a fixed threshold change things significatively? As this is, as far as I can tell, the main hyperparameter of the method, it seems that exploring the method's sensitivity to it is rather important.
- In Table 6, what is the IoU score reported? Is it the average over all 50 units? If so, both AlexNet and DenseNet produce averages close to 0.04, which according to the main body, means the neurons are generally not interpretable/not specialized. I think this is worth discussing in the paper. If most of the neurons are not interpretable with a provably optimal alignment, what are the implications? Is neuron interpretability simply "doomed" for that kind of model?
- Similarly, I have noticed in the Appendix that for certain units (e.g. 41, 45, 70), the optimal alignment produces a score that is exactly equal to the beam search one. However, the explanations are almost completely different (category 1 error). What makes the optimal explanation "better" than the other one in that case? More generally, the fact that multiple different explanations may obtain the same optimal scores seems to be a case of "non-identifiability" as has been identified in recent XAI literature (arxiv.org/abs/2512.18792). This makes sensitivity analysis even more important in this case, since if the "true" explanation is not identifiable, that means that one should observe high variance to hyperparameters or datasets. Have the authors noticed any such phenomenon?

**Limitations:**

yes

**Strengths And Weaknesses:**

# Soundness

This is, overall, a strong paper. The research questions are well-defined (how to find optimal alignment in a reasonable time, and what are the failure modes of beam search methods?) and motivated, and the authors answer them satisfactorily. The authors' framework allows them to produce two practical algorithms that can be directly used by practitioners:
- A provably optimal search method that runs in tractable time (a rather rare instance in this type of problem), which they then validate through experiments.
- An additional fast version using beam search and which is competitive with existing methods in terms of execution time.
In addition, the authors produce an interesting and detailed insight on the failure modes of beam search, supported by practical examples.

In terms of the theoretical framework, the paper is very detailed and claims seem to be well-supported. One important point here is that while the high-level goal of the analysis is clear, I was not able to fully verify every step of the proofs, as the paper contains several pages of theoretical derivations.

As far as I can tell, the datasets and experimental setups are consistent with the ones used in similar studies (e.g., the ones mentioned in the related work).

One aspect that is missing from the paper is the evaluation of the author's beam search variant in terms of performance (IoU) compared to the MMESH and vanilla baselines. The authors' variant is shown to be competitive with the others in terms of computational resources (Table 1: execution time, visited nodes), but it is unclear how that translates in terms of final score. This is important, as if the obtained IoU scores are much lower than those of the other baselines, it makes the heuristic significantly less useful in practice. While the beam search variant's performance is compared to the optimal algorithm (if I am right in assuming that this is what is depicted in Table 2), the paper is nonetheless missing a comparison of the IoU scores of all other beam search methods. Similarly, it would be interesting if there is a difference in IoU change between the three categories identified as failure modes.

# Presentation

The structure of the overall paper is logical. The appendix is very long, contains lots of details on additional proofs as well as expansive details on the algorithm and additional performance optimization techniques, which makes the authors' work reproducible even before the code is released (which the authors mention they will do upon acceptance). This is a sign that the author's algorithm was clearly designed with practical usage in mind.

However, the presentation aspect is where I believe the paper could use the most improvements.

I found the length of the theoretical framework to slightly detract from the overall contribution. While this is a major contribution of the work, the paper introduces a large number of definitions which are not immediately easy to grasp. It also seems to me that the derivations on page 5 could be moved to the appendix. In contrast, replacing Table 1 with its full appendix version (Table 4) and especially bringing Table 3 (identified quantities) to the main body of the paper may make the proof easier to follow. In general, the main body of the paper could use one or two more visualization figures to clarify the intuition behind the proof.

There are several additional clarity, syntax and notation issues:
- l.104: "one" should be "ones"
- 2.1: There is some notational confusion about $d$. It is initially defined as the size of inputs in the dataset (l.82, correct), then as the dimension of the activations (l.84, wrong; those are $< d$), then the text states that "$d$ is obtained by upscaling [the neuron's] bidimensional feature map" (l.108, clumsy: it is not $d$ that is obtained). It would be clearer to say that the inputs have dimension $d$, the activations/feature maps have dimension $< d$, and those are then upsampled to produce data of dimension $d$ again.
- l.115: It may be common knowledge within the field, but it would be nice to have a very short mention of typical choices for the values of $\tau_1$ and $\tau_2$. The authors briefly mentions that the authors use the top 0.005 percentile activations as the activation range, but this is buried in the appendix and it is unclear if that is standard practice and/or how other choices may affect the results.
- l.117: $\mathcal{L}^1$ is defined as the set of concepts, and then $\mathcal{L}^n$ as the set of formulas of concepts. This is acceptable (although a slight abuse of notations), but later on, $L$ is defined as a formula of $\mathcal{L}^n$ (l.120) but also as a "label" (l.125) which is confusing. It might help to clarify that in this case, the label is simply the formula.
- l.138-140: The definition of $\theta$ seems rather unclear to me (and $\theta$ a rather unintuitive choice of notation). If it is simply the function that applies the formula $L$ element-wise to the appropriate matrices of $\mathbf{M}$, then something like $\circ_L(\mathbf{M})$ seems clearer. This is also a point where the $M_k$ notation could be re-used.
- l.145: Please consider indexing $⊕_1$, $⊕_2$ etc. (as now, the reader may believe these are all the same operator)
Table 1: The column label "u/s" reads as "units per second", but the description seems to imply it should rather be "s/u" or simply "t/u (s)"
Tables 2 and 6: I cannot tell whether "Percentage of changed explanations of explanations" (in the caption) is a proofreading error. If it is intended, it is not understandable. "Diff" is also not explicit, although it seems to refer to "differ" in the text. It seems that the table displays the percentage of non-optimal explanations (Diff) found by beam search, and then how those non-optimal explanations can be categorized, but it takes multiple readings to understand that. A good change would be simply renaming the column to "Non-optimal". For Table 6, the caption should also indicate how the reported IoU score is aggregated (average over the 50 units? median?). Finally, it is not clear which beam search method is used in those results. I am assuming that it is the one that relies on the authors' heuristic?
- l.362: The reference should probably be to Table 2 rather than 6.
- l.420-422: The sentence "Among them..." is rather hard to read, consider adding a comma and an "of" between the prepositions.

# Significance and originality

As far as I can tell, compositional explanations of neurons is a rather narrow subfield of research. However, this paper seems like a major contribution in that field, as it essentially solves the optimal alignment problem without adding strong constraints and while opening up avenues for future work (additional operators, etc.). The fact that the algorithm is immediately usable and practical is a major asset of the work.

The work is rather original as it not only solves the optimal alignment problem in tractable time, but also offers insight into the failure modes of existing methods (beam search).

I recommend acceptance of this work, conditional on the fact that the authors (i) give quantitative performance (IoU-wise) of their beam search variant compared to the vanilla/MMESH methods (ii) address the presentation issues mentioned above.

---

> ### Author Rebuttal · Authors · 2026-03-31
>
> > Q1.  IoU obtained by the heuristic-guided variant
>
> A1. We specified the motivation in lines 380-383. In a few words, **MMESH, vanilla, and our beam variant all find the same solution, since all of them have access to the same state space (i.e., the one reachable by beam search)**. Beam search builds the search tree one level at a time. Because both MMESH and our beam variant use admissible heuristics, they are guaranteed to identify, at each level, the same best  *𝑏* candidates as the exhaustive search (vanilla) within the restricted search space, as **formally proven in La Rosa (2023) and in Appx. F**. Therefore, the found solution is guaranteed to be the same. Since the values are the same, we did not report them in the paper. We will add this explanation to the paper.
>
> > Q2. difference in IoU between the 3 categories
>
> A2. The categories that involve differences in IoU are Cat 1 and Cat 2. The differences are reported in Table 3 at the following link: https://tinyurl.com/2s4es9a2 . These results are in line with Sec. 3.2. Cat 2 includes explanations that share the same concepts but are chained in different ways. For instance, in the example discussed in ln 381-284, most of the sinks in the dataset are white. Therefore, most of the behavior related to *sink* is already captured by the beam explanation, resulting in a smaller difference in IoU. Conversely, Cat 1 includes explanations that differ in the concepts included in the explanation, meaning that the optimal algorithm finds solutions representing relationships that are not reachable by beam search, thus resulting in a larger gap in IoU.
>
> > Q3. In Table 6, what is the IoU score? If most of the neurons are not interpretable, what are the implications?
>
> A.3. The IoU score in Table 6 is computed as the **average IoU over all the units in the layer for which the optimal and the beam search find two different solutions**. The reviewer’s observation is indeed interesting. Based on evidence from previous literature, we believe the implication is that the impossibility to observe meaningful alignment for those specific neurons depends on the constraints of the search problem and the hyperparameters used.
>
> In this regard, **neurons with low IoU may have learned more complex relationships that cannot be captured by the given configuration considered**. Addressing this limitation would require either increasing the maximum length of the explanation, relaxing the assumptions, or supporting more expressive logical formalisms. Increasing the maximum length introduces a trade-off with human interpretability, since longer explanations tend to be harder to understand. The other two directions require further research in compositional explanations.
>
> > Q4. About the choice of top 0.005 percentile
>
> A4. **The top 0.005 quantile is the standard choice in all prior work** on compositional explanations. Each neuron has its own activation distribution. Fixing a global threshold across all neurons would result in some neurons having an extremely large (or small) number of activations, causing the $\mathbf{N}$ Matrix to converge to all ones or all zeros, respectively. To the best of our knowledge, the only work that diverges from this choice is La Rosa et al. (2023), which proposes clustering activations and then applying the algorithm to each cluster. In that case, the only difference lies in the values of the  $\mathbf{N}$ (computed over a different activation range), but **there are no fundamental changes that would affect the optimal algorithm**.  As a side note, based on preliminary results on cluster 4 (DenseNet model), we already observe differences between the optimal and beam solutions for a subset of units, confirming that the lack of optimality does not depend on the specific activation range. We will complete these experiments and report them in the camera-ready version.
>
> > Q5. for certain units (e.g. 41, 45, 70), the optimal alignment produces the same score but the explanations are different
>
> A5. In these cases, **the issue comes from the decimal precision used in the visualization** of the scores. When considering higher precision, the scores are not identical. For example, for unit 41 they are 0.06867 for MMESH and 0.06905 for the optimal solution. We will adjust and increase the precision in the camera-ready version.
>
> In all the experiments we ran, we did not observe cases where the IoU is the same while the concepts completely differ. If this would happen, it may indicate that the given configuration is insufficient to capture the full complexity of the neuron and the same mitigation strategies discussed in A3 could be applied. Another alternative would be to return all the optimal solutions instead of a single one, although this may increase the burden on the user.
>
> > W5. Presentation issues
>
> A5. We will fix the notation according to your suggestions and move some information from the Appx., thanks to the additional page available post-rebuttal.

---

> > ### Author Rebuttal · Reviewer_QvXr · 2026-04-02
> >
> > I thank the authors for their detailed rebuttal and clarifications. As my concerns were minor in the first place, and the authors have substantively and satisfactorily addressed them, I maintain my positive assessment.

---

### Official Review · Reviewer_TEx9 · 2026-03-12

**Soundness:** 3
**Presentation:** 2
**Significance:** 2
**Originality:** 2
**Overall Recommendation:** 4
**Confidence:** 4

**Summary:**

This paper studied the composition explanation problem. A new framework is proposed to decompose IoU into finer quantities. Using an admissible heuristic from that decomposition, it then uses a best-first search algorithm to compute a guaranteed optimal composition explanation. Empirically, the proposed framework finds that 10-40% of previously obtained explanations with beam search are suboptimal.

**Compliance With Llm Reviewing Policy:**

Affirmed.

**Final Justification:**

The authors have addressed my main points in their rebuttal. I suggest integrating the new discussion and results directly into the main text.

**Key Questions For Authors:**

Please refer to the weaknesses.

**Limitations:**

Yes

**Strengths And Weaknesses:**

**Strengths**
+ The studied problem is clearly defined.
+ Theoretical justifications are provided for the proposed framework.

**Weaknesses**
+ The propped framework is limited to CNN-based models. The practical impact may be limited.
+ In Table 1, for the high-complex setting, the proposed method takes 5,768 seconds per unit, which is still very expensive.
+ Currently, we have new tools like Saprse Autoencoder [1] and CLIP-disset [2] to understand the neurons. What’s the advantage of composition explanation compared to the new tools?
+ The empirical evaluation section appears to be weak. The main experiments use 50 random units from one layer of a ResNet18 trained on Places365.
+ The overall contribution is somewhat incremental. It improves the search procedure for an existing explanation formalism without introducing new concepts or ideas.

References:

[1] Cunningham, Hoagy, et al. "Sparse autoencoders find highly interpretable features in language models." arXiv preprint arXiv:2309.08600 (2023).

[2] Oikarinen, Tuomas, and Tsui-Wei Weng. "Clip-dissect: Automatic description of neuron representations in deep vision networks." arXiv preprint arXiv:2204.10965 (2022).

---

> ### Author Rebuttal · Authors · 2026-03-31
>
> > W1. In Table 1, for the high-complex setting, the method takes 5,768 seconds, which is expensive
>
> A1. We agree that the “absolute” value may appear expensive. However, **this number should be interpreted in the context of the compositional explanations. Finding the optimal solution in this search space has been classified as infeasible in practice in a reasonable time** in prior literature (Mu et al., 2020 and all related work). Our goal is to demonstrate *feasibility in finite time*. In this context, feasibility should be understood as *the ability to find the optimal solution in finite time and within the same order of magnitude as previous approaches on the most common settings used in the literature for compositional explanations* (we will clarify this definition in the introduction).
>
> In table 1, we are not aiming to improve runtime, especially compared to beam-based algorithms guided by heuristics, as this would likely be impossible given that the explored state space is significantly larger (by orders of magnitude) than the one considered by previous methods. Conversely, **the fact that an algorithm guaranteeing optimality achieves runtime comparable to the vanilla beam search proposed by Mu et al., 2020 represents an important milestone for this research area**.
>
> > W2. What’s the advantage of composition explanation compared to Sparse Autoencoder [1] and CLIP-disset [2]?
>
> A2. This topic is discussed extensively in Appx. G. In short, **compositional explanations allow us to capture complex logical relationships between multiple concepts learned by individual neurons**. In the case of images, they also capture the spatial alignment between these relationships and a given activation range. Both of these properties cannot be captured by other families of methods, including those mentioned by the reviewer.
>
> More generally, these approaches have historically been considered complementary rather than competing. For example, ClipDissect was proposed after compositional explanations and, appropriately in our opinion, was not compared against them but rather against NetDissect. This is because **ClipDissect assigns to each neuron the single concept** that maximizes correlation, whereas compositional explanations capture the alignment between logical combinations of concepts rather than a single concept. Similarly, **sparse autoencoders** focus on capturing features learned at the level of **entire layers** rather than individual neurons. For this reason, they are not considered an alternative to compositional explanations, but rather a complementary approach. For a more broad discussion, please refer to Appx. G.
>
> > W3. The empirical evaluation section appears to be weak. The main experiments use 50 random units from one layer of a ResNet18 trained on Places365.
>
> A3. Note that **Table 6 includes experiments over the full set of neurons across 3 different networks**, demonstrating the feasibility of the optimal approach (see A1). Nevertheless, to better support our beam variant, **we ran the same experiments to record timing for the other two models and included the results in** https://tinyurl.com/2s4es9a2 (Table 1 and 2). In some cases, the slowest configurations were computed over a reduced set of neurons (or none) due to the limited rebuttal time. We commit to completing these experiments and including them in the camera-ready version. These settings are consistent with prior work on compositional explanations.
>
> > W4. The overall contribution is somewhat incremental. It improves the search procedure for an existing explanation formalism without introducing new concepts or ideas.
>
> A4. We respectfully do not agree with the reviewer’s assessment. As also noted by other reviewers, we believe this work constitutes a **strong theoretical contribution** to the area of compositional explanations.
> Specifically, we:
> - introduce **new formalism by defining new quantities** in Section 2.2 and proposing a novel decomposition of the IoU score. These quantities, the decomposition, and the associated proof (Appx. A) were not available in prior work and represent new conceptual contributions and formalisms. As discussed in the conclusions, these ideas operate on binary data and may be **applicable beyond compositional explanations** (e.g., semantic segmentation, where IoU is commonly used).
> - introduce **novel heuristics** that do not rely on spatial information (unlike MMESH). These heuristics are designed from scratch rather than being incremental modifications of existing ones.
> introduce an **entirely new algorithm for compositional explanations**, which differs fundamentally from the beam search paradigm used in prior work.
> - **provide empirical evidence that previously reported explanations are not optimal** and identify specific failure modes of beam search in this context (e.g., “unverified” explanations discussed in Section 3.2).
>
> > W5. Limited applicability
>
> Please refer to A2 answer to Reviewer 4JCd

---

> > ### Author Rebuttal · Reviewer_TEx9 · 2026-04-03
> >
> > Thank you for the detailed rebuttal. Some of my concerns have been addressed. I have several follow-up questions.
> >
> > *For W2*:
> > + Do you want to argue “not comparable,” or “complementary but uniquely capable”? The latter is usually stronger.
> > + Can you give one concrete example of an explanation that compositional explanations can express but SAE / CLIP-Dissect cannot?
> >
> > *For W4*:
> > + Can you rank the contributions instead of listing them equally? Right now, it reads a bit diffuse.

---

> > > ### Author Response · Authors · 2026-04-04
> > >
> > > > Can you give one concrete example of an explanation that compositional explanations can express but SAE / CLIP-Dissect cannot?
> > >
> > > Consider Fig 1 (paper). This explanation expresses the fact that the regions of activation within images for neuron 56 match (i.e., are aligned with) the regions of the images corresponding to tables or chairs in a dining room, which correspond to the activation area highlighted in blue. This type of explanation combines three properties: (i) spatial alignment, (ii) logical composition of concepts, and (iii) neuron-level specificity. These properties cannot be simultaneosly expressed by CLIP-Dissect or SAE.
> > >
> > > For each sample, CLIP-Dissect extracts a single value (e.g., the max over the whole image). Therefore, for a dataset of 𝑁 samples, CLIP-Dissect computes a vector of N values representing those activations. It then assigns to the neuron the single concept whose representation is most correlated with this vector. In this case, it selects the concept ‘Inn indoor scene’, which refers to the whole image. As such, there are 3 fundamental differences. First, CLIP does not capture spatial alignment between regions, since it abstracts the encoding of an image into a single scalar. Second, the objective is different, and therefore the methods answer different questions. Indeed, correlation does not imply alignment, since the “value” representing the sample is not constrained to be within any specific region. Third, CLIP-Dissect assigns a single concept to each neuron. We could potentially extract the top k correlated concepts (for neuron 56, the top 3 would include “hunting lodge indoor scene” and “hotel breakfast area scene”). However, it would be unclear how these concepts relate to each other (if any relationship exists). In summary, there is a difference in objective, explanation specificity, and explanation complexity between the two methods.
> > >
> > > SAEs aim to find a projection that maps the layer representation into a more interpretable sparse representation. The sparse representation captures the “whole” layer without distinguishing between information recognized by individual neurons. This means that through SAE we can infer that the layer has learned to recognize a list of concepts [Table, Chair, Room, … ]. However, it cannot tell us exactly which individual neuron recognizes each of them and whether they are spatially aligned within individual neurons. Moreover, it typically does not support the identification of complex relationships between these concepts without external analysis tools. As such, SAE differs in scope (layer vs. neuron), objective, and complexity of explanations.
> > >
> > > >  “not comparable,” or “complementary but uniquely capable”?
> > >
> > > Both perspectives apply. However, we frame them as “complementary but uniquely capable”. Each method captures aspects that others cannot. For example, compositional explanations cannot capture global correlations between activations and concepts as CLIP-Dissect does, and they do not summarize layer-wide representations as SAE does. Conversely, CLIP-Dissect and SAE cannot capture logical (and complex) relationships with spatial grounding at the neuron level.
> > >
> > > This complementarity also explains why these methods are difficult to compare under a single unified protocol without forcing them outside their intended scope and why there exist no such protocols in the current literature.
> > >
> > > > Can you rank the contributions instead of listing them equally?
> > >
> > > The **main contribution is the theoretical framework that enables finding the optimal solution in a feasible time**. The secondary contribution is the beam variant.
> > >
> > > The main contribution is composed of **three necessary components**: the algorithm, the heuristics, and the quantities. These are **interdependent elements required to achieve the overall goal** of finding the optimal solution in a feasible time. As such, they cannot be ranked. The algorithm relies on two heuristics to reduce the search space. Without the heuristics, the algorithm would not be able to complete the search in a feasible time. Similarly, the heuristics without the algorithm design (e.g., pruning based on the minimum IoU, alternating heuristic variants) would not be effective, and the search space would remain prohibitively large. Finally, the heuristics rely on the quantities identified in the paper. Without these quantities, the heuristics would not be defined, and without heuristics, the algorithm would not be feasible.
> > > As such, these components cannot be strictly “ranked”, as they are all necessary to achieve the final goal. One could argue that the fundamental building block is represented by the theoretical contribution on the quantities and the decomposed IoU. We agree with this perspective. However, identifying these elements alone is not sufficient to solve the problem. Both the heuristics and the algorithm require non-trivial design choices to leverage these quantities and make the search tractable.

---

### Official Review · Reviewer_9QUJ · 2026-03-13

**Soundness:** 3
**Presentation:** 3
**Significance:** 1
**Originality:** 3
**Overall Recommendation:** 4
**Confidence:** 3

**Summary:**

This paper develops a method for improving compositional explanations of neurons, showing that via a clever decomposition of the main scoring technique, one can develop an algorithm to always return the optimal explanation for a given neuron. Compositional explanations of neurons especially for vision models are ways to explain what a given neuron is doing. Given an input image and a segmentation atlas on that image along with concept annotations on each segment, we can identify which neurons activate the most for each segment and which neurons activate the most for each concept. Compositionality then allows you to broaden or focus the concept space by taking logical combinations of concepts. For example, if you have a concept for "eye" and a concept for "nose", you can take the logical OR of those concepts and try to find for a given neuron which composed explanation gives the highest score with respect to the existence of that composition in the image. The main scoring technique for compositional explanations is the intersection over union (IoU) score, which measures the overlap between the segments of the image that activate a given neuron and the segments of the image that correspond to a given concept or composition of concepts. Problem with IoU score and compositionality is that size of compositions grows very rapidly, so impossible to search for optimal explanations via brute force. Related work suggested a beam search, by taking the top k concepts and then taking the top k compositions of those concepts, and so on. This is a heuristic method which does not guarantee optimality. Another heuristic method is to try to estimate IoU for large beams. Here, authors develop a novel heuristic based on a clever decomposition of the IoU score. They then prove this novel heuristic is admissible for estimating the IoU score and therefore for finding $\arg\max_l IOU(l)$. They show that their method using the heuristic is guaranteed to return the most aligned explanation and therefore the optimal one.

**Compliance With Llm Reviewing Policy:**

Affirmed.

**Final Justification:**

I have appreciated the rebuttal conversation with the authors and think this paper represents an advance in the understanding of compositional explanations of neurons. I think that some of the discussion in the rebuttals between myself and other reviewers and the authors, once included in the revised version, will lead to a stronger paper. I believe my score remains appropriate given the scale of the advance in this paper.

**Key Questions For Authors:**

- What is the comparison of IOU scores?
- What is the distribution of IOU scores for a given concept?
- How do we know that the optimal explanations are better than the beam search explanations, and by how much?
- Is it novel to show that this heuristic finds optimal explanations? Breadth first search also does, but that's a bad heuristic!

**Limitations:**

Yes

**Strengths And Weaknesses:**

Strengths:
- Authors' method based on novel heuristic is clever and thorough, with high expressive power for exploring the space of logical combinations between concepts
- Really good commentary on the path dependency of "optimal" composed explanations and disjoint concepts.

Weaknesses:
- Main results in Table 1 show a comparison of authors' heuristic to related works, but focus on the statistic of number of beam nodes listed, expanded and estimated. In particular, it does not claim to show that authors' heuristic is finding much better IOUs than other methods.
- More on the commentary of memory intensity of this and other methods before section 3.2
- I don't understand the $L_\to$ for left and right sides of a composed annotation.

---

> ### Author Rebuttal · Authors · 2026-03-31
>
> > W1/Q1/Q2. Table 1 and comparison/distribution of IOU scores
>
> A1. Optimal explanations are better than beam search explanations when they find and express a stronger alignment (i.e., a higher IoU). Regarding the comparison in terms of IoU, there are two separate answers: one for the optimal algorithm and one for the beam variant guided by our heuristic.
>
> In the first case, **the difference between the optimal solution found by our algorithm and the solution found by the beam search used in previous work can be found in Table 6 (Appx.)**. The main text (first par. of Sect. 3.2) refers to and cites Table 6 when discussing the results. In the camera-ready version, we will replace Table 2 with Table 6. As discussed below, all the beam variants are guaranteed to find the same solution, since they all explore the same restricted state space. Therefore, the “Beam IoU” column in Table 6 refers to the solutions found by all beam-based variants, while the “Optimal IoU” column refers to the solutions found by our optimal algorithm in the full state space.
>
> **Regarding the beam variants, please note that MMESH, vanilla, and our beam variant all find the same solution, since they have access to the same state space (i.e., the space reachable by beam search)**. Beam search builds the search tree one level at a time. Because both MMESH and our beam variant use admissible heuristics, they are guaranteed to identify, at each level, the same best $𝑏$ candidates as the exhaustive search (vanilla) within the restricted search space. Therefore, the reachable state space at each step is the same, and consequently the final solution is guaranteed to be the same. This result is formally proven in La Rosa et al., (2023) and in Appx. F. The experiments we ran automatically computed the IoU for all variants. However, since the values are identical, we did not report them in the paper.  We will add this explanation to the paper.
>
> Regarding the distribution of IoU, note that compositional explanations fix the neuron and search over logical combinations of concepts rather than fixing the concepts and searching for the neurons that recognize them. This is because the state space of all the possible logical combinations is too large to be enumerated (which is the core motivation of our work). **Listing all possible combinations and computing the IoU for each to derive a distribution is therefore impractical**. More generally, based on prior literature, **we expect that a specific combination is recognized by only a small number of neurons (i.e., IoU in the range 0.04-0.15), while yielding IoU values close to zero for most other neurons**.
>
>
>
> > Q3. Is it novel to show that this heuristic finds optimal explanations? Breadth first search also does, but that's a bad heuristic!
>
> A.3 Yes, **it is novel, and we argue that it is the strongest contribution of this theoretical work. While breadth-first search can also recover the optimal solution, its application is considered computationally infeasible in practice in these settings due to the size of the state space**. Our contribution is precisely to make this search tractable while preserving optimality guarantees. Conversely, previous literature on the topic relied on beam search. As such, to the best of our knowledge, this is the first work that proposes an algorithm that ensures optimality in this setting and empirically shows that explanations produced by previous approaches are not optimal.
>
>
>
> > Q4. I don't understand the L → for left and right sides of a composed annotation.
>
> A4. By Assumption 1, **concepts are combined incrementally** to form labels **(e.g., ((((A ⊕ B) ⊕ C) . . .) ⊕ Z))**. This means that, at each step, **one can identify the left-hand side and the right-hand side of an operator by considering the outermost parentheses in the label**. For example, in a label like (A ⊕ B) the left side L←corresponds to A, and the right side L→corresponds to B. In a label like (A ⊕ B) ⊕ C) the left side   L←corresponds to  (A ⊕ B),  and the right side L→corresponds to C, and so on. The arrow direction indicates whether it is the left side or the right side.
>
>
>
> > Q5. commentary of memory intensity of this and other methods
>
> A5.  Thank you for the suggestion. We will add the following information. Regarding memory, the dominant factors are the neuron activation matrix and the concept tensor. Heuristic-related information accounts for only a small fraction of the memory. The impact of storing the frontier is negligible compared to these quantities, since it can be represented by tuples of strings and numbers. Additionally, the optimal algorithm does not need to store the beam (concept) tensors. As a result, **the optimal algorithm is the lightest in terms of memory** (e.g., ~8GB vs ~17GB for beam search on the most complex settings). This footprint can be further reduced (by ~4GB) by loading the annotations from disk on demand rather than keeping them in memory, as described in the paper.

---

> > ### Author Rebuttal · Reviewer_9QUJ · 2026-04-01
> >
> > I thank the authors for their rebuttal.
> >
> > I am still having difficulties understanding whether the contribution of this paper is (1) to identify a heuristic which is provably optimal and indeed identifies better IOUs than previous approaches or (2) to propose an improvement to beam search which is less memory intensive and provably optimal but in practice merely finds the same solutions as previous approaches. It seems that the content of the rebuttal suggests the paper is doing both at once, but I don't understand how.
> >
> > The decomposition heuristic finds provably optimal compositions, as is shown in the paper. But the range of possible IOUs as the rebuttal notes is quite small, which is why I asked about the distribution of IOU scores. For Figure 2 Unit 39 for example is the difference between optimal IOU 0.047 and beam IOU 0.043 meaningful? What about Figure 2 Unit 41 where the IOUs are identical?
> >
> > I note another reviewer's questions about the computational complexity of the decomposition heuristic as being interesting. How much of an improvement is the decomposition heuristic over breadth first search?

---

> > > ### Author Response · Authors · 2026-04-02
> > >
> > > Thank you for your contribution to improving our work and for your follow-up. Below, we address and clarify the remaining concerns.
> > >
> > > > I am still having difficulties understanding whether the contribution of this paper is (1) to identify a heuristic which is provably optimal and indeed identifies better IOUs than previous approaches or (2) to propose an improvement to beam search which is less memory intensive and provably optimal but in practice merely finds the same solutions as previous approaches. It seems that the content of the rebuttal suggests the paper is doing both at once, but I don't understand how.
> > >
> > > The paper proposes a decomposition of IoU, associated heuristics, and a novel algorithm to find solutions that are provably optimal and associated with better IoUs than previous approaches (when those do not find the optimal solution). **This corresponds to the reviewer's point (1) and represents the main contribution of the paper**.
> > > Specifically, the algorithm uses two heuristics: one to estimate the label IoU and one to estimate the path IoU (i.e., the IoU discoverable by paths starting from a given label). In this context, our beam variant is a secondary (or side) contribution, where we show that the label-IoU heuristic (i.e., which is **exactly one of the two heuristics used by the optimal algorithm** and it is used to estimate the IoU of the possible succesors of the best *b* beam candidates) can also be applied within beam search to obtain practical benefits. Note that the algorithm for the beam search is available in the appendix. This corresponds to the reviewer's point (2) and it is a showcase of the flexibility of that heuristic. However, the core theoretical contribution remains point (1), which is the focus of Sections 3.2 and 3.3.
> > >
> > > > For Figure 2 Unit 39, is the difference between optimal IoU 0.047 and beam IoU 0.043 meaningful? What about Unit 41 where the IoUs are identical?
> > >
> > > Regarding identical IoU values, as mentioned in our response to reviewer QvXr, this is due to the limited decimal precision used in visualization. At higher precision, the scores are not identical (e.g., for Unit 41: 0.06867 for MMESH vs 0.06905 for the optimal solution). We will increase the precision in the camera-ready version.
> > >
> > > For cases like Unit 39, where totally different explanations achieve similar IoU values, we believe this is an interesting effect related to superposition. Superposition is the phenomenon where neurons may encode multiple relationships or combinations of concepts in their activations. Compositional explanations aim to (partially) capture this behavior, but the extent to which this is possible depends on the configuration (e.g., maximum explanation length and assumptions).
> > > In these scenarios, neurons may encode multiple parallel or more complex relationships that cannot be fully captured under the current configuration. Addressing this would require increasing the maximum explanation length, returning multiple solutions within a threshold $t$, relaxing assumptions, or supporting more expressive logical operators. However, increasing length or returning multiple solutions introduces a trade-off with interpretability, while the other directions require further research. Among these, our personal perspective is that to capture such behaviors, we would need to both go beyond the incrementality assumption and increase the length of the explanation, even if that would increase the burden for users.
> > >
> > > More in general, **the objective of compositional explanations is to find the maximum possible alignment**. Therefore, even small IoU differences indicate strictly better alignment under the defined objective and are thus meaningful within this framework. As mentioned in another response, we will include this discussion in the paper.
> > >
> > > >How much of an improvement is the decomposition heuristic over breadth-first search?
> > >
> > > **The improvement is substantial**. Breadth-first search is not feasible for most configurations. In the most complex settings, BFS would require evaluating approximately $2.8\times10^{14}$ nodes. Even under a highly optimistic assumption of 0.005 seconds per node (which is an underestimate, in our test it usually takes 0.01s or more to compute the label matrix and the IoU), and ignoring the problems related to memory and logical equivalence discussed in the appendix, this would require approximately 388 888 888 hours per neuron, making it clearly impractical. This time is similar for the middle complexity case. **This is the unfeasibility problem** we described in the paper and that our method addresses.
> > > The only cases where BFS is feasible are the lowest complexity settings (e.g., with very few concepts (19)), where it still requires ~140 seconds per neuron. This is already around $1000\times$ slower than the proposed optimal algorithm.

---

### Decision · Program_Chairs · 2026-04-30

**Decision:**

Accept (spotlight)

**Comment:**

This paper develops a method for improving compositional explanations of neurons, showing that via a clever decomposition of the main scoring technique, one can develop an algorithm to always return the optimal explanation for a given neuron. Compositional explanations of neurons especially for vision models are ways to explain what a given neuron is doing. Given an input image and a segmentation atlas on that image along with concept annotations on each segment, we can identify which neurons activate the most for each segment and which neurons activate the most for each concept.

The paper initially received mixed reviews. After the rebuttal, reviewers converged to weak accept to accept. It is an interesting approach and the result is significant, hence I believe this deserves an oral presentation as it is the most sound explainability paper I have area chaired in a couple of years and it's worth for it to have a broader audience. The authors should clarify their contributions and limitations as reviewers suggested for the final version.

Note that related IoU decomposition techniques exist in the past, albeit in different setups in Bayesian inference. It would be nice for the authors to cite these 2 works:

F. Li et al. Composite Statistical Inference for Semantic Segmentation. CVPR 2013
S. Nowozin. Optimal Decisions from Probabilistic Models: the Intersection-over-Union Case. CVPR 2014

to properly attribute the history of IoU decomposition. The existence of these work do not hinder the novelty of the proposed work as they are from very different contexts.